# RECTOR: Masked Region-Channel-Temporal Modeling for Cognitive Representation Learning

## Abstract

Affective and cognitive disorders are characterized by complex, distributed brain network dynamics across distinct functional regions, channels, and time, posing a significant challenge to learning robust representations for clinical diagnosis. We introduce **RECTOR** (Masked **Re**gion–**C**hannel–**T**emp**or**al Modeling), the first end-to-end, self-supervised brain region modeling framework that unifies region, channel, and temporal representation learning in a single architecture. At its core is **RECTOR-SA**, a novel hierarchical self-attention mechanism. It efficiently models region-channel-temporal interactions in a block-wise paradigm, incorporating both anatomical priors and dynamic functional gating. It further integrates **RECTOR-Mask**, a novel masking strategy that generates diverse region-channel-temporal views to establish a challenging pretext task. The self-supervision is driven by **NC$^2$-MM**, our learning objective. It synergistically encourages the encoder to learn both predictive and consistent representations across different views. Finally, we introduce **RCReg**, a tailored regularization on region–channel tokens. It prevents trivial region features and enables the model to explicitly learn and disentangle both region-common and channel-specific representations. Across diverse benchmarks, RECTOR sets a new state-of-the-art in EEG emotion recognition and sEEG task-engagement classification. In addition, it achieves superior computational efficiency in spatio-temporal self-attention, demonstrates strong potential for large-scale pre-training, and provides interpretable, multi-view insights into neural representations at both brain region and channel levels.

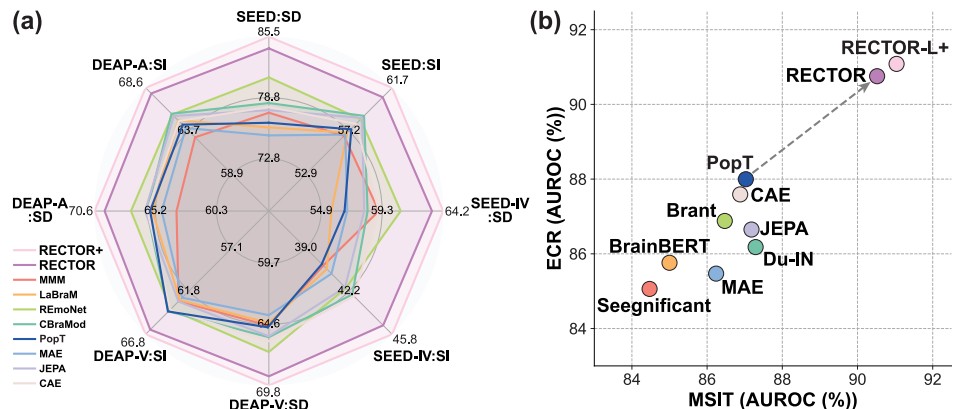

Figure 1: (a) RECTOR consistently outperforms leading EEG models on emotion recognition benchmarks (Weighted F1 Score (%)). SD: subject-dependent; SI: subject-independent; V: valence; A: arousal. (b) RECTOR outperforms state-of-the-art sEEG models on task engagement classification.

## 1 Introduction

Affective disorders and cognitive impairments affect hundreds of millions worldwide, substantially diminishing quality of life and imposing a heavy socio-economic burden (Organization, 2022). Despite decades of research, clinical assessment still relies on patient self-reports and behavioral

observations, which are inherently subjective and can delay diagnosis and treatment (Kret & Ploeger, 2015; Goschke, 2014). In contrast, neurophysiological signals such as electroencephalography (EEG) and stereo-EEG (sEEG) offer direct, noninvasive/invasive windows into the brain's electrical activity. Analysis of these signals promises objective biomarkers for affective and cognitive states, enabling novel brain–computer interfaces (BCIs) and personalized interventions (Houssein et al., 2022; Drane et al., 2021). However, EEG/sEEG data present formidable challenges for learning affective and cognitive neural representations.

Mood and cognitive disorders arise from distributed interactions across different brain regions, requiring models that explicitly capture region-level dynamics (Lindquist et al., 2012; Menon, 2011; Pessoa, 2017). Although recent methods aimed to capture regional brain dynamics, they exhibit several key limitations. Many approaches using bihemispheric modeling (Li et al., 2020; Huang et al., 2021), graph neural networks (Ding et al., 2023; Ye et al., 2022; Jin et al., 2024; Qiu et al., 2023) and transformers (Zheng et al., 2024; Yi et al., 2023), attempt to learn regional embeddings from scratch but lack a dedicated region learning framework to guide this process, resulting in suboptimal representations. Other methods oversimplify brain topology by performing population-level spatial encoding (Chau et al.; Mentzelopoulos et al., 2024), treating all channels as a single global region and thus ignoring the brain's established functional network architecture. Moreover, recent models like MMM (Yi et al., 2023) and Du-IN (Zheng et al., 2024) discard the fine-grained, channel-specific dynamics that are essential for preserving high electrophysiological fidelity.

The sparse labeling of EEG/sEEG data makes self-supervised learning (SSL) an essential paradigm (Rafiei et al., 2024; Chien et al., 2022; Li et al., 2022; Wang et al., 2023b). However, the predominant masked modeling (MM) with random masking (Zheng et al., 2024; Zhang et al., 2023; Jiang et al., 2024b;a; Wang et al., 2024) often creates a simple pretext task. This encourages models to learn superficial spatio-temporal shortcuts rather than generalizable features, limiting performance on downstream tasks (Assran et al., 2023). This risk is particularly acute for neural data, where affective and cognitive contexts induce significant spatial heterogeneity and temporal variability (Segal et al., 2023; Wei et al., 2023). While contrastive learning (CL) can promote discriminative features, existing methods fail to synergistically combine it with MM, often applying them in separate architectures or training stages (Jiang et al., 2024a). Moreover, most SSL approaches (Yi et al., 2023; Jiang et al., 2024a; Wang et al., 2025; 2023a) lack explicit constraints to structure the representation space, which can limit the quality and robustness of the final embeddings.

While transformers are prevalent in SSL for EEG/sEEG, a key challenge remains in adapting self-attention (SA) to effectively capture joint spatio-temporal interactions (Rafiei et al., 2024). Current spatial encoding approaches neglect temporal dynamics (Yi et al., 2023; Jiang et al., 2024a), while sequential or criss-cross schemes learns spatial and temporal interactions in isolation (Wang et al., 2025). Although full spatio-temporal SA models dense interactions (Jiang et al., 2024b; Zhang et al., 2023), it is not only computationally inefficient due to the large number of tokens but also risks amplifying uninformative relationships while obscuring critical dependencies. Furthermore, integrating anatomical priors and brain functional dynamics into SA remains a non-trivial challenge.

To tackle these critical challenges in cognitive neural representation learning, we introduce **RECTOR**: Masked **Re**gion-**C**hannel-**T**emp**or**al Modeling. RECTOR makes the following key contributions:

1. It presents **RECTOR-SA**, a novel and efficient hierarchical self-attention mechanism that incorporates anatomical priors to model region-channel-temporal interactions and disentangle region-channel representations The dynamic functional gating serves as a learned denoiser by pruning spurious, noisy connections between channels/regions.

2. It introduces a new, holistic SSL paradigm in a single architecture built on (a) **RECTOR-Mask** that generates diverse region-channel-temporal views to establish a challenging pretext task on the hierarchical neurophysiological space, (b) **NC$^2$-MM** that jointly encourages the encoder to learn both "predictive" and "consistent" representations in one forward step, and (c) **RCReg** that explicitly guides and unmixes the region-channel representation learning.

3. To our knowledge, it is the first end-to-end, self-supervised **region learning framework** that explicitly models brain region representations and their region-channel-temporal interactions in alignment with the brain's anatomical and functional networks.

Across diverse benchmarks, RECTOR sets a new state-of-the-art in EEG emotion recognition and sEEG task engagement classification by consistently outperforming leading supervised and self-

supervised methods. This superior performance is coupled with a remarkable computational efficiency in spatio-temporal SA and a strong potential for large-scale pre-training. Moreover, RECTOR offers deep interpretability through multi-level visualizations of neural dynamics, which align with established neurophysiology underlying cognitive conditions. This trifecta of accuracy, efficiency, and interpretability positions RECTOR as a powerful framework for advancing neurocognitive diagnostics and enabling adaptive interventions.

## 2 METHODOLOGY

### 2.1 PROBLEM SETTING AND FORMULATION

We model an EEG/sEEG trial as $\mathbf{X}_{\text{input}} \in \mathbb{R}^{C \times S}$, where $C$ denotes the number of recording channels and $S$ the number of time samples. The problems of task engagement classification and emotion recognition are formulated as binary/multi-class classification tasks with a label $y$ inferred from $\mathbf{X}_{\text{input}}$. We divide $\mathbf{X}_{\text{input}}$ into $N$ non-overlapping patches: $\mathbf{X}_{\text{patch}} = \left[ \mathbf{x}_1^\top; \dots; \mathbf{x}_N^\top \right] \in \mathbb{R}^{N \times P}$, where $P$ is the temporal patch length, $T = \frac{S}{P}$ is the number of time segments, and $N = CT$. Patches are mapped to $d$-dimensional tokens through a linear patch projection $\mathbf{W_p} \in \mathbb{R}^{P \times d}$: $\mathbf{X}^C = \mathbf{X}_{\text{patch}} \mathbf{W_p}$. The resulting $\mathbf{X}^C$ is the input channel-time tokens processed by RECTOR.

### 2.2 RECTOR

**RECTOR** (Masked **Re**gion-**C**hannel-**T**emp**o**ral Modeling) introduces a new, holistic self-supervised learning (SSL) paradigm for modeling region-channel-temporal representations. This requires the invention of four new, co-designed components: (1) **RECTOR-SA**, the efficient, hierarchical self-attention building block for the transformers (Fig. 2(b) and Section 2.3); (2) **RECTOR-Mask**, a structured multi-view masking strategy for creating a challenging pretext task (Fig. 2(a) and Section 2.4); (3) **NC²-MM**, an SSL objective that encourages the encoder to learn both "predictive" and "consistent" representations (Fig. 2(a) and Section 2.5); (4) **RCReg**, a regularization on region and channel tokens to guide and unmix the region-channel representation learning (Fig. 2(a) and Section 2.6). RECTOR comprises four RECTOR-Transformers (see Appendix B.8) using **RECTOR-SA** as shown in Fig. 2(a) and 2(b): context encoder, target encoder (an EMA replica of context encoder), predictor, and decoder.

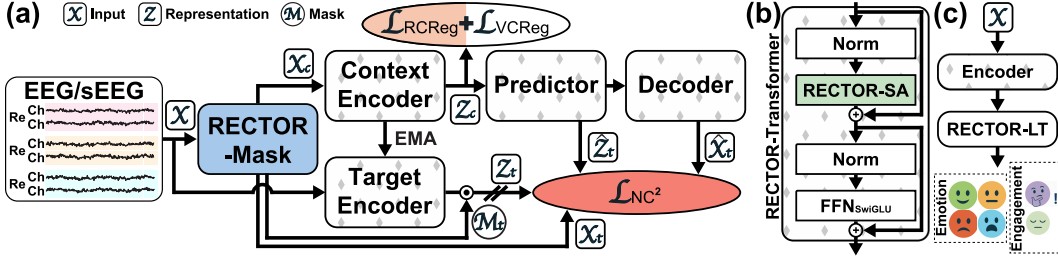

Figure 2: **RECTOR**'s: (a) SSL pipeline with **RECTOR-Mask**, **NC²-MM**, and **RCReg**; (b) Transformer block with **RECTOR-SA** in the context/target encoders, predictor, and decoder. (c) Fine-tuning pipeline with the pre-trained (active) encoder and a decoding head RECTOR-LT (see Appendix B.5) for emotion recognition and task engagement classification.

**Objectives** The objective of RECTOR's SSL is to train a robust encoder that learns rich and generalized region-channel-temporal neural representations. After that, this pre-trained encoder is used for fine-tuning in the downstream tasks, including emotion recognition and task engagement classification, as shown in Fig 2(c).

**Input Construction** Given the input channel-time tokens $\mathbf{X}^C \in \mathbb{R}^{CT \times d}$, we concatenate extra region tokens $\mathbf{X}^R \in \mathbb{R}^{RT \times d}$ The token sequence fed to RECTOR is: $\mathbf{X} = \left[ \mathbf{X}^C; \mathbf{X}^R \right] \in \mathbb{R}^{((C+R)T) \times d}$. $\mathbf{X}^R$ model *hierarchical* spatial structure to enable region-aware representation learning.

**Pipelines** The SSL pipeline in Fig. 2(a) illustrates that $\mathbf{X}$ is partitioned **RECTOR-Mask** into a context block $\mathbf{X_c}$ and target blocks $\mathbf{X_t}$ with target masks $\mathbf{M_t}$. Then $\mathbf{X_c}$ is forwarded through the

context encoder, predictor, and decoder to obtain the context representations $\mathbf{Z_c}$, predicted target representations $\hat{\mathbf{Z}}_\mathbf{t}$, and decoded target inputs $\hat{\mathbf{X}}_\mathbf{t}$. And $\mathbf{X}$ and $\mathbf{M_t}$ are forwarded through the target encoder to obtain the target representations $\mathbf{Z_t}$. Together $\mathbf{Z_t}$, $\hat{\mathbf{Z}}_\mathbf{t}$, $\mathbf{X_t}$, and $\hat{\mathbf{X}}_\mathbf{t}$ are used to calculate the NC$^2$-MM, RCReg, and VCReg losses for SSL. The fine-tuning pipeline is illustrated in Fig 2(c), showing that $\mathbf{X}$ is forwarded through the SSL pre-trained, active encoder and a decoding head RECTOR-LT (see Appendix B.5) for downstream classification using cross-entropy loss.

**Encoding, Predicting, and Decoding Processes**   The context encoder $f(\,\cdot\,;\theta)$ maps the unmasked context patches $\mathbf{X_c}$ to latent features $\mathbf{Z_c} = f(\mathbf{X_c}, \mathbf{P_c};\theta)$, where $\mathbf{P_c}$ is the standard positional embedding of the context block. An EMA-copy of this encoder, $\tilde{f}(\,\cdot\,;\tilde{\theta})$, processes the full set of patches $\mathbf{X}$ to yield the target representations $\mathbf{Z_t}$ masked by $\mathbf{M_t}$: $\mathbf{Z_t} = \tilde{f}\left(\mathbf{X}, \mathbf{P};\tilde{\theta}\right) \odot \mathbf{M_t}$, where $\mathbf{P}$ is the standard positional embeddings (see Fig. 2).

Building on these embeddings, a predictor $g(\,\cdot\,;\psi)$ uses the context features to predict target representations $\hat{\mathbf{Z}}_\mathbf{t}$. Finally, a decoder $h(\,\cdot\,;\phi)$ reconstructs the original masked inputs from these predictions to $\hat{\mathbf{X}}_\mathbf{t}$. Both the predictor and the decoder are conditioned on the target information using the positional embeddings of the target blocks $\{\mathbf{P}_{\mathbf{t}^{(m)}}\}_{m\in\mathcal{M}}$ for each masked domain $m$ in the domain set $\mathcal{M}$ (see RECTOR-Mask in Section 2.4). In compact form:

$$\hat{\mathbf{Z}}_\mathbf{t} = g\left(\mathbf{Z_c}, \{\mathbf{P}_{\mathbf{t}^{(m)}}\}_{m\in\mathcal{M}};\psi\right), \quad \hat{\mathbf{X}}_\mathbf{t} = h\left(\hat{\mathbf{Z}}_\mathbf{t}, \{\mathbf{P}_{\mathbf{t}^{(m)}}\}_{m\in\mathcal{M}};\phi\right) \tag{1}$$

During SSL, we extend target conditioning into the encoder itself–rather than only in the predictor/decoder–by concatenating *region positional embeddings* to the input tokens. We refer to this mechanism as Region Conditioning (Cd) (see Fig. 3). Concretely, the standard context and target encoding steps in SSL become:

$$\mathbf{Z_c} = f\left(\mathbf{X_c}, \mathbf{P_c};\theta\right) \;\rightarrow\; \mathbf{Z_c} = f\left(\left[\mathbf{X_c}; \{\mathbf{P}_{\mathbf{t}^{(m)}}^{\mathrm{Cd}}\}_{m\in\mathcal{M}}\right], \mathbf{P_c};\theta\right)$$
$$\mathbf{Z_t} = \tilde{f}\left(\mathbf{X}, \mathbf{P};\tilde{\theta}\right) \odot \mathbf{M_t} \;\rightarrow\; \mathbf{Z_t} = \tilde{f}\left(\left[\mathbf{X}; \mathbf{P}_\mathbf{c}^{\mathrm{Cd}}\right], \mathbf{P};\tilde{\theta}\right) \odot \mathbf{M_t} \tag{2}$$

where $\{\mathbf{P}_{\mathbf{t}^{(m)}}^{\mathrm{Cd}}\}_{m\in\mathcal{M}}$ and $\mathbf{P}_\mathbf{c}^{\mathrm{Cd}}$ are the region positional embeddings for each target or context block. At fine-tuning and inference time, we augment the encoder with region information by concatenating the full set of region positional embeddings $\mathbf{P}^{\mathrm{Cd}}$:

$$\mathbf{Z} = f\left(\mathbf{X}, \mathbf{P};\theta\right) \;\rightarrow\; \mathbf{Z} = f\left(\left[\mathbf{X}; \mathbf{P}^{\mathrm{Cd}}\right], \mathbf{P};\theta\right) \tag{3}$$

where $\mathbf{X}$ denotes the input patch matrix and $\mathbf{Z}$ the resulting encoded representations. This *region-aware* encoding, guided by target information, enables the model to learn rich region-specific representations directly in SSL.

## 2.3   REGION-CHANNEL-TEMPORAL SELF-ATTENTION (RECTOR-SA)

**Motivation**   Standard self-attention (SA) often treats EEG/sEEG channels as a simple sequence, overlooking the brain's inherent regional organization and failing to explicitly model region-level interactions. Moreover, a naive spatio-temporal SA is computationally prohibitive, as the quadratic complexity scales with the number of channel-time tokens. To efficiently capture joint region–channel–temporal representations, we propose **RECTOR-SA** (Fig. 3). This novel hierarchical SA is grounded in neuroanatomy and uses dynamic functional attention for efficient block-wise computation. RECTOR-SA innovatively disentangles representations into two distinct, parallel streams: region tokens (for region-common, high-level, shared dynamics) and channel tokens (for channel-specific, fine-grained, local dynamics). A data-driven top-$p$ gate acts as a learned denoiser, pruning spurious, noisy connections and focusing on the real signals between channels/regions.

**Brain Partitioning**   Specifically, we partition channels/electrodes in EEG/sEEG into brain regions. The electrode and region layouts for our EEG datasets are illustrated in Fig. 3(b) For EEG datasets in this work, sensors are arranged according to the international 10-20 system and grouped into eleven anatomically defined regions (Alarcao & Fonseca, 2017). For sEEG datasets, each channel is assigned to one of the 30 brain regions using the Electrode Labeling Algorithm (Peled et al., 2017). A detailed breakdown of channels per region is provided in Table 26 (see Appendix E for details on brain partitioning and electrode maps). RECTOR-SA is computed according to these region–channel assignments as well as temporal alignment.

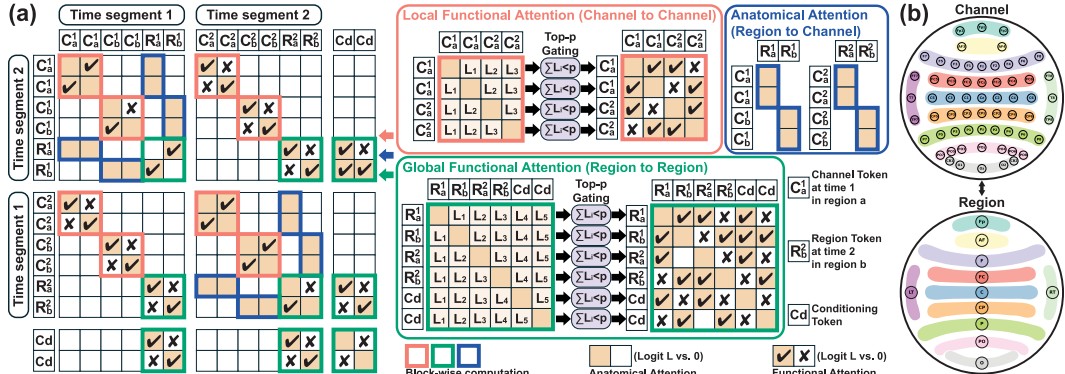

Figure 3: **RECTOR-SA**. (a) **Anatomical Attention**, **Local Functional Attention**, and **Global Functional Attention** are used for modeling region-channel-temporal interactions with block-wise computation. (b) Electrode and region maps of EEG. (Top) Channel layout, with colored overlays indicating anatomically defined functional regions. (Bottom) Region-level abstraction represented in our model by extra region tokens.

**RECTOR-SA**    The scaled dot-product SA can generally be written as:

$$\mathbf{Q} = \mathbf{X_Q W_Q} \quad \mathbf{K} = \mathbf{X_K W_K} \quad \mathbf{V} = \mathbf{X_V W_V} \quad \mathbf{L} = \frac{\mathbf{QK}^\top}{\sqrt{d}} \quad \mathbf{Z} = \mathrm{softmax}(\mathbf{L} \odot \mathbf{M_{Attn}})\mathbf{V} \quad (4)$$

Instead of computing the full SA on $\left[\mathbf{X}^{\mathrm{C}}; \mathbf{X}^{\mathrm{R}}; \mathbf{P}^{\mathrm{Cd}}\right]$, RECTOR-SA splits them into disjoint blocks and uses different $\mathbf{X_Q}, \mathbf{X_K}, \mathbf{X_V},$ and $\mathbf{M_{Attn}}$ in **Anatomical Attention**, **Local Functional Attention**, and **Global Functional Attention** during $\mathbf{QKV}$ and $\mathbf{M_{Attn}}$ operations (see Appendix B.1 for corresponding softmax$(\cdot)$ efficient computation). Let $r \in \{1, \cdots, R\}$ be the index of region and $t \in \{1, \cdots, T\}$ be the index of time segment, and we can define $\mathbf{X}^{\mathrm{R}}_{(r,t)}$ as the region token of region $r$ and time segment $t$, and $\mathbf{X}^{\mathrm{C}}_{(r,t)}$ as the set of channel tokens of time segment $t$ that belongs to region $r$.

**Anatomical Attention (Fig 3(a))**    We impose a region–channel topology on the attention pattern to embed anatomical priors. It is computed via Eq. 4 with

$$(1) \, \forall \, r, t, \ \mathbf{X_Q} = \mathbf{X}^{\mathrm{R}}_{(r,t)}, \, \mathbf{X_K} = \mathbf{X_V} = \mathbf{X}^{\mathrm{C}}_{(r,t)}, \, \mathbf{M_{Attn}} = \mathbf{1}, \text{ and}$$
$$(2) \, \forall \, r, t, \ \mathbf{X_Q} = \mathbf{X}^{\mathrm{C}}_{(r,t)}, \, \mathbf{X_K} = \mathbf{X_V} = \mathbf{X}^{\mathrm{R}}_{(r,t)}, \, \mathbf{M_{Attn}} = \mathbf{1} \quad (5)$$

It allows a channel token to only attend to its own region token at the same time slice.

**Functional Attention**    To capture dynamic functional channel interactions alongside region dependencies over time, we further split SA into two gated streams:

**(1) Local Functional Attention (Fig 3(a))**: Let $P_{(r)}$ be the number of channels in region $r$. Denoting $i_{(r)} \in \{1, \cdots P_{(r)}T\}, N^{P_{(r)}T \backslash i_{(r)}} = \{1, \ldots P_{(r)}T\} \backslash \{i_{(r)}\}$, it is computed via Eq. 4 using

$$\forall \, r, \ \mathbf{X_Q} = \mathbf{X_K} = \mathbf{X_V} = \mathbf{X}^{\mathrm{C}}_{(r,:)}$$
$$\forall \, r, i_{(r)}, \ [\mathbf{M_{Attn}}]_{i_{(r)}, i_{(r)}} = 1, \, [\mathbf{M_{Attn}}]_{i_{(r)}, N^{P_{(r)}T \backslash i_{(r)}}} = \text{top-}p\left([\mathbf{L}]_{i_{(r)}, N^{P_{(r)}T \backslash i_{(r)}}}\right) \quad (6)$$

**(2) Global Functional Attention (Fig 3(a))**: Denoting $j \in \{1, \cdots RT + K\}, N^{RT+K \backslash j} = \{1, \ldots RT + K\} \backslash \{j\}$, it is computed via Eq. 4 using

$$\mathbf{X_Q} = \mathbf{X_K} = \mathbf{X_V} = \left[\mathbf{X}^{\mathrm{R}}; \mathbf{P}^{\mathrm{Cd}}\right]$$
$$\forall \, j, \ [\mathbf{M_{Attn}}]_{j, j} = 1, \, [\mathbf{M_{Attn}}]_{j, N^{RT+K \backslash j}} = \text{top-}p\left([\mathbf{L}]_{j, N^{RT+K \backslash j}}\right) \quad (7)$$

For functional attention, $\mathbf{M_{Attn}}$ denotes *dynamic*, row-wise attention masks that always retain the diagonal and top-$p$ off-diagonal logits in each row (see Appendix B.6 for details on top-$p$ gating). See Appendix B.8 for details on the RECTOR-Transformer block using multi-head RECTOR-SA.

## 2.4 REGION-CHANNEL-TEMPORAL MASK (RECTOR-MASK)

The core challenge of creating a structured pretext task that operates on the hierarchical neurophysiological space was previously unsolved. To achieve SSL across different dynamics for robust representation learning, we introduce a new structured multi-view masking strategy called **RECTOR-Mask**. We first draw a context block $\mathbf{X_c}$ from $\mathbf{X}^C$ by randomly selecting a contiguous sub-matrix whose spatial and temporal extents are governed by two range ratios: $\rho_{\mathbf{c}}$ (channel) and $\eta_{\mathbf{c}}$ (time). Its binary mask is $\mathbf{M_c} \in \{0, 1\}^{C \times T}$. The patches outside $\mathbf{X_c}$ constitute the target blocks $\mathbf{X_t}$, and this residual patch grid is partitioned into five mutually exclusive domains: cross-region ($\times$r), cross-channel ($\times$c), cross-time ($\times$t), cross-region-time ($\times$rt), and cross-channel-time ($\times$ct) (Fig. 4(a)). For each domain $m \in \mathcal{M} = \{\times$r$, \times$c$, \times$t$, \times$rt$, \times$ct$\}$, we draw one rectangular target block $\mathbf{X}_{\mathbf{t}(m)}$ sampled from the range ratios $\rho_{\mathbf{t}(m)}$ and $\eta_{\mathbf{t}(m)}$ (see Appendix B.7 for details on sampling in RECTOR-Mask). The resulting binary mask is $\mathbf{M}_{\mathbf{t}(m)} \in \{0, 1\}^{C \times T}$. The split of input $\mathbf{X}^C$ using RECTOR-Mask is ($\odot$ denotes element-wise multiplication): $\mathbf{X_c} = \left[ \mathbf{X}^C \odot \mathbf{M_c}; \mathbf{X}^R \right]$, $\mathbf{X_t} = \left[ \{\mathbf{X}^C \odot \mathbf{M}_{\mathbf{t}(m)}\}_{m \in \mathcal{M}}; \mathbf{X}^R \right]$.

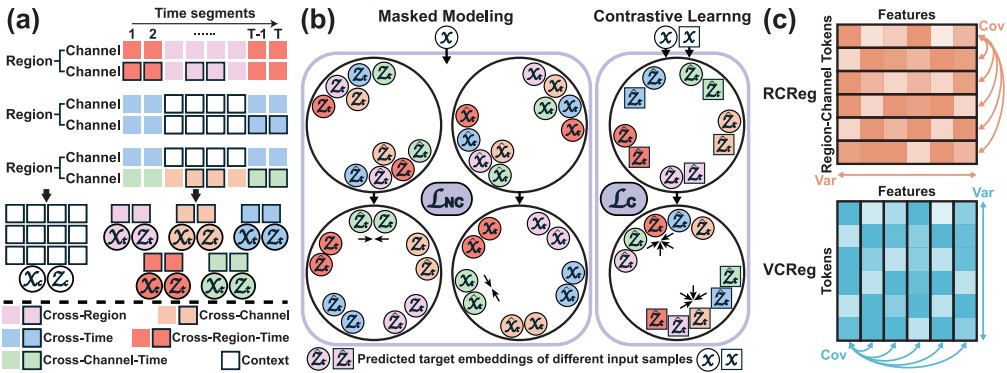

Figure 4: RECTOR's SSL is enabled by (a) **RECTOR-Mask**, which creates one context block and five target blocks from distinct domains for SSL; (b) **NC²-MM**, which jointly optimizes multi-view intra-sample masked modeling and inter-sample contrastive learning; (c) **RCReg** and VCReg, which explicitly structure the representational space and guide region-channel representation learning.

## 2.5 NON-CONTRASTIVE × CONTRASTIVE MASKED MODELING (NC²-MM)

We introduce **NC²-MM**: Non-Contrastive × Contrastive Masked Modeling (Fig. 4(b)). The overall SSL objective for NC²-MM is a composite loss $\mathcal{L}_{\mathrm{NC}^2}$ that combines a non-contrastive masked modeling loss ($\mathcal{L}_{\mathrm{NC}}$) with a contrastive loss ($\mathcal{L}_{\mathrm{C}}$): $\mathcal{L}_{\mathrm{NC}^2} = \mathcal{L}_{\mathrm{NC}} + \mathcal{L}_{\mathrm{C}}$.

**Non-Contrastive Loss** Our objective is to learn features that are both *semantically rich* for downstream tasks and *grounded in signal-level detail*. To achieve this, we combine two complementary losses: a **representation alignment loss** ($\mathcal{L}_{\mathrm{rep}}$) and an **input reconstruction loss** ($\mathcal{L}_{\mathrm{input}}$): $\mathcal{L}_{\mathrm{NC}} = \lambda_{\mathrm{input}} \mathcal{L}_{\mathrm{input}} + \lambda_{\mathrm{rep}} \mathcal{L}_{\mathrm{rep}}$. While $\mathcal{L}_{\mathrm{rep}}$ learns high-level features by aligning representations that can be directly used in downstream tasks, this approach is prone to representational collapse. The complementary $\mathcal{L}_{\mathrm{input}}$ provides a more principled mechanism to mitigate this risk by forcing the representations to remain informative and non-trivial to reconstruct the original input (see Appendix B.3). With the target inputs and representations $\{\mathbf{X}_{\mathbf{t}(m)}\}_{m \in \mathcal{M}}$, $\{\mathbf{Z}_{\mathbf{t}(m)}\}_{m \in \mathcal{M}}$, and the decoded/predicted target inputs and representations $\{\hat{\mathbf{X}}_{\mathbf{t}(m)}\}_{m \in \mathcal{M}}$, $\{\hat{\mathbf{Z}}_{\mathbf{t}(m)}\}_{m \in \mathcal{M}}$, let $|\mathcal{M}|$ be the number of target blocks and $\mathrm{SG}(\cdot)$ denotes stop-gradient, $\mathcal{L}_{\mathrm{input}}$ and $\mathcal{L}_{\mathrm{rep}}$ can then be formulated as (Fig. 4(b)):

$$\mathcal{L}_{\mathrm{input}} = \frac{1}{|\mathcal{M}|} \sum_{m \in \mathcal{M}} \sum_{n} \left\| \left[ \hat{\mathbf{X}}_{\mathbf{t}(m)} \right]_{n,:} - \left[ \mathbf{X}_{\mathbf{t}(m)} \right]_{n,:} \right\|_2^2$$

$$\mathcal{L}_{\mathrm{rep}} = \frac{1}{|\mathcal{M}|} \sum_{m \in \mathcal{M}} \sum_{n} \left\| \left[ \hat{\mathbf{Z}}_{\mathbf{t}(m)} \right]_{n,:} - \mathrm{SG}\left( \left[ \mathbf{Z}_{\mathbf{t}(m)} \right]_{n,:} \right) \right\|_2^2 \quad (8)$$

**Contrastive Loss** Each target block partitioned by RECTOR-Mask can be regarded as an *intra-sample view* that shares global context of the same input. To make the model learn consistent, global representations that are invariant to these spatio-temporal views, we add $\mathcal{L}_{\mathrm{C}}$ that attracts all target blocks belonging to the same input (Fig. 4(b)). Concretely, for a pair of predicted targets *from the*

*same input sample* $\left(\hat{\mathbf{Z}}_{\mathbf{t}^{(m_1)}}, \hat{\mathbf{Z}}_{\mathbf{t}^{(m_2)}}\right)$ projected by $\mathbf{W}_{\mathrm{C}}$, $\mathcal{L}_{\mathrm{C}}$ is formulated as:

$$\mathcal{L}_{\mathrm{C}} = \frac{\lambda_{\mathrm{C}}}{|\mathcal{M}|(|\mathcal{M}|-1)} \sum_{m_1 \in \mathcal{M}} \sum_{m_2 \in \mathcal{M}\setminus\{m_1\}} \sum_n \left\| \left[\hat{\mathbf{Z}}_{\mathbf{t}^{(m_1)}}\right]_{n,:} \mathbf{W}_{\mathrm{C}} - \left[\hat{\mathbf{Z}}_{\mathbf{t}^{(m_2)}}\right]_{n,:} \mathbf{W}_{\mathrm{C}} \right\|_2^2 \quad (9)$$

Crucially, the pretext tasks $\mathcal{L}_{\mathrm{rep}}$, $\mathcal{L}_{\mathrm{input}}$, and $\mathcal{L}_{\mathrm{C}}$ are together enabled by RECTOR-Mask within a single architecture in one forward pass. They also serve distinct purposes: $\mathcal{L}_{\mathrm{rep}}$ and $\mathcal{L}_{\mathrm{input}}$ force the encoder to learn representations that are "predictive" in embedding space and input space, respectively, while $\mathcal{L}_{\mathrm{C}}$ forces the encoder to learn "consistent" representations across different views.

### 2.6 REGION-CHANNEL TOKEN REGULARIZATION (RCREG)

We further augment our SSL with **RCReg**: Region-Channel Regularization. This module provides dedicated guidance for region and channel learning using a variance hinge loss $\mathcal{L}_{\mathrm{RCVar}}(\cdot)$ and a covariance regularization $\mathcal{L}_{\mathrm{RCCov}}(\cdot)$ on representations $\mathbf{Z}$ (Fig. 4(c)).

Let $B$ be the batch size, $N$ the number of tokens, and $d$ the embedding size. Given a batch of hidden representations $\mathbf{Z} = \mathbf{Z_c}\mathbf{W}_{\mathrm{Reg}} \in \mathbb{R}^{B \times N \times d}$, extracted by the context encoder $\mathbf{Z_c}$ and followed by a linear projection $\mathbf{W}_{\mathrm{Reg}}$, we can define RCReg as a combination of a variance hinge loss and a covariance loss on region-channel tokens $\mathcal{L}_{\mathrm{RCReg}} = \lambda_{\mathrm{RCVar}}\mathcal{L}_{\mathrm{RCVar}}(\mathbf{Z}) + \lambda_{\mathrm{RCCov}}\mathcal{L}_{\mathrm{RCCov}}(\mathbf{Z})$:

$$\mathcal{L}_{\mathrm{RCVar}}(\mathbf{Z}) = \frac{1}{N^{\mathrm{RC}}} \sum_{n=1}^{N^{\mathrm{RC}}} \max\left(0, \gamma - \sqrt{\frac{1}{BTd-1} \sum_{b=1}^{B} \sum_{t=1}^{T} \sum_{k=1}^{d} \left(\mathbf{Z}_{b,t,k}^{(n)} - \bar{\mathbf{Z}}^{(n)}\right)^2 + \epsilon}\right)$$

$$\mathcal{L}_{\mathrm{RCCov}}(\mathbf{Z}) = \frac{1}{N^{\mathrm{RC}}} \sum_{i \neq j} \left[\frac{1}{BTd-1} \sum_{b=1}^{B} \sum_{t=1}^{T} \sum_{k=1}^{d} (\mathbf{Z}_{b,t,k} - \bar{\mathbf{Z}})(\mathbf{Z}_{b,t,k} - \bar{\mathbf{Z}})^T\right]_{i,j}^2$$

$$(10)$$

Here, $N^{\mathrm{RC}}$ is the number of region and channel tokens. Sample means $\bar{\mathbf{Z}}$ are computed across specified sample axes.

**Motivation** RCReg operates on the token dimension to solve a new problem: hierarchical representational mixing. Its two components serve distinct purposes: $\mathcal{L}_{\mathrm{RCVar}}(\mathbf{Z})$ prevents collapse in the representations by ensuring that the variance of each region/channel token's embeddings stays above a threshold. $\mathcal{L}_{\mathrm{RCCov}}(\mathbf{Z})$ encourages region/channel token decorrelation in two conditions: (1) It minimizes the correlation between channel tokens from the same region. (2) It minimizes the correlation between each channel token and its corresponding region token. Therefore, RCReg compels the model to propagate shared, regional information exclusively into the region token. This avoids trivial region features and explicitly encourages region and channel tokens to encode *region-common* and *channel-specific* information (see Appendix B.2 for details on RCReg).

After RCReg and variance-covariance regularization on feature dimension (VCReg, Fig. 4(c) and Appendix B.4), RECTOR's overall SSL objective function becomes: $\mathcal{L} = \mathcal{L}_{\mathrm{NC}^2} + \mathcal{L}_{\mathrm{RCReg}} + \mathcal{L}_{\mathrm{VCReg}}$.

## 3 EXPERIMENTS

### 3.1 EXPERIMENTAL SETUP

**Datasets** We evaluate RECTOR on three publicly available EEG emotion recognition benchmarks: **SEED** (Zheng & Lu, 2015) (three-valence classification), **SEED-IV** (Zheng et al., 2018) (four-valence classification), and **DEAP** (Koelstra et al., 2011) (binary high/low valence & arousal classifications), and on two sEEG task engagement datasets, **MSIT** and **ECR** (Provenza et al., 2019) (binary rest/task classifications). For SEED, SEED-IV, and DEAP, we follow both subject-dependent (SD) and subject-independent (SI) evaluation protocols; for MSIT and ECR, we adhere to the SD paradigm (see Appendix D for details on the datasets).

**Baselines** We compare against three transformer self-supervised methods across all datasets: masked autoencoding in the input space (**MAE** (He et al., 2022)), joint-embedding predictive architecture in the representation space (**JEPA** (Assran et al., 2023)), and context-augmented modeling in both input and representation spaces (**CAE** (Chen et al., 2024)). In EEG emotion recognition,

we benchmark against leading *end-to-end* supervised models: **TSception** (Ding et al., 2022), **EEG Conformer** (Song et al., 2022), **LGGNet** (Ding et al., 2023), as well as self-supervised approaches: **LaBraM** (Jiang et al., 2024b), **REmoNet** (Jiang et al., 2024a), and **CBraMod** (Wang et al., 2025). We also compare against *non–end-to-end* methods, including the supervised **PGCN** (Jin et al., 2024), **MASA-TCN** (Ding et al., 2024), **EmT** (Ding et al., 2025) and the self-supervised **MMM** (Yi et al., 2023). In sEEG cognitive-state decoding, we compare against the supervised baseline **Seegnificant** (Mentzelopoulos et al., 2024) and leading self-supervised methods: **BrainBERT** (Wang et al., 2023a), **Brant** (Zhang et al., 2023), and **Du-IN** (Zheng et al., 2024). The self-supervised **PopT** (Chau et al.) is included in both EEG and sEEG experiments. For a comprehensive evaluation, we also include the EEG models (TSception, EEG Conformer, LGGNet) in sEEG experiments as additional references (see Appendix A for details on the comparison with baselines).

Table 1: Classification performance on SEED, SEED-IV, and DEAP (Weighted F1 Score (%)). SD: Subject Dependent; SI: Subject Independent. **BOLD**/UNDERLINE indicate the best/second-best results. Yellow (high) to blue (low). * denotes it significantly outperforms the second-best model.

| Model | SEED | | SEED-IV | | DEAP-Valence | | DEAP-Arousal | |
|---|---|---|---|---|---|---|---|---|
| | SD | SI | SD | SI | SD | SI | SD | SI |
| TSception | $74.51_{\pm11.82}$ | $49.20_{\pm12.09}$ | $54.29_{\pm14.01}$ | $35.47_{\pm08.13}$ | $62.00_{\pm14.47}$ | $61.53_{\pm15.34}$ | $62.94_{\pm16.23}$ | $61.31_{\pm16.04}$ |
| EEG Conformer | $66.64_{\pm12.64}$ | $51.00_{\pm11.86}$ | $55.62_{\pm14.28}$ | $37.47_{\pm08.38}$ | $64.25_{\pm14.28}$ | $62.84_{\pm15.38}$ | $64.47_{\pm16.09}$ | $62.03_{\pm16.52}$ |
| LGGNet | $68.56_{\pm11.36}$ | $50.73_{\pm10.98}$ | $50.67_{\pm13.58}$ | $36.80_{\pm08.17}$ | $64.50_{\pm13.46}$ | $61.06_{\pm15.74}$ | $62.50_{\pm15.57}$ | $62.22_{\pm15.84}$ |
| PGCN | $76.80_{\pm12.08}$ | $56.60_{\pm10.64}$ | $58.40_{\pm13.27}$ | $40.20_{\pm07.72}$ | $64.84_{\pm13.84}$ | $62.50_{\pm14.02}$ | $64.53_{\pm15.49}$ | $64.31_{\pm15.05}$ |
| MASA-TCN | $71.04_{\pm11.34}$ | $55.27_{\pm10.27}$ | $55.84_{\pm13.63}$ | $38.87_{\pm07.91}$ | $64.75_{\pm12.83}$ | $62.34_{\pm14.37}$ | $62.44_{\pm17.37}$ | $62.03_{\pm16.64}$ |
| EmT | $76.20_{\pm10.56}$ | $57.20_{\pm10.75}$ | $56.47_{\pm12.85}$ | $39.93_{\pm08.39}$ | $64.66_{\pm13.16}$ | $63.03_{\pm14.26}$ | $64.00_{\pm16.41}$ | $64.19_{\pm16.25}$ |
| MMM | $77.24_{\pm10.53}$ | $56.67_{\pm10.79}$ | $59.09_{\pm12.57}$ | $40.27_{\pm07.46}$ | $64.72_{\pm12.39}$ | $62.88_{\pm13.70}$ | $63.50_{\pm15.52}$ | $62.97_{\pm16.31}$ |
| LaBraM | $75.82_{\pm12.05}$ | $57.00_{\pm10.47}$ | $55.62_{\pm13.18}$ | $40.53_{\pm07.62}$ | $64.50_{\pm12.63}$ | $65.44_{\pm15.07}$ | $64.84_{\pm15.47}$ | |
| REmoNet | $\underline{80.67}_{\pm10.24}$ | $58.33_{\pm09.78}$ | $\underline{60.69}_{\pm13.04}$ | $41.93_{\pm07.01}$ | $\mathbf{66.81}_{\pm12.21}$ | $63.75_{\pm12.92}$ | $\underline{67.16}_{\pm14.28}$ | $65.50_{\pm15.09}$ |
| CBraMod | $78.18_{\pm10.19}$ | $\underline{58.67}_{\pm09.57}$ | $58.27_{\pm12.74}$ | $42.40_{\pm06.47}$ | $65.63_{\pm11.35}$ | $\underline{63.97}_{\pm12.18}$ | $65.88_{\pm14.08}$ | $\underline{65.59}_{\pm14.51}$ |
| PopT | $76.22_{\pm11.37}$ | $57.29_{\pm10.10}$ | $56.57_{\pm13.04}$ | $40.20_{\pm07.53}$ | $64.81_{\pm12.27}$ | $63.84_{\pm12.57}$ | $65.50_{\pm15.43}$ | $64.39_{\pm15.82}$ |
| MAE | $75.04_{\pm10.27}$ | $56.80_{\pm10.14}$ | $56.78_{\pm12.73}$ | $40.87_{\pm07.69}$ | $63.88_{\pm12.64}$ | $62.50_{\pm12.92}$ | $64.69_{\pm14.54}$ | $64.00_{\pm15.07}$ |
| JEPA | $77.49_{\pm09.64}$ | $58.47_{\pm10.32}$ | $58.02_{\pm11.74}$ | $41.87_{\pm07.37}$ | $65.56_{\pm11.97}$ | $62.91_{\pm12.44}$ | $66.03_{\pm13.92}$ | $65.34_{\pm14.62}$ |
| CAE | $77.80_{\pm10.16}$ | $57.87_{\pm09.78}$ | $59.09_{\pm12.31}$ | $\underline{42.53}_{\pm07.15}$ | $65.97_{\pm11.72}$ | $63.78_{\pm12.59}$ | $65.84_{\pm13.51}$ | $64.78_{\pm14.26}$ |
| **RECTOR** | $\mathbf{84.32}^{*}_{\pm08.80}$ | $\mathbf{60.58}^{*}_{\pm09.12}$ | $\mathbf{63.07}^{*}_{\pm11.16}$ | $\mathbf{44.79}^{*}_{\pm06.56}$ | $\mathbf{68.75}^{*}_{\pm10.10}$ | $\mathbf{66.22}^{*}_{\pm11.33}$ | $\mathbf{69.68}^{*}_{\pm12.52}$ | $\mathbf{68.01}^{*}_{\pm12.72}$ |

## 3.2 CLASSIFICATION RESULTS AND EFFICIENCY

We pre-trained RECTOR and all self-supervised baselines exclusively on each downstream dataset. Table 1 presents classification results on SEED, SEED-IV, and DEAP (see Appendix C.1 for balanced accuracy and Cohen's kappa). RECTOR establishes a new state-of-the-art across all settings, consistently outperforming the strongest supervised baselines (PGCN and EmT) and leading EEG-specific self-supervised methods (REmoNet, CBraMod). Table 2 shows AUROC results on MSIT and ECR. Here again, RECTOR outperforms all supervised and self-supervised methods. Moreover, REC-TOR exhibits the lowest standard deviations in both tasks, underscoring its superior robustness over other baselines.

We further benchmarked against MAE, JEPA, and CAE, three general masked modeling frameworks from computer vision, on our EEG and sEEG tasks. Although partially inspired by CAE's architecture, RECTOR consistently outperforms all three models.

This advantage stems from our end-to-end integration of: (1) RECTOR-SA for learning hierarchical region-channel-temporal interactions and pruning noisy connections (see Appendix C.5 for robustness to parcellation), (2) RECTOR-Mask for inducing a more challenging pretext task, (3) $NC^2$-MM objective for more predictive and consistent features, and (4) RCReg for robust and unmixed region-channel embeddings.

Across all benchmarks, RECTOR demonstrates superior training efficiency compared to SSL methods that use full spatio-

Table 2: Classification performance on MSIT and ECR (AUROC (%)). **BOLD**/UNDERLINE indicates the best/second-best results. Yellow (high) to blue (low). * denotes it significantly outperforms the second-best model.

| Model | MSIT | ECR |
|---|---|---|
| TSception | $83.71_{\pm07.73}$ | $83.24_{\pm07.82}$ |
| EEG Conformer | $82.65_{\pm06.36}$ | $83.29_{\pm06.79}$ |
| LGGNet | $82.94_{\pm06.48}$ | $82.18_{\pm07.79}$ |
| Seegnificant | $84.47_{\pm06.94}$ | $85.06_{\pm06.86}$ |
| BrainBERT | $85.00_{\pm06.32}$ | $85.76_{\pm06.26}$ |
| Brant | $86.47_{\pm06.28}$ | $86.88_{\pm05.92}$ |
| Du-IN | $\underline{87.29}_{\pm05.43}$ | $86.18_{\pm05.89}$ |
| PopT | $87.03_{\pm05.84}$ | $\underline{87.99}_{\pm05.58}$ |
| MAE | $86.24_{\pm05.82}$ | $85.47_{\pm06.24}$ |
| JEPA | $87.18_{\pm06.14}$ | $86.65_{\pm05.76}$ |
| CAE | $86.88_{\pm06.46}$ | $87.59_{\pm06.16}$ |
| **RECTOR** | $\mathbf{90.55}^{*}_{\pm05.17}$ | $\mathbf{90.79}^{*}_{\pm05.36}$ |

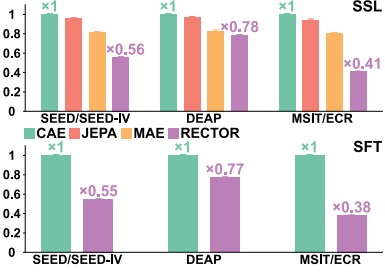

Figure 5: Training time comparison.

temporal self-attention (Fig. 5). In our analysis of training times for both SSL and supervised fine-tuning (SFT), RECTOR consistently outperformed MAE, JEPA, and a CAE baseline ($\times 1$). This efficiency gain is most evident on high-density neural recordings with many channels and regions (e.g., MSIT/ECR and SEED/SEED-IV), a direct result of our proposed RECTOR-SA architecture.

### 3.3 ABLATION STUDIES

We conducted ablation studies on each of RECTOR's core components to validate their contributions. Each ablation removes a specific module, targeting the challenges identified in Section 1. **Full SA** replaces RECTOR-SA with a dense spatio-temporal self-attention. **Random Mask** reverts RECTOR-Mask to a simpler pretext task using random masking instead of structured multi-view blocks. For **NC$^2$-MM**, we individually ablated its loss components: input reconstruction ($\mathcal{L}_{\text{input}}$), representation alignment ($\mathcal{L}_{\text{rep}}$), and contrastive learning ($\mathcal{L}_{\text{C}}$). Finally, we remove the **RCReg** module ($\mathcal{L}_{\text{RCReg}}$) to assess its impact on structuring the representation space.

Table 3: Ablation studies on MSIT/ECR (AUROC (%)). **BOLD** indicates the best result. Yellow (high) to blue (low). * denotes it is significantly worse than the full model.

| Model | MSIT | ECR |
|---|---|---|
| Full SA | $88.94^*_{\pm 06.12}$ | $88.82^*_{\pm 05.69}$ |
| Random Mask | $88.16^*_{\pm 06.48}$ | $87.92^*_{\pm 06.38}$ |
| $-\mathcal{L}_{\text{input}}$ | $88.49^*_{\pm 05.88}$ | $88.08^*_{\pm 05.78}$ |
| $-\mathcal{L}_{\text{rep}}$ | $88.42^*_{\pm 06.27}$ | $88.51^*_{\pm 06.08}$ |
| $-\mathcal{L}_{\text{C}}$ | $89.22^*_{\pm 06.12}$ | $89.01^*_{\pm 05.69}$ |
| $-\mathcal{L}_{\text{RCReg}}$ | $89.26^*_{\pm 06.01}$ | $89.40^*_{\pm 05.88}$ |
| **RECTOR** | $\mathbf{90.55}_{\pm 05.17}$ | $\mathbf{90.79}_{\pm 05.36}$ |

Tables 3 and 4 validate our design choices and demonstrate that each of RECTOR's core components is essential for its high performance in downstream cognitive tasks. Notably, replacing the structured RECTOR-Mask with random masking results in the most significant performance degradation. This highlights the critical role of our challenging pretext task in encouraging the model to learn robust region-channel-temporal representations. Similarly, substituting RECTOR-SA with a dense self-attention also leads to a major performance drop, validating our approach embedded with anatomical priors and dynamic gating. Furthermore, ablating the individual loss terms confirms the necessity of each component within NC$^2$-MM and RCReg, with $\mathcal{L}_{\text{input}}$ and $\mathcal{L}_{\text{rep}}$ being particularly evident (see Appendix C.3 for fine-grained ablations, e.g., gating mechanism).

Table 4: Ablation studies on SEED, SEED-IV, and DEAP (Weighted F1 Score (%)). SD: Subject Dependent; SI: Subject Independent. **BOLD** indicates the best result. Yellow (high) to blue (low). * denotes it is significantly worse than the full model.

| Model | SEED | | SEED-IV | | DEAP-Valence | | DEAP-Arousal | |
|---|---|---|---|---|---|---|---|---|
| | SD | SI | SD | SI | SD | SI | SD | SI |
| Full SA | $82.47^*_{\pm 09.63}$ | $59.60^*_{\pm 09.56}$ | $61.27^*_{\pm 12.40}$ | $43.53^*_{\pm 06.93}$ | $67.31^*_{\pm 11.08}$ | $65.16^*_{\pm 11.70}$ | $67.72^*_{\pm 13.19}$ | $66.44^*_{\pm 13.36}$ |
| Random Mask | $81.84^*_{\pm 09.84}$ | $59.47^*_{\pm 09.46}$ | $61.07^*_{\pm 12.57}$ | $42.80^*_{\pm 07.28}$ | $66.81^*_{\pm 11.47}$ | $64.19^*_{\pm 12.22}$ | $67.38^*_{\pm 13.58}$ | $66.31^*_{\pm 14.08}$ |
| $-\mathcal{L}_{\text{input}}$ | $82.31^*_{\pm 09.56}$ | $59.47^*_{\pm 09.78}$ | $60.92^*_{\pm 12.46}$ | $42.88^*_{\pm 07.47}$ | $67.00^*_{\pm 12.21}$ | $64.82^*_{\pm 13.12}$ | $67.69^*_{\pm 12.67}$ | $66.53^*_{\pm 14.01}$ |
| $-\mathcal{L}_{\text{rep}}$ | $81.92^*_{\pm 09.76}$ | $59.43^*_{\pm 09.68}$ | $60.98^*_{\pm 12.72}$ | $42.76^*_{\pm 07.16}$ | $66.84^*_{\pm 11.03}$ | $64.63^*_{\pm 12.17}$ | $67.78^*_{\pm 13.48}$ | $65.98^*_{\pm 14.09}$ |
| $-\mathcal{L}_{\text{C}}$ | $82.56^*_{\pm 10.02}$ | $59.71^*_{\pm 09.16}$ | $61.38^*_{\pm 11.84}$ | $43.78^*_{\pm 07.11}$ | $67.31^*_{\pm 12.04}$ | $65.28^*_{\pm 11.32}$ | $67.98^*_{\pm 12.58}$ | $67.02^*_{\pm 12.64}$ |
| $-\mathcal{L}_{\text{RCReg}}$ | $82.80^*_{\pm 09.70}$ | $59.67^*_{\pm 09.62}$ | $61.79^*_{\pm 11.22}$ | $43.98^*_{\pm 06.94}$ | $67.86^*_{\pm 11.08}$ | $65.48^*_{\pm 12.49}$ | $68.20^*_{\pm 12.77}$ | $66.92^*_{\pm 13.14}$ |
| **RECTOR** | $\mathbf{84.32}_{\pm 08.80}$ | $\mathbf{60.58}_{\pm 09.12}$ | $\mathbf{63.07}_{\pm 11.16}$ | $\mathbf{44.79}_{\pm 06.56}$ | $\mathbf{68.75}_{\pm 10.10}$ | $\mathbf{66.22}_{\pm 11.33}$ | $\mathbf{69.68}_{\pm 12.52}$ | $\mathbf{68.01}_{\pm 12.72}$ |

### 3.4 IMPACT OF EXPANDED PRE-TRAINING REGIMES

Tables 5 and 6 report downstream classification on MSIT, ECR, SEED, and SEED-IV under four progressively larger pre-training regimes (see Appendix C.2 for DEAP and more, and Appendix F.9 for implementation details), ranging from (1) training from scratch to pre-training on (2) each individual dataset, (3) all candidate datasets (**RECTOR+**, CHB-MIT (Shoeb, 2009) is added as a large candidate for EEG), (4) all candidate datasets using a larger RECTOR (**RECTOR-L+**). The re-

Table 5: Classification performance on MSIT/ECR under expanded pre-training (AUROC (%)). **BOLD**/UNDERLINE indicates the best/second-best result. Yellow (high) to blue (low). * denotes it significantly outperforms training from scratch.

| Model | Pre-Training Dataset | MSIT | ECR |
|---|---|---|---|
| | None | $85.47_{\pm 06.07}$ | $85.82_{\pm 06.44}$ |
| **RECTOR** | MSIT | $90.55^*_{\pm 05.17}$ | $88.27^*_{\pm 06.20}$ |
| | ECR | $87.62^*_{\pm 05.58}$ | $90.79^*_{\pm 05.36}$ |
| **RECTOR+** | MSIT+ECR | $90.85^*_{\pm 04.93}$ | $91.00^*_{\pm 05.48}$ |
| **RECTOR-L+** | MSIT+ECR | $\mathbf{91.11}^*_{\pm 04.98}$ | $\mathbf{91.14}^*_{\pm 05.33}$ |

sults demonstrate a clear scaling trend: performance consistently improves with both the amount of pre-training data and the model's size, with RECTOR-L+ consistently achieving the best results. The model also exhibits strong transferability, as pre-training on datasets excluding the target domain still substantially outperforms training from scratch. This effect is more evident for EEG tasks, while sEEG shows a greater benefit from in-domain data. Collectively, these findings confirm RECTOR's robust transferability and underscore the significant potential of large-scale pre-training.

Table 6: Classification performance on SEED/SEED-IV under expanded pre-training (Weighted F1 Score (%)). SD: Subject Dependent; SI: Subject Independent. **BOLD**/UNDERLINE indicates the best/second-best results. Yellow (high) to blue (low). * denotes it significantly outperforms training from scratch.

| Model | Pre-Training Dataset | SEED | | SEED-IV | |
|---|---|---|---|---|---|
| | | SD | SI | SD | SI |
| | None | $77.02_{\pm10.68}$ | $56.47_{\pm10.86}$ | $57.69_{\pm12.57}$ | $40.20_{\pm07.97}$ |
| **RECTOR** | SEED | $84.32^*_{\pm08.80}$ | $60.58^*_{\pm09.12}$ | $62.92^*_{\pm11.62}$ | $44.33^*_{\pm06.80}$ |
| | SEED-IV | $83.52^*_{\pm09.19}$ | $60.86^*_{\pm09.23}$ | $63.07^*_{\pm11.16}$ | $44.79^*_{\pm06.56}$ |
| | SEED+SEED-IV | $84.90^*_{\pm08.83}$ | $61.16^*_{\pm09.77}$ | $63.85^*_{\pm11.24}$ | $45.19^*_{\pm06.10}$ |
| **RECTOR+** | SEED+SEED-IV+DEAP+CHB-MIT | $84.87^*_{\pm08.70}$ | $\underline{61.67}^*_{\pm09.13}$ | $63.92^*_{\pm11.46}$ | $\underline{45.37}^*_{\pm06.03}$ |
| **RECTOR-L+** | SEED+SEED-IV+DEAP+CHB-MIT | $\mathbf{85.50}^*_{\pm08.82}$ | $\mathbf{61.69}^*_{\pm09.01}$ | $\mathbf{64.21}^*_{\pm10.93}$ | $\mathbf{45.76}^*_{\pm06.14}$ |

## 3.5 INTERPRETABILITY ANALYSIS

Fig. 6 illustrates class activation maps (Selvaraju et al., 2017) at both channel and region levels, together with the top-10 cross-region attention pairs for SEED and SEED-IV. These maps reveal clear emotion-specific spatial signatures: in both datasets, the model learns distinct and opposing patterns of frontal lobe engagement for positive and negative affective states. These engagement patterns align closely with motivational-direction models of frontal asymmetry (Harmon-Jones, 2003; 2004; Light et al., 2009; Parro et al., 2018), and reflect multivariate EEG signatures that a deep classifier can exploit. At region-level, Fig. 6 (right) underscores strong fronto-temporal and occipito-temporal interactions during emotional processing (Sun et al., 2023) (see Appendix C.8 for details on interpretability analysis). These results demonstrate that RECTOR robustly captures neurophysiologically plausible, region–channel dynamics underlying affective and cognitive conditions.

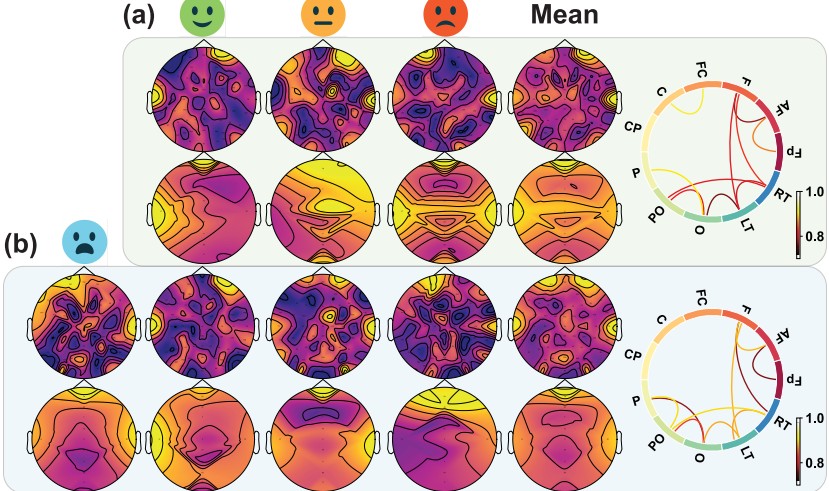

Figure 6: Class activation maps (left) and top-10 cross-region attention scores (right) on (a) SEED and (b) SEED-IV datasets. For each dataset, the top row shows channel-specific activation patterns for different emotions, and the bottom row shows region-common activations ("Mean" denoting the average activation across emotions), while the right panels list the crucial cross-region attention pairs.

## 4 CONCLUSION

We introduced RECTOR, the first end-to-end, self-supervised brain region learning framework that unifies region, channel, and temporal modeling for cognitive representation learning within a single, anatomically-guided architecture. Our experiments demonstrate that RECTOR's hierarchical self-attention and synergistic learning objective are highly effective, establishing a new state-of-the-art in both EEG emotion recognition and sEEG task-engagement classification, and it achieves superior computational efficiency in spatio-temporal self-attention. The success of expanded pre-training highlights RECTOR's strong potential for large-scale applications. Moreover, our visualizations confirmed that RECTOR learns neurophysiologically plausible representations, capturing distinct region-common and channel-specific neural patterns. These findings establish RECTOR as a powerful framework for learning dynamic, brain network-level neural representations, thereby paving the way for more advanced neurocognitive diagnostics and adaptive, personalized neuro-interventions.

## 5 ETHICAL STATEMENTS

The data collection and experiments conducted in our work have been approved by the Institutional Review Board (IRB) and passed ethical review.

## 6 LLM USAGE STATEMENTS

Large Language Models (LLMs) were used by the authors strictly as a general-purpose writing assist tool. The role of the LLM was limited to improving clarity, grammar, and style in non-technical sections. The LLM was never used for research ideation, generating core arguments, conducting experiments, or generating data.

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

# A COMPARISONS WITH RELATED WORKS

Learning effective neural representations underlying cognitive conditions from EEG and sEEG signals is a central challenge in cognitive neuroscience. We review prior work in four key areas: CNN-based and hybrid models, graph neural networks, transformers for spatio-temporal modeling, and self-supervised, region-aware learning frameworks.

## A.1 CNN AND HYBRID ARCHITECTURES

Convolutional Neural Networks (CNNs) are widely used for their ability to extract local spatio-temporal features from EEG signals. Models like TSception (Ding et al., 2022) and MASA-TCN (Ding et al., 2024) employ multi-scale temporal convolutions to capture neural dynamics across different frequencies and durations. Hybrid architectures such as EEG Conformer (Song et al., 2022) and REmoNet (Jiang et al., 2024a) combine CNNs with transformers or RNNs to pair local feature extraction with long-range dependency modeling. While effective, these methods often rely on fixed convolutional filters for spatial processing and can struggle to flexibly model the brain's dynamic, long-range functional interactions. In contrast, RECTOR uses a fully attention-based approach (RECTOR-SA) that learns these relationships adaptively from data, guided by a flexible, anatomically-informed structure rather than rigid filters.

## A.2 GRAPH NEURAL NETWORKS FOR SPATIAL MODELING

Graph Neural Networks (GNNs) have been proposed to explicitly model the brain's network topology. LGGNet (Ding et al., 2023) and PGCN (Jin et al., 2024) represent electrodes as nodes in a graph, using graph convolutions to learn spatial features that respect neurophysiological priors. These models often use a hierarchical or pyramidal structure to aggregate information from local electrodes to global brain regions. GNN-based approaches typically rely on a pre-defined, static graph of electrode connectivity. RECTOR differs by learning both static and dynamic relationships. Its Functional Attention dynamically models input-dependent interactions between regions and channels, offering a more flexible and adaptive framework.

## A.3 TRANSFORMERS FOR SPATIO-TEMPORAL MODELING

Transformers are the dominant architecture for sequence modeling, but adapting them for the joint spatio-temporal nature of EEG is non-trivial. Prior works fall into several categories:

1. Decoupled Attention: Many models handle spatial and temporal dimensions separately to maintain efficiency. CBraMod (Wang et al., 2025) uses criss-cross attention to model channel-wise and time-wise dependencies in parallel. Others like Brant and its successor Brant-2 (Yuan et al., 2024b), BrainWave (Yuan et al., 2024a), Seegnificant (Mentzelopoulos et al., 2024), and EmT (Ding et al., 2025) apply sequential transformer blocks. The primary limitation of these methods is their failure to capture joint, simultaneous spatio-temporal interactions.

2. Full Spatio-Temporal Attention: Foundation models like LaBraM (Jiang et al., 2024b) apply a dense, full self-attention across all channel-time tokens. While this enables the modeling of all possible interactions, it is computationally prohibitive and risks amplifying noise.

3. Generalist Architectures: Some works adapt general-purpose time-series transformers like Medformer for medical data. While powerful, these generic models lack the specific inductive biases for neurophysiological signals.

RECTOR is designed to resolve these issues. Its hierarchical RECTOR-SA mechanism models joint region-channel-temporal interactions in a sparse, block-wise manner, providing an efficient and effective middle ground between decoupled and full attention. Unlike generalist models like Medformer, RECTOR's design is explicitly neuro-anatomically grounded, incorporating a region-channel hierarchy and anatomical priors that are absent in generic architectures.

## A.4 Self-Supervised and Region-Aware Learning

SSL is critical for leveraging sparsely labeled EEG/sEEG data. While many works use masked modeling, some have begun to incorporate regional brain structure. MMM (Yi et al., 2023) introduces region-wise tokens, and Du-IN (Zheng et al., 2024) forms patch embeddings by fusing channels within specific regions. These approaches tackle important, often orthogonal, challenges but do not offer a unified solution. PopT (Chau et al.) and Seegnificent (Mentzelopoulos et al., 2024) oversimplify brain topology by performing population-level spatial encoding, treating all channels as a single global region and thus ignoring the brain's established functional network architecture. While EEGPT (Wang et al., 2024) introduces a powerful dual objective with representation alignment, it lacks the explicit region-channel structure and dedicated regularization (RCReg) that RECTOR uses to learn a non-redundant, hierarchical representation. Similarly, Du-IN fuses channels, discarding the fine-grained channel-specific dynamics that RECTOR preserves and models. RECTOR is unique in its synergistic combination of a structured mask (RECTOR-Mask), a hybrid learning objective (NC$^2$-MM), and a dedicated regularizer (RCReg) for learning an explicit hierarchy of region, channel, and temporal dynamics.

# B  Methodology

## B.1  Computational and Complexity

Let $R$ be the number of regions, $T$ the number of time segments, and $d$ the embedding dimension. For simplicity, we assume a constant number of channels $P$ per region. The full spatio-temporal SA has a complexity of: (1) $\mathcal{O}(3RPTd^2)$ for $\mathbf{QKV}$ projections, (2) $\mathcal{O}(2R^2P^2T^2d)$ for $\mathbf{Q}$-$\mathbf{K}$ multiplication and $\mathbf{A}$-$\mathbf{V}$ multiplication, where $\mathbf{A} = \text{softmax}(\frac{\mathbf{QK}^\top}{\sqrt{d}} \odot \mathbf{M}_{\text{Attn}})$, (3) $\mathcal{O}(R^2P^2T^2)$ for softmax($\cdot$), and (4) $\mathcal{O}(RPTd^2)$ for output projection. Therefore, the total complexity of full self-attention is $\mathcal{O}(2R^2P^2T^2d + 4RPTd^2)$ for $\mathbf{QKV}$ multiplications and projections, and $\mathcal{O}(R^2P^2T^2)$ for softmax($\cdot$).

For RECTOR-SA, we can split it into:

1. Local functional attention: We split the $RP$ channels into $R$ regions and compute self-attention within each region over the full temporal extent $T$. The complexity of its $\mathbf{QKV}$ multiplications is $\mathcal{O}(2R(PT)^2d)$. We compute softmax($\cdot$) on a matrix of size $P \times (PT+1)$ for $RT$ times. Therefore, the complexity of softmax($\cdot$) is $\mathcal{O}(RTP(PT+1)) \approx \mathcal{O}(RP^2T^2)$.

2. Global functional attention: We treat region tokens plus conditioning tokens as a sequence of length $RT + K$. The complexity of its $\mathbf{QKV}$ multiplications is $\mathcal{O}(2(RT+K)^2d) \approx \mathcal{O}(2R^2T^2d)$. We compute softmax($\cdot$) on a matrix of size $(RT+k) \times (RPT+k)$ attention block. Therefore, the complexity of softmax($\cdot$) is $\mathcal{O}((RT+K)(RPT+k)) \approx \mathcal{O}(R^2PT^2)$.

3. Anatomical attention: We allow each channel token to only attend to its corresponding region token at the same time index and vice versa. The complexity of its $\mathbf{QKV}$ multiplications is $\mathcal{O}(2RPTd)$, which is a lower order term and can be ignored.

4. Total projection: The complexity of all projections is $\mathcal{O}(4(RPT + RT + k)d^2) \approx \mathcal{O}(4RPTd^2 + 4RTd^2)$.

Therefore, the complexity of $\mathbf{QKV}$ multiplications and projections in RECTOR-SA is $\mathcal{O}(2RP^2T^2d + 2R^2T^2d + 4RPTd^2 + 4RTd^2)$, and the complexity for softmax($\cdot$) is $\mathcal{O}(RP^2T^2 + R^2PT^2)$. For $\mathbf{QKV}$ operations, RECTOR-SA is more efficient when $d < T(P^2(R-1)-R)/2$. For the softmax($\cdot$) computation, RECTOR-SA is advantageous when $P > R/(R-1)$.

Additionally, unlike approaches that require separate forward passes for each prediction target, RECTOR computes all five masked-region predictions in a single forward pass.

## B.2  Region-Channel Regularization (RCReg)

To understand how RCReg enforces a structured representation space, we can break down its covariance penalty. Given region tokens $\mathbf{Z}^{\text{R}} \in \mathbb{R}^{R \times d}$ and channel tokens $\mathbf{Z}^{\text{C}} \in \mathbb{R}^{C \times d}$, the covariance

penalty in Equation (10) can be formulated as:

$$\mathbf{Z} = \left[ \mathbf{Z}^{\mathrm{C}}; \mathbf{Z}^{\mathrm{R}} \right] \qquad \tilde{\mathbf{Z}} = \mathbf{Z} - \bar{\mathbf{Z}}\mathbf{1}^{\top} \qquad \mathbf{C} = \frac{1}{d-1}\tilde{\mathbf{Z}}\tilde{\mathbf{Z}}^{\top}$$

$$\mathcal{L}_{\mathrm{cov}}^{\mathrm{RC}} = \frac{1}{C+R}\sum_{i \neq j}[\mathbf{C}_{i,j}]^2 = \frac{1}{C+R}\|\mathbf{C} \odot \mathbf{M}\|_{\mathrm{F}}^2 \tag{11}$$

where $\mathbf{M} = \mathbf{1}\mathbf{1}^{\top} - \mathbf{I}_{(C+R)} \in \{0,1\}^{(C+R)\times(C+R)}$ is a binary mask on the diagonals of $\mathbf{C}$. Starting from:

$$\frac{d\mathcal{L}_{\mathrm{cov}}^{\mathrm{RC}}}{d\mathbf{C}} = \frac{2}{C+R}(\mathbf{C} \odot \mathbf{M}) = \mathbf{U} \qquad d\mathbf{C} = \frac{1}{d-1}((d\tilde{\mathbf{Z}})\tilde{\mathbf{Z}}^{\top} + \tilde{\mathbf{Z}}(d\tilde{\mathbf{Z}}^{\top})) \tag{12}$$

The chain rule gives:

$$\begin{aligned} d\mathcal{L}_{\mathrm{cov}}^{\mathrm{RC}} &= \langle \mathbf{U}, d\mathbf{C} \rangle \\ &= \frac{1}{d-1}\langle \mathbf{U}, (d\tilde{\mathbf{Z}})\tilde{\mathbf{Z}}^{\top} + \tilde{\mathbf{Z}}(d\tilde{\mathbf{Z}}^{\top}) \rangle \\ &= \frac{2}{d-1}\langle \mathbf{U}\tilde{\mathbf{Z}}, d\tilde{\mathbf{Z}} \rangle \end{aligned} \tag{13}$$

So each gradient step updates the centered tokens $\tilde{\mathbf{Z}}$ by subtracting the covariance penalty gradient:

$$\nabla_{\tilde{\mathbf{Z}}}\mathcal{L}_{\mathrm{cov}}^{\mathrm{RC}} = \frac{2}{d-1}\mathbf{U}\tilde{\mathbf{Z}} = \frac{4}{(C+R)(d-1)}(\mathbf{C} \odot \mathbf{M})\tilde{\mathbf{Z}} \tag{14}$$

We partition the full covariance matrix into the four sub-blocks: $\mathbf{C}^{\mathrm{CC}} \in \mathbb{R}^{C \times C}, \mathbf{C}^{\mathrm{RR}} \in \mathbb{R}^{R \times R}, \mathbf{C}^{\mathrm{RC}} \in \mathbb{R}^{R \times C}, \mathbf{C}^{\mathrm{CR}} = \mathbf{C}^{\mathrm{RC}^{\top}}$, that captures inter-channel, inter-region, and region-channel covariances. This yields the iterative block-wise updates with learning rate $\alpha$:

$$\begin{aligned} \tilde{\mathbf{Z}}_{(n+1)}^{\mathrm{R}} &= \tilde{\mathbf{Z}}_{(n)}^{\mathrm{R}} - \alpha\frac{4}{(C+R)(d-1)}\left(\mathbf{C}^{\mathrm{RR}}\tilde{\mathbf{Z}}_{(n)}^{\mathrm{R}} + \mathbf{C}^{\mathrm{RC}}\tilde{\mathbf{Z}}_{(n)}^{\mathrm{C}}\right) \\ \tilde{\mathbf{Z}}_{(n+1)}^{\mathrm{C}} &= \tilde{\mathbf{Z}}_{(n)}^{\mathrm{C}} - \alpha\frac{4}{(C+R)(d-1)}\left(\mathbf{C}^{\mathrm{CR}}\tilde{\mathbf{Z}}_{(n)}^{\mathrm{R}} + \mathbf{C}^{\mathrm{CC}}\tilde{\mathbf{Z}}_{(n)}^{\mathrm{C}}\right) \end{aligned} \tag{15}$$

Crucially, the cross-covariance term $\mathbf{C}^{\mathrm{RC}}\tilde{\mathbf{Z}}^{\mathrm{C}}$ subtracts from each region token the component lying in the span of channel tokens–and similarly $\mathbf{C}^{\mathrm{CR}}\tilde{\mathbf{Z}}^{\mathrm{R}}$, $\mathbf{C}^{\mathrm{CC}}\tilde{\mathbf{Z}}^{\mathrm{C}}$, and $\mathbf{C}^{\mathrm{RR}}\tilde{\mathbf{Z}}^{\mathrm{R}}$ for subtracting channel-region, inter-channel, and inter-region components. Repeating these updates drives the elements of $\mathbf{C}^{\mathrm{RR}}, \mathbf{C}^{\mathrm{CC}}, \mathbf{C}^{\mathrm{RC}}$ toward zero, thereby *decorrelating/orthogonalizing* the region and channel subspaces. This covariance penalty yields non-collapsed embeddings that better capture complementary spatial patterns for downstream EEG/sEEG tasks.

It should be noted that when using RCReg, we should treat $\lambda_{\mathrm{RCVar}}$ and separate $\lambda_{\mathrm{RCCov}}$ on $\mathbf{C}^{\mathrm{RR}}$, $\mathbf{C}^{\mathrm{RC}}$, $\mathbf{C}^{\mathrm{CR}}$, and $\mathbf{C}^{\mathrm{CC}}$ as hyperparameters during self-supervised learning.

### B.3 INPUT RECONSTRUCTION LOSS MITIGATES REPRESENTATION COLLAPSE

Here is a detailed theoretical analysis grounded in the mutual information maximization to justify why our NC$^2$-MM loss mitigates representation collapse and yields superior representations.

We want the learned representation $\mathbf{Z}$ to capture the maximum amount of information about the input $\mathbf{X}$. That is, we want to maximize the mutual information $I(\mathbf{X}; \mathbf{Z})$, where:

$$I(\mathbf{X}; \mathbf{Z}) = H(\mathbf{X}) - H(\mathbf{X}|\mathbf{Z}) \tag{16}$$

Since the entropy of the dataset $H(\mathbf{X})$ is constant, maximizing $I(\mathbf{X}; \mathbf{Z})$ is equivalent to minimizing the conditional entropy $H(\mathbf{X}|\mathbf{Z})$. If we model the conditional distribution $P(\mathbf{X}|\mathbf{Z})$ as a Gaussian distribution $\mathcal{N}(\hat{\mathbf{X}}(\mathbf{Z}), \sigma^2\mathbf{I})$, then minimizing the negative log-likelihood corresponds exactly to minimizing the Mean Squared Error (MSE):

$$\mathcal{L}_{\mathrm{input}} = ||\mathbf{X} - \hat{\mathbf{X}}(\mathbf{Z})||^2 \propto -\log P(\mathbf{X}|\mathbf{Z}) \tag{17}$$

The expectation of this negative log-likelihood over the data distribution is the conditional entropy (plus a constant):

$$\mathbb{E}_{\mathbf{X},\mathbf{Z}}[-\log P(\mathbf{X}|\mathbf{Z})] = H(\mathbf{X}|\mathbf{Z}) + C \tag{18}$$

Therefore, minimizing $\mathcal{L}_{\text{input}}$ over the data distribution minimizes the conditional entropy $H(\mathbf{X}|\mathbf{Z})$.

A collapsed representation where $\mathbf{Z} = c$ (constant) has $I(\mathbf{X}, \mathbf{Z} = c) = 0$ and maximizes conditional entropy : $H(\mathbf{X}|\mathbf{Z} = c) = H(\mathbf{X})$. By minimizing $\mathcal{L}_{\text{input}}$, we force $H(\mathbf{X}|\mathbf{Z})$ to be lower than $H(\mathbf{X})$, which guarantees that $I(\mathbf{X}; \mathbf{Z}) > 0$. It shows that the global minimum of $\mathcal{L}_{\text{input}}$ cannot be a collapsed state. The reconstruction loss forces $\mathbf{Z}$ to retain information about $\mathbf{X}$.

The effect of $\mathcal{L}_{\text{input}}$ can also be understood via variance analysis. Consider the loss function $\mathcal{L}_{\text{input}} = \mathbb{E}[||\mathbf{X} - \hat{X}(\mathbf{Z})||^2]$. If we assume the encoder collapses such that $\mathbf{Z} = c$ for all $\mathbf{X}$. The decoder $\hat{\mathbf{X}}(\mathbf{Z})$ then outputs a constant vector $\mu = \hat{\mathbf{X}}(c)$. The optimal constant vector $\mu$ that minimizes MSE is the mean of the dataset: $\mu = \mathbb{E}[\mathbf{X}]$. In this collapsed state, the loss becomes the variance of the dataset:

$$\mathcal{L}_{\text{input}}^{\text{collapse}} = \mathbb{E}[||\mathbf{X} - \mathbb{E}[\mathbf{X}]||^2] = \text{Var}(\mathbf{X}) \tag{19}$$

For any non-trivial dataset, $\text{Var}(\mathbf{X}) > 0$. A non-collapsed encoder that retains even a single bit of information about $\mathbf{X}$ can achieve a lower reconstruction error than $\text{Var}(\mathbf{X})$. The collapsed state $\mathbf{Z} = c$ is never the global minimum of $\mathcal{L}_{\text{input}}$. The optimization landscape of $\mathcal{L}_{\text{input}}$ inherently drives the model away from collapse.

### B.4 VARIANCE-COVARIANCE REGULARIZATION (VCREG)

To prevent the self-supervised model from collapsing to a trivial solution while simultaneously encouraging it to learn non-redundant features, we employ Variance-Covariance Regularization (**VCReg**). Let $B$ be the batch size, $N$ the number of tokens, and $d$ the embedding size. Given a batch of hidden representations $\mathbf{Z} \in \mathbb{R}^{B \times N \times d}$, extracted by the context encoder and followed by a linear projection, we can define VCReg as a combination of a variance hinge loss and a covariance loss: $\mathcal{L}_{\text{VCReg}} = \lambda_{\text{Var}}\mathcal{L}_{\text{Var}}(\mathbf{Z}) + \lambda_{\text{Cov}}\mathcal{L}_{\text{Cov}}(\mathbf{Z})$ , where

$$\mathcal{L}_{\text{Var}}(\mathbf{Z}) = \frac{1}{d}\sum_{k=1}^{d}\max\left(0, \gamma - \sqrt{\frac{1}{BN-1}\sum_{b=1}^{B}\sum_{n=1}^{N}\left(\mathbf{Z}_{b,n}^{(k)} - \bar{\mathbf{Z}}^{(k)}\right)^2 + \epsilon}\right)$$

$$\mathcal{L}_{\text{Cov}}(\mathbf{Z}) = \frac{1}{d}\sum_{i \neq j}\left[\frac{1}{BN-1}\sum_{b=1}^{B}\sum_{n=1}^{N}(\mathbf{Z}_{b,n} - \bar{\mathbf{Z}})(\mathbf{Z}_{b,n} - \bar{\mathbf{Z}})^T\right]_{i,j}^{2} \tag{20}$$

It should be noted that when using VCReg, we should treat $\lambda_{\text{Var}}$ and $\lambda_{\text{Cov}}$ as hyperparameters during self-supervised learning.

### B.5 REGION-CHANNEL-TEMPORAL INTERMEDIATE LAYER TUNING (RECTOR-LT)

We propose **RECTOR-LT**: Region-Channel-Temporal Intermediate Layer Tuning (Figure 7), which (1) unlocks and aggregates high-level features from multiple intermediate layers, (2) routes these features into three lightweight, cross-attention domain heads (region, channel, temporal), each specialized in one facet of EEG/sEEG structure, and (3) adaptively selects the most informative features. By disentangling and specializing intermediate features, RECTOR-LT yields rich representations for downstream tasks.

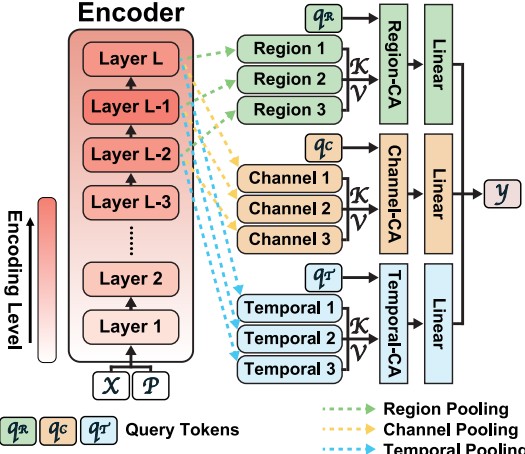

Figure 7: **RECTOR-LT**. Intermediate-layer tokens are separated into region, channel, and temporal streams, then dynamically aggregated via dedicated cross-attention (CA) heads for downstream tasks.

Given input patches $\mathbf{X}$ and a pre-trained encoder with $L$ layers, we extract features from the last $L_{\mathbf{I}}$ intermediate hidden layers (Figure 7). For each layer $l \in \{L - L_{\mathbf{I}} + 1, \cdots, L\}$, we reshape its output $\mathbf{Z}_{(l)} \in \mathbb{R}^{((C+R)T+K) \times d}$ into three domain tensors: $\mathbf{Z}_{(l)}^{\mathrm{R,T}} \in \mathbb{R}^{R \times T \times d}$, $\mathbf{Z}_{(l)}^{\mathrm{C,T}} \in \mathbb{R}^{C \times T \times d}$, $\mathbf{Z}_{(l)}^{\mathrm{RC,T}} \in \mathbb{R}^{(C+R) \times T \times d}$. We then collapse the temporal/spatial dimension via average-pooling:

$$\mathbf{Z}_{(l)}^{\mathrm{R}} = \frac{1}{T} \sum_{i=1}^{T} \left[\mathbf{Z}_{(l)}^{\mathrm{R,T}}\right]_{:,i,:}, \quad \mathbf{Z}_{(l)}^{\mathrm{C}} = \frac{1}{T} \sum_{i=1}^{T} \left[\mathbf{Z}_{(l)}^{\mathrm{C,T}}\right]_{:,i,:} \quad \mathbf{Z}_{(l)}^{\mathrm{T}} = \frac{1}{C+R} \sum_{i=1}^{C+R} \left[\mathbf{Z}_{(l)}^{\mathrm{RC,T}}\right]_{i,:,:} \quad (21)$$

To specialize downstream representations, we introduce three learnable query tokens $\mathbf{q}^{(u)} \in \mathbb{R}^d$ for $(u) \in \{\mathrm{R, C, T}\}$. Each domain head applies scaled dot-product cross-attention (CA) over the stacked intermediate domain tokens $\left[\mathbf{Z}_{(L-L_{\mathbf{I}}+1)}^{(u)}; \ldots; \mathbf{Z}_{(L)}^{(u)}\right]$ for learning region, channel, and temporal representations separately:

$$\mathbf{h}^{(u)} = \mathrm{CA}^{(u)} \left(\mathbf{q}^{(u)}, \left[\mathbf{Z}_{(L-L_{\mathbf{I}}+1)}^{(u)}; \ldots; \mathbf{Z}_{(L)}^{(u)}\right], \left[\mathbf{Z}_{(L-L_{\mathbf{I}}+1)}^{(u)}; \ldots; \mathbf{Z}_{(L)}^{(u)}\right]\right) \in \mathbb{R}^d \quad (22)$$

Finally, we concatenate $\mathbf{h}^{\mathrm{R}}$, $\mathbf{h}^{\mathrm{C}}$, $\mathbf{h}^{\mathrm{T}}$ and pass them through an output head $\mathbf{W}_{\mathrm{out}}$:

$$\hat{y} = \mathrm{softmax}(\left[\mathbf{h}^{\mathrm{R}}; \mathbf{h}^{\mathrm{C}}; \mathbf{h}^{\mathrm{T}}\right] \mathbf{W}_{\mathrm{out}}) \quad (23)$$

This RECTOR-LT strategy (1) preserves the rich, high-level features of deep encoder layers, (2) selectively adapts the most informative intermediate representations, and (3) disentangles region, channel, and temporal information into dedicated pathways, improving downstream performance, especially in low-label regimes.

### B.6 TOP-$p$ GATING

Given an input vector $\mathbf{x} \in \mathbb{R}^n$ with decreasing order $\{\mathrm{x}_1, \ldots, \mathrm{x}_n\}$, we first compute its softmax-normalized probabilities $\mathbf{p} = \mathrm{softmax}(\mathbf{x}) = \{\mathrm{p}_1, \ldots, \mathrm{p}_n\}$. For a fixed threshold $p \in (0, 1]$, we define the top-$p$ gating index $\tau$ as the smallest integer satisfying:

$$\sum_{i=1}^{\tau} \mathrm{p}_i \geq p \quad (24)$$

The resulting **top-$p$ gating mask** (Figure 3) top-$p(\mathbf{x}) = \mathbf{m} \in \{0, 1\}^n$ is then:

$$m_i = \begin{cases} 1, & i \leq \tau \\ 0, & i > \tau \end{cases} \quad (25)$$

and the gated vector $\mathbf{x}_{\mathrm{g}}$ is obtained by elementwise multiplication:

$$\mathbf{x}_{\mathrm{g}} = \mathbf{x} \odot \mathrm{top}\text{-}p(\mathbf{x}) \quad (26)$$

Integrated into RECTOR-SA (see Section 2.3), this top-$p$ gating mechanism enables the model to dynamically focus on the most informative region-channel-temporal attention entries–automatically adjusting sparsity to the data's distribution–without the need to hand-tune a fixed amount of entries in top-$k$ gating (Shazeer et al., 2017), yielding both great adaptivity and computational efficiency in RECTOR-SA.

## B.7 BLOCK MASKING IN RECTOR-MASK

To generate both context and target block masks in Section 2.4, we first partition the input patch matrix $\mathbf{X}^C \in \mathbb{R}^{C \times T}$ along the channel axis into contiguous regions (Figure 3). Then, for any sub-matrix of interest $\mathbf{Y} = \mathbf{X}^C_{c_1:c_2, t_1:t_2}$, we define a generic *block-masking operator*:

$$\mathbf{M} = \mathrm{BlockMask}(\mathbf{Y}, \rho, \eta) \tag{27}$$

parameterized by two interval-ratios $\rho = (\rho_1, \rho_2)$ (along channels) and $\eta = (\eta_1, \eta_2)$ (along time). Concretely:

(1) We sample the block size by drawing the channel-width $w_\rho$ and time-length $w_\eta$ as integers from discrete uniform distributions $\mathrm{U}\{a, b\}$ in an interval $[a, b]$:

$$\begin{aligned} w_\rho &\sim \mathrm{U}\left\{\lceil \rho_1(c_2 - c_1 + 1) \rceil, \lfloor \rho_2(c_2 - c_1 + 1) \rfloor\right\} \\ w_\eta &\sim \mathrm{U}\left\{\lceil \eta_1(t_2 - t_1 + 1) \rceil, \lfloor \eta_2(t_2 - t_1 + 1) \rfloor\right\} \end{aligned} \tag{28}$$

(2) We sample the block location by choosing a random starting channel $s_\rho$ and time index $s_\eta$ such that the block fits within the $(c_2 - c_1 + 1) \times (t_2 - t_1 + 1)$ grid, each drawn uniformly:

$$s_\rho \sim \mathrm{U}\{c_1, c_2 - w_\rho + 1\} \qquad s_\eta \sim \mathrm{U}\{t_1, t_2 - w_\eta + 1\} \tag{29}$$

(3) We form the block mask as:

$$\mathbf{M}_{i,j} = \begin{cases} 1, & \text{if } i \in [s_\rho, s_\rho + w_\rho) \ \wedge \ j \in [s_\eta, s_\eta + w_\eta) \\ 0, & \text{otherwise} \end{cases} \tag{30}$$

In this way, the initial context-block mask and its corresponding visible context are:

$$\mathbf{M}'_{\mathbf{c}} = \mathrm{BlockMask}(\mathbf{X}^C, \rho_{\mathbf{c}}, \eta_{\mathbf{c}}) \qquad \mathbf{X}'_{\mathbf{c}} = \mathbf{X}^C \odot \mathbf{M}'_{\mathbf{c}} \tag{31}$$

Here $(\rho_{\mathbf{c}}, \eta_{\mathbf{c}})$ denote the context block's range ratios, and $\odot$ indicates element-wise multiplication. We then sample each target-block mask $\mathbf{M}_{\mathbf{t}(m)}$ in domain $m \in \{\times \mathrm{r}, \times \mathrm{c}, \times \mathrm{t}, \times \mathrm{ct}, \times \mathrm{rt}\}$ via:

$$\mathbf{M}_{\mathbf{t}(m)} = \mathrm{BlockMask}\left([\mathbf{Y}_{\mathbf{t}(m)} \| \mathbf{X}'_{\mathbf{c}}], \rho_{\mathbf{t}(m)}, \eta_{\mathbf{t}(m)}\right) \tag{32}$$

Here, $\mathbf{Y}_{\mathbf{t}(m)}$ and $(\rho_{\mathbf{t}(m)}, \eta_{\mathbf{t}(m)})$ denote the patches in the area of target domain $m$ and their corresponding range ratios; $\|$ indicates concatenation along the channel axis (for $\times \mathrm{c}$) or the time axis (for $\times \mathrm{t}$), with no concatenation used in the other target domains. Finally, to ensure the context block does not overlap any target blocks, we update the context-block mask as:

$$[\mathbf{M}_{\mathbf{c}}]_{i,j} = \begin{cases} 1, & \text{if } [\mathbf{M}'_{\mathbf{c}}]_{i,j} = 1 \ \wedge \ [\mathbf{M}_{\mathbf{t}(\times \mathrm{c})}]_{i,j} = 0 \ \wedge \ [\mathbf{M}_{\mathbf{t}(\times \mathrm{t})}]_{i,j} = 0 \\ 0, & \text{otherwise} \end{cases} \tag{33}$$

Therefore, RECTOR-Mask yields one context block and five mutually disjoint target blocks per sample, providing *diverse intra-sample views* for self-supervised learning.

## B.8 RECTOR-TRANSFORMER BLOCK

The **RECTOR-Tramsformer block** serves as the core building block in our encoder, predictor, and decoder. It comprises two key components: (1) Multi-head RECTOR-SA (see Section 2.3), which performs both structured and dynamic token mixing across spatial (region, channel) and temporal dimensions (Figure 3); (2) SwiGLU-style feed-forward network (Shazeer, 2020), which further enhances feature learning.

Concretely, given the post-attention tokens $\mathbf{X}$, we apply:

$$\begin{aligned} \mathbf{X}_{\mathrm{FFN}} &= \mathrm{FFN}_{\mathrm{SwiGLU}}(\mathbf{X}) \\ &= (\mathrm{Swish}_1(\mathbf{X}\mathbf{W}_1) \odot \mathbf{X}\mathbf{V})\mathbf{W}_2 \end{aligned} \tag{34}$$

where $\mathrm{Swish}_\beta(x) = x\sigma(\beta x)$. With residual connections and layer normalization, the $\ell_{\mathrm{th}}$ block is:

$$\mathbf{X}_{\mathrm{SA}}^{(\ell)} = \text{RECTOR-SA}(\mathrm{Norm}(\mathbf{X}^{(\ell)})) + \mathbf{X}^{(\ell)}$$
$$\mathbf{X}^{(\ell+1)} = \mathrm{FFN}_{\mathrm{SwiGLU}}(\mathrm{Norm}(\mathbf{X}_{\mathrm{SA}}^{(\ell)})) + \mathbf{X}_{\mathrm{SA}}^{(\ell)} \tag{35}$$

Here, $\mathbf{X}^{(\ell)}$ and $\mathbf{X}^{(\ell+1)}$ denote the input and output token matrices of the $\ell_{\mathrm{th}}$ block.

## C  RESULTS

### C.1  CLASSIFICATION RESULTS

We also report balanced accuracy in Table 7 and Cohen's kappa in Table 8 on SEED, SEED-IV, and DEAP. Across all datasets, RECTOR maintains the top performance using both evaluation metrics, and it significantly outperforms the other baselines in 15 out of 16 comparisons except subject-independent valence decoding on DEAP using balanced accuracy.

Table 7: Classification performance on SEED, SEED-IV, and DEAP (Balanced Accuracy (%)). SD: Subject Dependent; SI: Subject Independent. **BOLD** and UNDERLINE indicate the best and second-best results. **BOLD**/UNDERLINE indicate the best/second-best results. * denotes it significantly outperforms the second-best model.

| Model | SEED SD | SEED SI | SEED-IV SD | SEED-IV SI | DEAP-Valence SD | DEAP-Valence SI | DEAP-Arousal SD | DEAP-Arousal SI |
|---|---|---|---|---|---|---|---|---|
| TSception | $68.46_{\pm 12.44}$ | $50.42_{\pm 10.22}$ | $49.88_{\pm 14.64}$ | $37.42_{\pm 08.45}$ | $58.46_{\pm 14.88}$ | $56.28_{\pm 15.29}$ | $58.84_{\pm 15.59}$ | $57.70_{\pm 16.30}$ |
| EEG Conformer | $66.47_{\pm 13.97}$ | $51.06_{\pm 10.47}$ | $49.29_{\pm 13.18}$ | $38.18_{\pm 07.62}$ | $59.47_{\pm 12.63}$ | $57.50_{\pm 13.64}$ | $59.63_{\pm 15.07}$ | $58.22_{\pm 15.47}$ |
| LGGNet | $65.58_{\pm 13.72}$ | $47.79_{\pm 10.88}$ | $49.63_{\pm 13.34}$ | $37.44_{\pm 07.58}$ | $57.07_{\pm 13.47}$ | $56.27_{\pm 13.81}$ | $58.83_{\pm 15.85}$ | $59.27_{\pm 15.19}$ |
| PGCN | $70.92_{\pm 13.02}$ | $52.83_{\pm 10.27}$ | $53.85_{\pm 12.59}$ | $38.27_{\pm 08.44}$ | $59.82_{\pm 12.85}$ | $57.28_{\pm 13.96}$ | $59.12_{\pm 14.47}$ | $60.48_{\pm 15.49}$ |
| MASA-TCN | $65.18_{\pm 13.30}$ | $51.92_{\pm 10.28}$ | $51.58_{\pm 13.16}$ | $36.48_{\pm 07.88}$ | $59.28_{\pm 13.17}$ | $58.48_{\pm 14.89}$ | $57.29_{\pm 15.83}$ | $58.85_{\pm 16.18}$ |
| EmT | $70.12_{\pm 13.17}$ | $54.08_{\pm 10.47}$ | $51.70_{\pm 12.18}$ | $36.82_{\pm 08.28}$ | $59.02_{\pm 13.59}$ | $58.49_{\pm 14.18}$ | $59.70_{\pm 16.48}$ | $60.48_{\pm 16.29}$ |
| MMM | $72.01_{\pm 12.01}$ | $54.49_{\pm 09.57}$ | $54.79_{\pm 12.17}$ | $38.22_{\pm 07.44}$ | $59.48_{\pm 12.29}$ | $57.64_{\pm 13.40}$ | $59.10_{\pm 15.19}$ | $58.29_{\pm 15.28}$ |
| LaBraM | $70.24_{\pm 11.94}$ | $53.47_{\pm 10.48}$ | $52.85_{\pm 13.19}$ | $37.91_{\pm 07.92}$ | $60.48_{\pm 12.58}$ | $58.79_{\pm 13.10}$ | $61.27_{\pm 15.59}$ | $60.62_{\pm 15.28}$ |
| REmoNet | $74.58_{\pm 10.29}$ | $54.80_{\pm 10.48}$ | $55.74_{\pm 12.94}$ | $39.91_{\pm 07.49}$ | $62.88_{\pm 12.57}$ | $59.46_{\pm 12.94}$ | $61.70_{\pm 14.47}$ | $61.22_{\pm 15.92}$ |
| CBraMod | $74.82_{\pm 10.29}$ | $53.01_{\pm 09.59}$ | $54.84_{\pm 12.18}$ | $38.66_{\pm 07.37}$ | $59.07_{\pm 12.47}$ | $58.86_{\pm 12.80}$ | $60.72_{\pm 13.06}$ | $61.44_{\pm 14.48}$ |
| PopT | $70.77_{\pm 11.33}$ | $52.85_{\pm 10.44}$ | $52.66_{\pm 13.10}$ | $37.79_{\pm 08.49}$ | $61.74_{\pm 12.29}$ | $59.48_{\pm 12.94}$ | $61.03_{\pm 15.39}$ | $60.52_{\pm 15.05}$ |
| MAE | $70.18_{\pm 10.59}$ | $50.74_{\pm 10.29}$ | $53.22_{\pm 12.01}$ | $38.80_{\pm 07.58}$ | $58.82_{\pm 12.30}$ | $57.70_{\pm 12.94}$ | $59.96_{\pm 14.85}$ | $59.47_{\pm 15.72}$ |
| JEPA | $72.46_{\pm 10.39}$ | $54.28_{\pm 10.59}$ | $54.10_{\pm 12.59}$ | $38.72_{\pm 08.50}$ | $61.47_{\pm 12.94}$ | $59.10_{\pm 12.29}$ | $62.47_{\pm 15.01}$ | $61.34_{\pm 14.49}$ |
| CAE | $72.89_{\pm 10.48}$ | $52.49_{\pm 09.94}$ | $54.72_{\pm 12.20}$ | $39.64_{\pm 07.94}$ | $60.17_{\pm 11.59}$ | $59.44_{\pm 12.26}$ | $60.42_{\pm 13.54}$ | $60.07_{\pm 14.19}$ |
| **RECTOR** | **$77.64^*_{\pm 10.08}$** | **$56.28^*_{\pm 09.47}$** | **$57.74^*_{\pm 12.38}$** | **$41.46^*_{\pm 07.08}$** | **$63.88^*_{\pm 11.40}$** | **$60.45_{\pm 12.07}$** | **$64.40^*_{\pm 13.17}$** | **$63.06^*_{\pm 13.46}$** |

Table 8: Classification performance on SEED, SEED-IV, and DEAP (Cohen's Kappa (%)). SD: Subject Dependent; SI: Subject Independent. **BOLD** and UNDERLINE indicate the best and second-best results. **BOLD**/UNDERLINE indicate the best/second-best results. * denotes it significantly outperforms the second-best model.

| Model | SEED SD | SEED SI | SEED-IV SD | SEED-IV SI | DEAP-Valence SD | DEAP-Valence SI | DEAP-Arousal SD | DEAP-Arousal SI |
|---|---|---|---|---|---|---|---|---|
| TSception | $56.99_{\pm 15.77}$ | $23.60_{\pm 10.50}$ | $33.90_{\pm 14.19}$ | $12.06_{\pm 05.12}$ | $20.47_{\pm 09.42}$ | $17.90_{\pm 08.61}$ | $21.74_{\pm 09.55}$ | $18.87_{\pm 09.11}$ |
| EEG Conformer | $49.63_{\pm 17.30}$ | $25.43_{\pm 10.51}$ | $34.39_{\pm 13.48}$ | $14.01_{\pm 05.80}$ | $23.82_{\pm 09.22}$ | $20.50_{\pm 08.15}$ | $24.02_{\pm 09.92}$ | $20.15_{\pm 08.85}$ |
| LGGNet | $50.39_{\pm 16.30}$ | $22.75_{\pm 10.21}$ | $31.33_{\pm 13.15}$ | $13.01_{\pm 05.66}$ | $21.61_{\pm 09.24}$ | $17.41_{\pm 08.16}$ | $21.30_{\pm 09.92}$ | $21.42_{\pm 08.80}$ |
| PGCN | $60.54_{\pm 16.32}$ | $31.15_{\pm 09.59}$ | $39.24_{\pm 12.52}$ | $16.00_{\pm 05.89}$ | $24.79_{\pm 09.02}$ | $19.92_{\pm 08.01}$ | $23.58_{\pm 09.86}$ | $24.81_{\pm 08.83}$ |
| MASA-TCN | $51.95_{\pm 16.02}$ | $29.42_{\pm 09.38}$ | $36.05_{\pm 13.07}$ | $13.79_{\pm 05.68}$ | $24.14_{\pm 09.40}$ | $20.99_{\pm 08.60}$ | $19.73_{\pm 09.69}$ | $20.79_{\pm 09.75}$ |
| EmT | $59.49_{\pm 15.42}$ | $32.57_{\pm 09.79}$ | $36.54_{\pm 12.02}$ | $14.87_{\pm 06.17}$ | $23.78_{\pm 10.08}$ | $21.71_{\pm 09.05}$ | $23.63_{\pm 09.88}$ | $24.69_{\pm 09.38}$ |
| MMM | $61.68_{\pm 14.65}$ | $32.48_{\pm 09.23}$ | $40.32_{\pm 11.84}$ | $16.02_{\pm 05.20}$ | $24.31_{\pm 09.21}$ | $20.68_{\pm 08.81}$ | $22.55_{\pm 09.35}$ | $21.18_{\pm 09.61}$ |
| LaBraM | $59.30_{\pm 15.59}$ | $31.95_{\pm 09.62}$ | $36.74_{\pm 12.82}$ | $15.98_{\pm 05.55}$ | $25.16_{\pm 09.69}$ | $21.73_{\pm 08.13}$ | $26.59_{\pm 09.48}$ | $25.50_{\pm 09.47}$ |
| REmoNet | $66.16_{\pm 13.34}$ | $33.99_{\pm 10.17}$ | $42.01_{\pm 12.59}$ | $18.39_{\pm 05.98}$ | $29.96_{\pm 09.30}$ | $23.44_{\pm 08.87}$ | $28.71_{\pm 09.37}$ | $26.80_{\pm 08.63}$ |
| CBraMod | $64.48_{\pm 13.31}$ | $32.88_{\pm 09.45}$ | $39.81_{\pm 11.95}$ | $17.83_{\pm 05.61}$ | $24.83_{\pm 09.44}$ | $23.05_{\pm 08.29}$ | $26.48_{\pm 09.13}$ | $27.11_{\pm 09.52}$ |
| PopT | $62.69_{\pm 14.76}$ | $32.85_{\pm 09.35}$ | $37.25_{\pm 12.68}$ | $15.66_{\pm 05.81}$ | $26.73_{\pm 09.10}$ | $23.55_{\pm 08.96}$ | $25.88_{\pm 09.32}$ | $24.94_{\pm 08.60}$ |
| MAE | $58.67_{\pm 13.56}$ | $31.79_{\pm 10.28}$ | $37.75_{\pm 11.84}$ | $16.85_{\pm 05.40}$ | $22.77_{\pm 09.45}$ | $20.35_{\pm 08.95}$ | $24.56_{\pm 09.41}$ | $23.46_{\pm 09.64}$ |
| JEPA | $62.20_{\pm 13.02}$ | $33.70_{\pm 09.59}$ | $39.16_{\pm 11.60}$ | $17.50_{\pm 05.73}$ | $27.22_{\pm 09.42}$ | $22.21_{\pm 08.99}$ | $28.35_{\pm 08.80}$ | $26.75_{\pm 08.73}$ |
| CAE | $62.76_{\pm 13.42}$ | $31.86_{\pm 09.82}$ | $40.28_{\pm 11.70}$ | $18.62_{\pm 05.30}$ | $26.31_{\pm 08.98}$ | $23.45_{\pm 08.90}$ | $26.15_{\pm 08.99}$ | $24.87_{\pm 09.47}$ |
| **RECTOR** | **$71.17^*_{\pm 09.27}$** | **$36.86^*_{\pm 10.08}$** | **$44.91^*_{\pm 11.12}$** | **$21.50^*_{\pm 05.50}$** | **$32.98^*_{\pm 09.35}$** | **$26.99^*_{\pm 08.40}$** | **$33.84^*_{\pm 09.53}$** | **$31.27^*_{\pm 09.28}$** |

We have also evaluated all models on MSIT and ECR using 5-fold cross-validation. Results are reported in Table 9, demonstrating a consistent performance with our chronological split and confirming that our findings are robust under both evaluation protocols.

Table 9: Classification performance on MSIT and ECR using 5-fold cross-validation (AUROC (%)). **BOLD**/UNDERLINE indicates the best/second-best results. * denotes it significantly outperforms the second-best model.

| Model | MSIT | ECR |
|---|---|---|
| TSception | $83.47_{\pm07.86}$ | $83.30_{\pm07.90}$ |
| EEG Conformer | $83.31_{\pm06.68}$ | $82.89_{\pm06.58}$ |
| LGGNet | $83.10_{\pm06.74}$ | $82.30_{\pm07.22}$ |
| Seegnificant | $84.08_{\pm06.88}$ | $84.69_{\pm07.05}$ |
| BrainBERT | $85.12_{\pm06.58}$ | $86.03_{\pm06.79}$ |
| Brant | $86.42_{\pm06.49}$ | $86.29_{\pm06.27}$ |
| Du-IN | $\underline{86.68}_{\pm05.80}$ | $86.36_{\pm05.94}$ |
| PopT | $86.58_{\pm05.90}$ | $\underline{87.44}_{\pm05.62}$ |
| MAE | $85.91_{\pm06.47}$ | $85.41_{\pm06.10}$ |
| JEPA | $86.39_{\pm06.17}$ | $87.42_{\pm05.57}$ |
| CAE | $86.37_{\pm06.23}$ | $87.18_{\pm06.22}$ |
| **RECTOR** | $\mathbf{90.18}^{*}_{\pm05.49}$ | $\mathbf{90.36}^{*}_{\pm05.67}$ |

## C.2 IMPACT OF EXPANDED PRE-TRAINING REGIMES

Table 10: Classification performance on DEAP using expanded dataset for pre-training (Weighted F1 Score (%)). SD: Subject Dependent; SI: Subject Independent. **BOLD**/UNDERLINE indicates the best/second-best results. * denotes it significantly outperforms training from scratch.

| Model | Pre-Training Dataset | DEAP-V | | DEAP-A | |
|---|---|---|---|---|---|
| | | SD | SI | SD | SI |
| | None | $65.00_{\pm13.18}$ | $62.69_{\pm14.56}$ | $64.41_{\pm15.21}$ | $64.31_{\pm15.42}$ |
| **RECTOR** | DEAP | $68.75^{*}_{\pm10.10}$ | $66.22^{*}_{\pm11.33}$ | $69.68^{*}_{\pm12.52}$ | $68.01^{*}_{\pm12.72}$ |
| **RECTOR+** | DEAP+SEED+SEED-IV+CHB-MIT | $\underline{69.61}^{*}_{\pm09.88}$ | $\underline{66.24}^{*}_{\pm11.48}$ | $\underline{70.02}^{*}_{\pm12.23}$ | $\underline{68.52}^{*}_{\pm12.47}$ |
| **RECTOR-L+** | DEAP+SEED+SEED-IV+CHB-MIT | $\mathbf{69.82}^{*}_{\pm09.48}$ | $\mathbf{66.83}^{*}_{\pm10.94}$ | $\mathbf{70.56}^{*}_{\pm11.82}$ | $\mathbf{68.57}^{*}_{\pm12.42}$ |

Table 10 reports downstream classification on DEAP under progressively expanded pre-training regimes. We evaluate four pre-training regimes, ranging from: (1) training from scratch to pre-training on (2) DEAP, (3) all candidate datasets (**RECTOR+**, CHB-MIT (Shoeb, 2009) is added as a large candidate for EEG), (4) all candidate datasets using a larger RECTOR (**RECTOR-L+**). RECTOR-L+ yields the best performance across various evaluation protocols on DEAP.

RECTOR consistently outperforms foundation models when all models are pre-trained on the same expanded dataset. Table 11 shows the results of this comparison, where RECTOR, LaBraM, and CBraMod were all pre-trained on an expanded dataset comprising SEED, SEED-IV, DEAP, and CHB-MIT. Across all downstream classification tasks, RECTOR demonstrates a clear performance advantage.

Table 11: Classification performance on SEED, SEED-IV, and DEAP using expanded dataset for pre-training RECTOR and foundation models (Weighted F1 Score (%)). SD: Subject Dependent; SI: Subject Independent. **BOLD**/UNDERLINE indicates the best/second-best results. * denotes it significantly outperforms the second-best model.

| | SEED | | SEED-IV | | DEAP-V | | DEAP-A | |
|---|---|---|---|---|---|---|---|---|
| | SD | SI | SD | SI | SD | SI | SD | SI |
| **LaBraM** | $77.04^{*}_{\pm10.04}$ | $57.92^{*}_{\pm11.30}$ | $57.24^{*}_{\pm12.70}$ | $42.17^{*}_{\pm12.37}$ | $65.42^{*}_{\pm10.62}$ | $63.78^{*}_{\pm11.07}$ | $67.04^{*}_{\pm12.84}$ | $65.38^{*}_{\pm12.52}$ |
| **CBraMod** | $\underline{79.26}^{*}_{\pm09.52}$ | $\underline{59.26}^{*}_{\pm10.28}$ | $\underline{59.06}^{*}_{\pm11.46}$ | $\underline{43.58}^{*}_{\pm06.33}$ | $\underline{65.28}^{*}_{\pm09.92}$ | $\underline{64.35}^{*}_{\pm11.70}$ | $\underline{66.58}^{*}_{\pm12.52}$ | $\underline{66.42}^{*}_{\pm12.67}$ |
| **RECTOR-L+** | $\mathbf{85.50}^{*}_{\pm08.82}$ | $\mathbf{61.69}^{*}_{\pm09.01}$ | $\mathbf{64.21}^{*}_{\pm10.93}$ | $\mathbf{45.76}^{*}_{\pm06.14}$ | $\mathbf{69.82}^{*}_{\pm09.48}$ | $\mathbf{66.83}^{*}_{\pm10.94}$ | $\mathbf{70.56}^{*}_{\pm11.82}$ | $\mathbf{68.57}^{*}_{\pm12.42}$ |

## C.3 ABLATION STUDIES

**Masking Strategy** Tables 12 and 13 present ablation studies on masking strategy. We define **Random Mask** as MAE-style patch-level random masking, and **Block Mask** as selecting one single block as the target without considering the spatio-temporal structure in the neural signals. $-\times r$ Mask removes masking on the region domain, $-\times c$ Mask removes masking on the channel domain, and $-\times t$ Mask removes masking on the temporal domain.

Across both EEG emotion and sEEG task engagement tasks, RECTOR-Mask outperforms all alternatives. Random and Block Masking incur the largest performance drops, confirming the importance

of preserving spatio-temporal context. Ablations that omit either the region, channel, or temporal domain also degrade accuracy, demonstrating that jointly masking region, channel, and time is essential for learning rich neural representations.

Table 12: Ablation studies on masking strategy, evaluated on SEED, SEED-IV, and DEAP (Weighted F1 Score (%)). SD: Subject Dependent; SI: Subject Independent. **BOLD** indicates the best result. * denotes it is significantly worse than the full model.

| Model | SEED | | SEED-IV | | DEAP-Valence | | DEAP-Arousal | |
|---|---|---|---|---|---|---|---|---|
| | SD | SI | SD | SI | SD | SI | SD | SI |
| Random Mask | $81.84^*_{\pm09.84}$ | $59.47^*_{\pm09.46}$ | $61.07^*_{\pm12.57}$ | $42.80^*_{\pm07.28}$ | $66.81^*_{\pm11.47}$ | $64.19^*_{\pm12.22}$ | $67.38^*_{\pm13.58}$ | $66.31^*_{\pm14.08}$ |
| Block Mask | $81.98^*_{\pm09.76}$ | $59.32^*_{\pm09.88}$ | $61.22^*_{\pm12.46}$ | $42.58^*_{\pm07.36}$ | $66.67^*_{\pm11.10}$ | $64.38^*_{\pm12.47}$ | $67.44^*_{\pm12.57}$ | $67.01^*_{\pm12.83}$ |
| $-\times r$ Mask | $82.81^*_{\pm09.46}$ | $59.89_{\pm09.02}$ | $61.99^*_{\pm12.27}$ | $43.57^*_{\pm06.58}$ | $67.32^*_{\pm10.76}$ | $64.56^*_{\pm11.41}$ | $68.53^*_{\pm12.37}$ | $66.78^*_{\pm12.57}$ |
| $-\times c$ Mask | $83.18^*_{\pm09.68}$ | $60.15^*_{\pm09.47}$ | $62.20^*_{\pm12.02}$ | $44.09^*_{\pm06.74}$ | $68.20^*_{\pm10.28}$ | $65.09^*_{\pm11.17}$ | $69.12^*_{\pm12.73}$ | $66.77^*_{\pm12.86}$ |
| $-\times t$ Mask | $83.30^*_{\pm09.27}$ | $60.02^*_{\pm09.76}$ | $62.32^*_{\pm11.85}$ | $44.48^*_{\pm06.52}$ | $68.18^*_{\pm10.47}$ | $65.58^*_{\pm12.02}$ | $68.36^*_{\pm12.52}$ | $67.07^*_{\pm13.35}$ |
| **RECTOR-Mask** | $\mathbf{84.32}_{\pm08.80}$ | $\mathbf{60.58}_{\pm09.12}$ | $\mathbf{63.07}_{\pm11.16}$ | $\mathbf{44.79}_{\pm06.56}$ | $\mathbf{68.75}_{\pm10.10}$ | $\mathbf{66.22}_{\pm11.33}$ | $\mathbf{69.68}_{\pm12.52}$ | $\mathbf{68.01}_{\pm12.72}$ |

Table 13: Ablation studies on masking strategy, evaluated on MSIT and ECR (AUROC (%)). **BOLD** indicates the best result. * denotes it is significantly worse than the full model.

| Model | MSIT | ECR |
|---|---|---|
| Random Masking | $88.16^*_{\pm06.48}$ | $87.92^*_{\pm06.38}$ |
| Block Masking | $88.19^*_{\pm06.35}$ | $88.24^*_{\pm06.12}$ |
| $-\times r$ Mask | $88.79^*_{\pm05.68}$ | $89.35^*_{\pm06.12}$ |
| $-\times c$ Mask | $89.39^*_{\pm05.73}$ | $89.52^*_{\pm05.94}$ |
| $-\times t$ Mask | $89.30^*_{\pm05.46}$ | $89.42^*_{\pm05.74}$ |
| **RECTOR-Mask** | $\mathbf{90.55}_{\pm05.17}$ | $\mathbf{90.79}_{\pm05.36}$ |

**Region, Channel, Temporal Tokens** Tables 14 and 15 present ablation studies on different tokens. We define $-$ Region Tokens as not using region tokens for self-supervised learning, $-\times c$ as not using channel tokens for self-supervised learning while still using region tokens like (Yi et al., 2023), $-\times t$ as directly using the entire neural signal without segmenting it into temporal patches, and $-$ Conditioning Tokens as not concatenating conditioning tokens to channel and region tokens. Across both EEG emotion and sEEG task engagement tasks, RECTOR using all tokens outperforms all alternatives, confirming the importance of each individual type of tokens.

Table 14: Ablation studies on region, channel, and temporal tokens, evaluated on SEED, SEED-IV, and DEAP (Weighted F1 Score (%)). SD: Subject Dependent; SI: Subject Independent. **BOLD** indicates the best result. * denotes it is significantly worse than the full model.

| Model | SEED | | SEED-IV | | DEAP-Valence | | DEAP-Arousal | |
|---|---|---|---|---|---|---|---|---|
| | SD | SI | SD | SI | SD | SI | SD | SI |
| $-$ Region Tokens | $82.52^*_{\pm09.76}$ | $59.68^*_{\pm09.94}$ | $61.78^*_{\pm12.47}$ | $43.03^*_{\pm07.06}$ | $67.28^*_{\pm11.55}$ | $64.71^*_{\pm12.28}$ | $67.92^*_{\pm13.69}$ | $66.72^*_{\pm14.02}$ |
| $-$ Channel Tokens | $82.57^*_{\pm09.10}$ | $59.48^*_{\pm09.44}$ | $61.64^*_{\pm12.28}$ | $43.13^*_{\pm07.42}$ | $67.64^*_{\pm11.10}$ | $64.92^*_{\pm12.62}$ | $67.96^*_{\pm12.03}$ | $66.53^*_{\pm13.17}$ |
| $-$ Temporal Tokens | $82.83^*_{\pm09.55}$ | $59.66^*_{\pm09.47}$ | $61.86^*_{\pm12.66}$ | $44.22^*_{\pm06.48}$ | $67.92^*_{\pm10.26}$ | $65.21^*_{\pm11.55}$ | $68.49^*_{\pm12.47}$ | $66.79^*_{\pm13.12}$ |
| $-$ Conditioning Tokens | $82.98^*_{\pm09.28}$ | $60.02^*_{\pm09.94}$ | $62.28^*_{\pm12.47}$ | $43.42^*_{\pm06.18}$ | $67.80^*_{\pm10.48}$ | $64.96^*_{\pm11.92}$ | $68.37^*_{\pm12.40}$ | $66.96^*_{\pm13.54}$ |
| **RECTOR** | $\mathbf{84.32}_{\pm08.80}$ | $\mathbf{60.58}_{\pm09.12}$ | $\mathbf{63.07}_{\pm11.16}$ | $\mathbf{44.79}_{\pm06.56}$ | $\mathbf{68.75}_{\pm10.10}$ | $\mathbf{66.22}_{\pm11.33}$ | $\mathbf{69.68}_{\pm12.52}$ | $\mathbf{68.01}_{\pm12.72}$ |

Table 15: Ablation studies on region, channel, and temporal tokens, evaluated on MSIT and ECR (AUROC (%)). **BOLD** indicates the best result. * denotes it is significantly worse than the full model.

| Model | MSIT | ECR |
|---|---|---|
| $-$ Region Tokens | $88.43^*_{\pm06.34}$ | $88.94^*_{\pm06.02}$ |
| $-$ Channel Tokens | $88.44^*_{\pm06.10}$ | $88.79^*_{\pm06.17}$ |
| $-$ Temporal Tokens | $88.78^*_{\pm05.89}$ | $89.01^*_{\pm05.86}$ |
| $-$ Conditioning Tokens | $89.02^*_{\pm05.33}$ | $89.27^*_{\pm05.55}$ |
| **RECTOR** | $\mathbf{90.55}_{\pm05.17}$ | $\mathbf{90.79}_{\pm05.36}$ |

**Self-Attention** Tables 16 and 17 present ablation studies on self-attention. **Full SA** replaces the dynamic region-channel-temporal attention mask with a dense spatio-temporal mask, treating region tokens as generic global tokens. **Static SA** conducts all self-attention components in RECTOR-SA in a static way without dynamic gating. Across both EEG emotion and sEEG task engagement tasks, each modification degrades performance: Full self-attention incurs the largest drop. These results

confirm that anatomical attention and data-adaptive gating are both critical to modeling hierarchical region–channel–temporal interactions for downstream cognitive tasks.

Table 16: Ablation studies on self-attention, evaluated on SEED, SEED-IV, and DEAP (Weighted F1 Score (%)). SD: Subject Dependent; SI: Subject Independent. **BOLD** indicates the best result. * denotes it is significantly worse than the full model.

| Model | SEED | | SEED-IV | | DEAP-Valence | | DEAP-Arousal | |
|---|---|---|---|---|---|---|---|---|
| | SD | SI | SD | SI | SD | SI | SD | SI |
| Full SA | $82.47^*_{\pm09.63}$ | $59.60^*_{\pm09.56}$ | $61.27^*_{\pm12.40}$ | $43.53^*_{\pm06.93}$ | $67.31^*_{\pm11.08}$ | $65.16^*_{\pm11.70}$ | $67.72^*_{\pm13.19}$ | $66.44^*_{\pm13.36}$ |
| Static SA | $82.87^*_{\pm09.38}$ | $60.01^*_{\pm09.92}$ | $61.58^*_{\pm12.21}$ | $44.04^*_{\pm07.17}$ | $68.09^*_{\pm11.01}$ | $65.48^*_{\pm11.96}$ | $68.59^*_{\pm12.52}$ | $67.15^*_{\pm13.10}$ |
| **RECTOR-SA** | $\mathbf{84.32}_{\pm08.80}$ | $\mathbf{60.58}_{\pm09.12}$ | $\mathbf{63.07}_{\pm11.16}$ | $\mathbf{44.79}_{\pm06.56}$ | $\mathbf{68.75}_{\pm10.10}$ | $\mathbf{66.22}_{\pm11.33}$ | $\mathbf{69.68}_{\pm12.52}$ | $\mathbf{68.01}_{\pm12.72}$ |

Table 17: Ablation studies on self-attention, evaluated on MSIT and ECR (AUROC (%)). **BOLD** indicates the best result. * denotes it is significantly worse than the full model.

| Model | MSIT | ECR |
|---|---|---|
| Full SA | $88.94^*_{\pm06.12}$ | $88.82^*_{\pm05.69}$ |
| Static SA | $89.58_{\pm06.27}$ | $89.44^*_{\pm05.48}$ |
| **RECTOR-SA** | $\mathbf{90.55}_{\pm05.17}$ | $\mathbf{90.79}_{\pm05.36}$ |

**VCReg**    Tables 18 and 19 present ablation studies on the variance-covariance regularization (VCReg) module. The results show that marginal performance degradation after removing the VCReg module. We hypothesize that this result occurs because the input reconstruction process becomes the primary mechanism for preserving informative embeddings and avoiding representational collapse.

Table 18: Ablation studies on VCReg, evaluated on SEED, SEED-IV, and DEAP (Weighted F1 Score (%)). SD: Subject Dependent; SI: Subject Independent. **BOLD** indicates the best result. * denotes it is significantly worse than the full model.

| Model | SEED | | SEED-IV | | DEAP-Valence | | DEAP-Arousal | |
|---|---|---|---|---|---|---|---|---|
| | SD | SI | SD | SI | SD | SI | SD | SI |
| $-\mathcal{L}_{\mathrm{VCReg}}$ | $83.08^*_{\pm08.46}$ | $\mathbf{60.62}_{\pm08.85}$ | $62.38_{\pm11.79}$ | $44.02_{\pm07.27}$ | $67.80^*_{\pm09.57}$ | $65.64_{\pm11.21}$ | $68.94_{\pm12.32}$ | $66.96^*_{\pm12.88}$ |
| **RECTOR** | $\mathbf{84.32}_{\pm08.80}$ | $60.58_{\pm09.12}$ | $\mathbf{63.07}_{\pm11.16}$ | $\mathbf{44.79}_{\pm06.56}$ | $\mathbf{68.75}_{\pm10.10}$ | $\mathbf{66.22}_{\pm11.33}$ | $\mathbf{69.68}_{\pm12.52}$ | $\mathbf{68.01}_{\pm12.72}$ |

Table 19: Ablation studies on VCReg, evaluated on MSIT and ECR (AUROC (%)). **BOLD** indicates the best result. * denotes it is significantly worse than the full model.

| Model | MSIT | ECR |
|---|---|---|
| $-\mathcal{L}_{\mathrm{VCReg}}$ | $89.98_{\pm05.76}$ | $89.72^*_{\pm05.58}$ |
| **RECTOR** | $\mathbf{90.55}_{\pm05.17}$ | $\mathbf{90.79}_{\pm05.36}$ |

**Non-Contrastive Loss**    In Table 20, we show the input reconstruction loss ($\mathcal{L}_{\mathrm{input}}$) and representation alignment loss ($\mathcal{L}_{\mathrm{rep}}$) of RECTOR and how the losses are changed if we remove the other loss component in self-supervised learning. Together with Tables 3 and 4, we observed that removing $\mathcal{L}_{\mathrm{input}}$ reduces $\mathcal{L}_{\mathrm{rep}}$ but degrades downstream performance, indicating a slight feature collapse. Conversely, omitting $\mathcal{L}_{\mathrm{rep}}$ causes a substantial increase in $\mathcal{L}_{\mathrm{input}}$ and also worsens downstream results. These findings confirm that both objectives are essential.

Table 20: Ablation studies on MSE loss between predicted and ground-truth neural representations ($\mathcal{L}_{\mathrm{rep}}$) and input signals ($\mathcal{L}_{\mathrm{input}}$). We used $\lambda_{\mathrm{rep}} = 1$ and $\lambda_{\mathrm{input}} = 0.5$ with standardized neural inputs. **BOLD** indicates the best result.

| Model | SEED | | SEED-IV | | DEAP | | MSIT | | ECR | |
|---|---|---|---|---|---|---|---|---|---|---|
| | $\mathcal{L}_{\mathrm{rep}}$ | $\mathcal{L}_{\mathrm{input}}$ | $\mathcal{L}_{\mathrm{rep}}$ | $\mathcal{L}_{\mathrm{input}}$ | $\mathcal{L}_{\mathrm{rep}}$ | $\mathcal{L}_{\mathrm{input}}$ | $\mathcal{L}_{\mathrm{rep}}$ | $\mathcal{L}_{\mathrm{input}}$ | $\mathcal{L}_{\mathrm{rep}}$ | $\mathcal{L}_{\mathrm{input}}$ |
| $-\mathcal{L}_{\mathrm{rep}}$ | N/A | 0.3872 | N/A | 0.4165 | N/A | 0.4230 | N/A | 0.3828 | N/A | 0.3783 |
| $-\mathcal{L}_{\mathrm{input}}$ | **0.0352** | N/A | **0.0197** | N/A | **0.0409** | N/A | **0.0261** | N/A | **0.0244** | N/A |
| **RECTOR** | 0.0413 | **0.2081** | 0.0200 | **0.1849** | 0.0457 | **0.2224** | 0.0293 | **0.2376** | 0.0264 | **0.2331** |

## C.4 COMPARISON WITH CONTRASTIVE LEARNING BASELINES

We further compared RECTOR against contrastive learning methods for EEG emotion recognition (**CLISA** Shen et al. (2022)) and medical time-series data (**TF-C** Zhang et al. (2022), **COMET** Wang et al. (2023c)), which are specifically designed for cross-subject applications. We conducted experiments on subject-independent paradigms on SEED, SEED-IV, and DEAP, using the same time-series input. The performance in Table 21 shows that RECTOR even outperforms the contrastive learning baselines, which specialize in cross-subject paradigm. Only on the SEED dataset, RECTOR achieves a marginally lower performance compared with CLISA. These findings confirm that REC-TOR acquires robust, subject-invariant representations capable of capturing inter-individual neural variability in affective and cognitive states.

Table 21: Comparison with contrastive learning baselines for subject-independent performance on SEED, SEED-IV, and DEAP (Weighted F1 Score (%)). **BOLD**: the best performance. UNDERLINE: the second-best performance.

| Model | SEED | SEED-IV | DEAP-Valence | DEAP-Arousal |
|---|---|---|---|---|
| CLISA | **60.58**$_{\pm 09.35}$ | 43.36$_{\pm 06.46}$ | 64.13$_{\pm 11.41}$ | 65.86$_{\pm 12.93}$ |
| TF-C | 58.62$_{\pm 09.35}$ | 42.28$_{\pm 06.46}$ | 63.05$_{\pm 11.41}$ | 64.59$_{\pm 12.93}$ |
| COMET | 60.05$_{\pm 09.35}$ | 44.02$_{\pm 06.46}$ | 64.82$_{\pm 11.41}$ | 66.87$_{\pm 12.93}$ |
| **RECTOR** | 60.53$_{\pm 09.35}$ | **44.67**$_{\pm 06.46}$ | **65.94**$_{\pm 11.41}$ | **67.84**$_{\pm 12.93}$ |

## C.5 ROBUSTNESS TO VARIOUS PARCELLATION SCHEMES

We conducted a sensitivity analysis to assess RECTOR's robustness to different anatomical parcellation schemes on the SEED, SEED-IV, and DEAP datasets. In addition to our 11-region baseline, we evaluated performance using a 9-region hemispheric scheme (Ding et al., 2023) and a high-resolution 17-region parcellation scheme (Yi et al., 2023), as shown in Fig. 8.

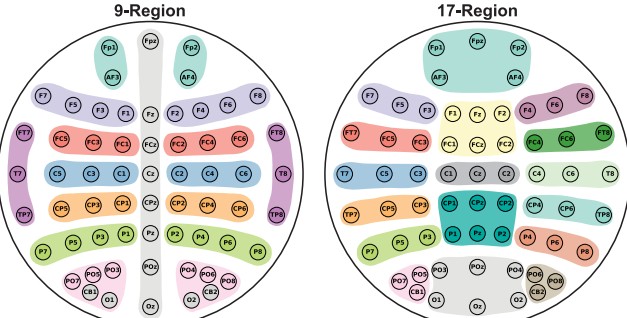

Figure 8: 9-Region (Left) and 17-Region (Right) EEG parcellation schemes, with colored overlays indicating anatomically defined functional regions.

The results in Table 22 demonstrate that RECTOR is highly robust to these variations in anatomical priors. Minor performance degradation was observed when using the dense 17-region parcellation scheme on the DEAP (32-channel) dataset. We attribute this slight drop to the low channel density of the input. Extracting 17 distinct region-common but not channel-specific features from only 32 channels is very difficult, and it significantly reduces the benefits inherent in our hierarchical structure. This robustness confirms that RECTOR is not overfit to a single, arbitrary parcellation choice but effectively learns the fundamental hierarchical nature of the data, supporting the generalizability of our approach.

Table 22: Classification performance on SEED, SEED-IV, and DEAP with various parcellation schemes (Weighted F1 Score (%)). **BOLD**: the best performance. UNDERLINE: the second-best performance. * denotes it is significantly better than the second-best model.

| Model | SEED | | SEED-IV | | DEAP-Valence | | DEAP-Arousal | |
|---|---|---|---|---|---|---|---|---|
| | SD | SI | SD | SI | SD | SI | SD | SI |
| 9-Region (Ding et al., 2023) | 83.84$_{\pm 09.02}$ | **60.85**$_{\pm 09.22}$ | 62.77$_{\pm 11.64}$ | 43.87$_{\pm 06.83}$ | **69.02**$_{\pm 09.79}$ | 65.80$_{\pm 11.43}$ | 69.42$_{\pm 12.81}$ | 67.53$_{\pm 13.32}$ |
| 17-Region (Yi et al., 2023) | 83.27$_{\pm 09.30}$ | 59.10$_{\pm 09.62}$ | 62.69$_{\pm 12.03}$ | **44.89**$_{\pm 06.72}$ | 66.88$_{\pm 11.27}$ | 64.26$_{\pm 12.70}$ | 67.74$_{\pm 13.78}$ | 65.97$_{\pm 14.46}$ |
| 11-Region (**RECTOR**) | **84.32**$_{\pm 08.80}$ | 60.58$_{\pm 09.12}$ | **63.07**$_{\pm 11.16}$ | 44.79$_{\pm 06.56}$ | 68.75$_{\pm 10.10}$ | **66.22**$_{\pm 11.33}$ | **69.68**$_{\pm 12.52}$ | **68.01**$_{\pm 12.72}$ |

## C.6 ROBUSTNESS TO LOW-DENSITY SETUP

We conducted an explicit analysis to assess RECTOR's robustness to low-density setup. We randomly dropped 50% of the EEG channels (resulting in 31 channels, comparable to DEAP) in the SEED and SEED-IV datasets, and 50% of the sEEG channels in MSIT and ECR. We omitted DEAP from this simulation as its channel count is already low. To ensure reliable metrics, channel sub-sampling was performed randomly for 10 trials, with performance averaged across all experiments for each model.

We compared RECTOR against the two strongest EEG and sEEG baselines (from Tables 1 and 2). The results shown in Tables 23 and 24 confirm that RECTOR's hierarchical structure proves highly robust to the low-density setup. Although all models experience a performance drop, RECTOR's performance remains significantly superior to all tested baselines. This robustness demonstrates that the region-level tokens in RECTOR-SA effectively learn to generalize and maintain predictive power even when large portions of the underlying channel information are lost.

Table 23: Classification performance on SEED, SEED-IV using 50% and 100% of EEG channels (Weighted F1 Score (%)). **BOLD**: the best performance. UNDERLINE: the second-best performance. * denotes it is significantly better than the second-best model within the same comparison setting.

| Model | SEED | | SEED-IV | |
|---|---|---|---|---|
| | SD | SI | SD | SI |
| REmoNet (50%) | 70.18$_{\pm 11.52}$ | 49.58$_{\pm 12.33}$ | 54.54$_{\pm 13.88}$ | 38.15$_{\pm 07.73}$ |
| REmoNet (100%) | 80.67$_{\pm 10.24}$ | 58.33$_{\pm 09.78}$ | 60.69$_{\pm 13.04}$ | 41.93$_{\pm 07.01}$ |
| CBraMod (50%) | 69.58$_{\pm 11.86}$ | 51.46$_{\pm 12.04}$ | 50.81$_{\pm 13.92}$ | 37.82$_{\pm 08.90}$ |
| CBraMod (100%) | 78.18$_{\pm 10.19}$ | 58.67$_{\pm 09.57}$ | 58.27$_{\pm 12.74}$ | 42.40$_{\pm 06.47}$ |
| **RECTOR** (50%) | **75.04**$^{*}_{\pm 10.98}$ | **53.85**$^{*}_{\pm 10.84}$ | **56.45**$^{*}_{\pm 13.01}$ | **39.95**$^{*}_{\pm 08.46}$ |
| **RECTOR** (100%) | **84.32**$^{*}_{\pm 08.80}$ | **60.58**$^{*}_{\pm 09.12}$ | **63.07**$^{*}_{\pm 11.16}$ | **44.79**$^{*}_{\pm 06.56}$ |

Table 24: Classification performance on MSIT and ECR using 50% and 100% of sEEG channels (AUROC (%)). **BOLD**: the best performance. UNDERLINE: the second-best performance. * denotes it is significantly better than the second-best model within the same comparison setting.

| Model | MSIT | ECR |
|---|---|---|
| Du-IN (50%) | 80.31$_{\pm 07.66}$ | 78.08$_{\pm 08.49}$ |
| Du-IN (100%) | 87.29$_{\pm 05.43}$ | 86.18$_{\pm 05.89}$ |
| PopT (50%) | 78.68$_{\pm 08.65}$ | 80.86$_{\pm 07.96}$ |
| PopT (100%) | 87.03$_{\pm 05.84}$ | 87.99$_{\pm 05.58}$ |
| **RECTOR** (50%) | **82.85**$^{*}_{\pm 07.46}$ | **83.01**$^{*}_{\pm 07.83}$ |
| **RECTOR** (100%) | **90.55**$^{*}_{\pm 05.17}$ | **90.79**$^{*}_{\pm 05.36}$ |

## C.7 INTER-SUBJECT AND INTRA-SUBJECT VARIANCE

In Table 25, we provide a granular breakdown of the total standard deviation into inter-subject and intra-subject standard deviations for the SEED and SEED-IV subject-dependent (SD) protocols. The total standard deviation is calculated as the standard deviation across the weighted F1-Score of all subjects and sessions together. The inter-subject term is calculated as the standard deviation across the weighted F1-Score (averaged across sessions) of different subjects. The intra-subject term is calculated as the cross-subject average of the standard deviation (across sessions).

This rigorous analysis confirms our model's robustness: the majority of the total variance is attributed to differences between subjects, while the variability within the same subject across different sessions is relatively smaller. This result proves that RECTOR learns stable representations for individual users over time.

Table 25: Total, inter-subject, and intra-subject standard deviation on SEED and SEED-IV under subject-dependent (SD) protocol (Weighted F1 Score (%)). **BOLD**: the best performance.

| Model | SEED:SD | | | SEED-IV:SD | | |
|---|---|---|---|---|---|---|
| | Total | Inter-Subject | Intra-Subject | Total | Inter-Subject | Intra-Subject |
| **RECTOR** | ±08.80 | ±06.82 | ±04.93 | ±11.16 | ±08.42 | ±06.63 |
| **RECTOR+** | **±08.70** | **±06.70** | ±05.04 | ±11.46 | ±08.56 | ±06.77 |
| **RECTOR-L+** | ±08.82 | ±06.88 | **±04.86** | **±10.93** | **±08.29** | **±06.35** |

C.8    INTERPRETABILITY ANALYSIS

**MSIT & ECR**    Fig. 9 visualizes the top-10 cross–region attention scores learned by RECTOR on the MSIT (left) and ECR (right) sEEG tasks. And Fig. 10 illustrates the detailed region-level attention maps for MSIT (left) and ECR (right). In MSIT, we observe strong couplings most notably between hippocampus (hipp) and caudate, as well as between dorsal anterior cingulate cortex (dACC) and ventral lateral prefrontal cortex (vlPFC), reflecting the engagement of cognitive control circuits during MSIT tasks. By contrast, ECR highlights amygdala-hippocampus interactions, consistent with affect-based memory retrieval and valuation processes (Costa et al., 2022).

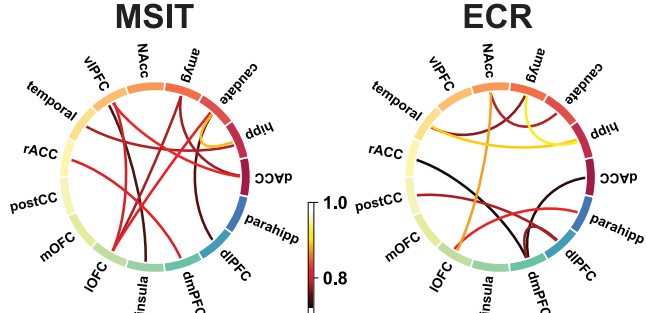

Figure 9: Top-10 cross-region attention pairs for MSIT and ECR.

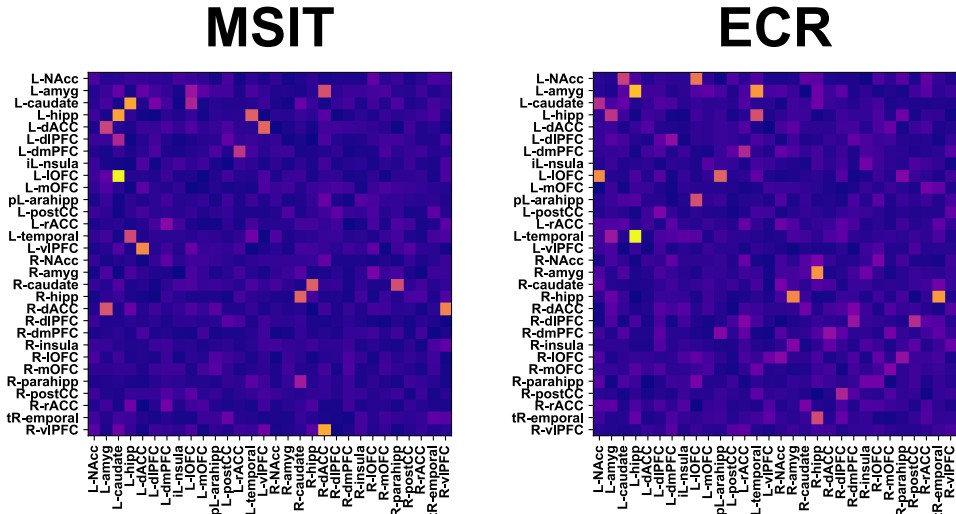

Figure 10: Region-level attention maps for MSIT and ECR.

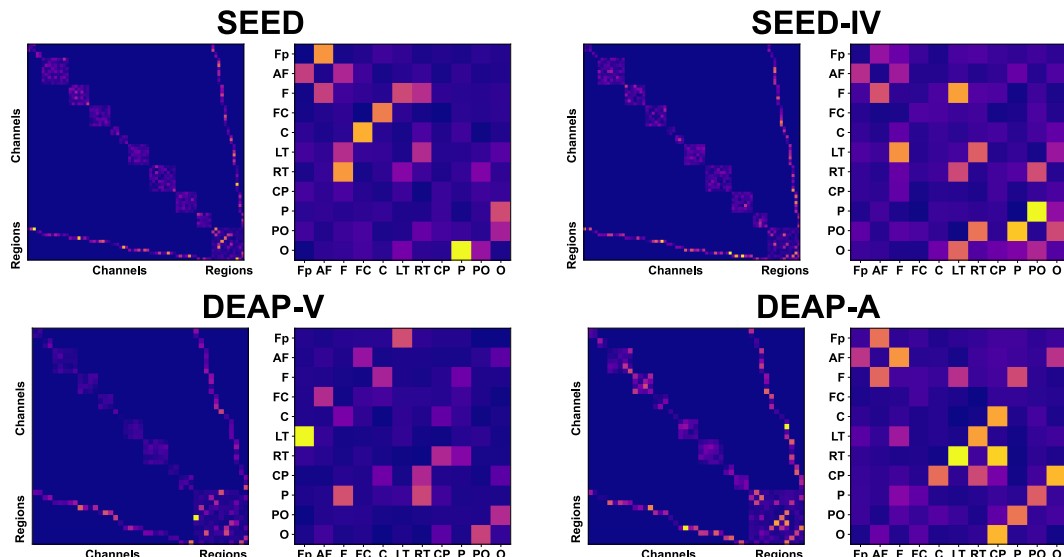

Figure 11: Full attention maps (Left) and region-level attention maps (Right) for SEED, SEED-IV, DEAP-V (Valence), and DEAP-A (Arousal).

**SEED, SEED-IV, and DEAP**    Our analysis of attention maps across SEED, SEED-IV, and DEAP confirms the core principles of RECTOR-SA's hierarchical design. Fig. 11 illustrates the full attention matrices decomposed by token type (Channels ↔ Channels, Regions ↔ Channels, Regions ↔ Regions) for four tasks. Fig. 12 visualizes the top-10 cross-region attention scores learned by RECTOR on the DEAP dataset. Both Figs. 6 and 12 demonstrate the strong fronto-temporal interactions during emotional processing, aligning with neuroscientific studies (Sun et al., 2023).

The block representing direct channel-to-channel attention is sparse across all four tasks. This indicates that RECTOR-SA successfully prunes the uninformative connections. In sharp contrast, the blocks representing attention (1) between region tokens and (2) between region and channel tokens are more visibly active. This confirms that the RECTOR-SA mechanism successfully transfers the modeling of long-range functional connectivity away from the noisy channel space and concentrates it in the high-signal region space.

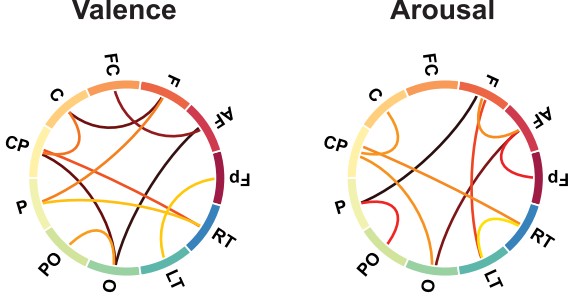

Figure 12: Top-10 cross-region attention pairs for DEAP.

# D    DATASETS

**MSIT & ECR**    The MSIT and ECR datasets (Provenza et al., 2019) probe human cognitive and emotional conflict responses using intracranial electrophysiological recordings from 17 participants with pharmaco-resistant partial seizures. Each subject performed two distinct conflict-based tasks: the Multi-Source Interference Task (MSIT), in which they identified a target number among distractors under varying congruency, and the Emotional Conflict Resolution (ECR) task, which required

resolving conflict between facial expressions and superimposed emotional words (Figure 13). Local field potentials (LFPs) were recorded via depth electrodes implanted across up to 30 anatomically defined brain regions, yielding 64–195 channels per participant. These LFP recordings were then used to classify rest-state versus task-state activity. By combining richly sampled, multi-site LFPs with behaviorally precise conflict paradigms, these datasets offer a unique window into both task-specific and generalizable neural mechanisms–insights that could guide the development of adaptive deep-brain stimulation therapies for cognitive disorders.

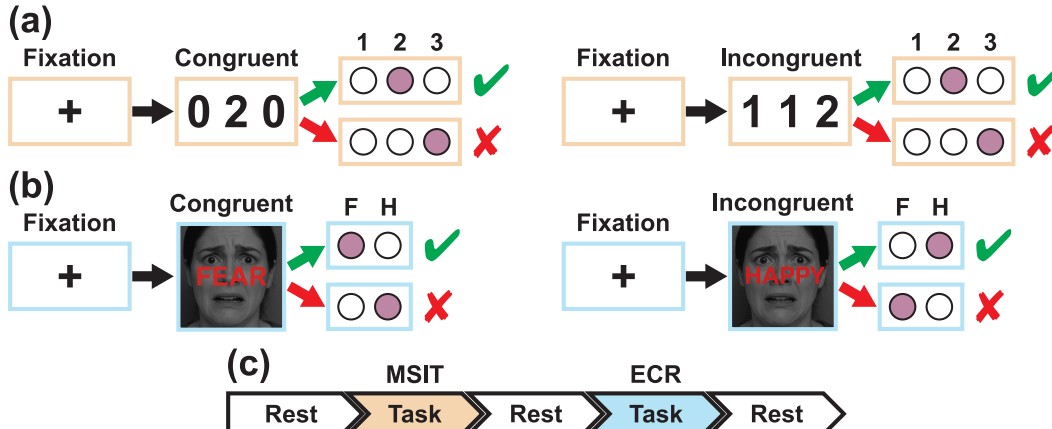

Figure 13: Behavioral paradigms for the MSIT and ECR tasks. (a) **MSIT**: Participants view three simultaneously presented digits and must press the button matching the value of the unique target digit, ignoring its position. Distractors are either zeros (congruent trial) or valid digits (incongruent trial). (b) **ECR**: Participants see an emotional word superimposed on a facial expression and must select the emotion denoted by the word, disregarding the face. Trials are congruent when word and expression match, and incongruent otherwise. (c) All trials are annotated as either rest or task (MSIT/ECR) for subsequent classification.

We adopt a subject-dependent evaluation paradigm. Each recording session is chronologically split into 80% training + validation sets (with 72% used for training and a random 8% for validation) and a 20% unshuffled hold-out test set. We insert a 4-second buffer between the training and test splits to ensure that no adjacent segments appear in both sets. Hyperparameters were chosen based on performance on this validation set. This approach prevents leakage of temporally adjacent sEEG patterns into both train and test sets. And it simulates real-world deployment, where one must train on past data and predict on future recordings (future recordings could include information from the past data and cause implicit data leakage). LFPs were recorded at 2 kHz and downsampled to 1 kHz. We applied a 1–150 Hz band-pass filter to remove low- and high-frequency noise, and notch filters at 60 Hz and its harmonics to eliminate line noise. Before training, we segment each trial into non-overlapping 4-second epochs and treat each epoch as an independent sample for classification.

**SEED & SEED-IV**    The SEED dataset (Zheng & Lu, 2015) comprises 62-channel EEG recordings from 15 participants who watched 15 Chinese film clips per session, each designed to elicit one of three emotional states (positive, neutral, negative). Each subject completed three sessions (45 trials in total). SEED-IV (Zheng et al., 2018) retains the same 62-channel, 15-subject, three-session design, but extends to four emotion classes (happy, neutral, sad, fear) with 24 trials per session (72 trials per subject).

We adopt two evaluation paradigms: (1) Subject-dependent (SD): experiments are conducted per session. For SEED, each session's 15 trials are split into the first 9 for training and the last 6 for testing, with 10% of the training samples as a validation set; for SEED-IV, each session's 24 trials are split into the first 16 for training and the remaining 8 for testing. (2) Subject independent (SI): for both SEED and SEED-IV, we perform leave-one-subject-out (LOSO) cross-validation across all subjects. For each fold of the LOSO, the remaining subjects form the full training set. This full training set is then randomly split into a 90% training partition and a 10% validation partition in a trial-wise manner. We train the model on the 90% partition and use the 10% validation partition for

hyperparameter tuning. Therefore, the validation set is "hold-out" as it is unseen by the trained model. The final model is then evaluated on the entirely unseen test subject. All data were recorded at 1 kHz and downsampled to 200 Hz, then band-pass filtered between 1–50 Hz to remove artifacts. Prior to training, each trial is divided into non-overlapping 4-second segments, and each segment is treated as an independent sample for emotion classification.

**DEAP** The DEAP dataset (Koelstra et al., 2011) captures 32-channel EEG recordings from 32 participants as they watched 40 one-minute music video excerpts (40 trials). After each clip, participants provided subjective ratings on arousal, valence, dominance, liking, and familiarity, yielding rich physiological–affective annotations for downstream modeling.

We evaluate our models under two standard protocols: (1) Subject-dependent (SD): trial-wise 10-fold cross-validation for each subject, with 10% of the training samples as a validation set in each fold. (2) Subject independent (SI): leave-one-subject-out (LOSO) cross-validation across all subjects. All EEG recordings were acquired with a 32-electrode montage at 512 Hz and downsampled to 128 Hz. A 3-second pre-trial baseline was removed from each trial, and the data were band-pass filtered between 4-45 Hz to suppress low- and high-frequency noise. For affective labels (valence and arousal), participants rated each dimension on a 1–9 scale; we binarize these into "low" ($\leq 5$) and "high" ($> 5$) classes for downstream classification.

**Trial-Wise Splitting** For our subject-dependent protocols, randomly shuffling the segments across trials before splitting into train and test sets can inadvertently place highly correlated adjacent segments into both splits, inflating measured accuracy. In real-world deployment, however, such overlap rarely occurs, which can lead to lower measured performance. To obtain a more realistic assessment, we instead split data at the trial level, ensuring that all segments from any given trial reside entirely in either the training or testing set, thereby preventing information leakage and yielding a more generalizable evaluation.

## E  ELECTRODE AND REGION MAPS OF EEG/sEEG

The electrode and region layouts for our EEG datasets are illustrated in Figure 14. In both SEED/SEED-IV and DEAP datasets, sensors are arranged according to the international 10-20 system and grouped into eleven anatomically defined regions, including prefrontal, anterior frontal, frontal, fronto-central, central, centro-parietal, parietal, parieto-occipital, occipital, left temporal, and right temporal (Alarcao & Fonseca, 2017).

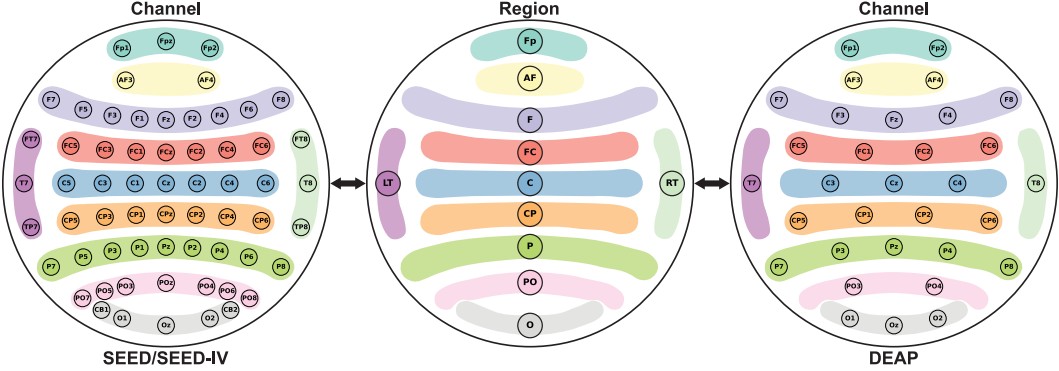

Figure 14: Electrode and region maps of EEG for SEED/SEED-IV and DEAP datasets. (Left) SEED/SEED-IV channel layout, with colored overlays indicating anatomically defined functional regions. (Center) Region-level abstraction shared across both SEED/SEED-IV and DEAP, represented in our model by an extra region token. (Right) DEAP channel layout with the same region overlays. Regions–Fp: pre-frontal; AF: anterior frontal; F: frontal; FC: fronto-central; C: central; CP: centro-parietal; P: parietal; PO: parieto-occipital; O: occipital; LT: left temporal; RT: right temporal.

For the sEEG data in MSIT and ECR datasets, each subject was implanted with bilateral depth electrodes, each comprising 8-16 contacts. Subjects had 5 to 9 electrodes in the right hemisphere, and

5 to 8 electrodes in the left hemisphere, yielding 64 to 195 bipolar-referenced channels per subject. Each channel is assigned to one of the 30 brain regions using the Electrode Labeling Algorithm (Peled et al., 2017). A detailed breakdown of channels per region is provided in Table 26.

Table 26: Number of sEEG channels per brain region across 17 subjects (Provenza et al., 2019).

| Region | Subject | | | | | | | | | | | | | | | | |
|---|---|---|---|---|---|---|---|---|---|---|---|---|---|---|---|---|---|
| | 1 | 2 | 3 | 4 | 5 | 6 | 7 | 8 | 9 | 10 | 11 | 12 | 13 | 14 | 15 | 16 | 17 |
| L-NAcc | 0 | 0 | 0 | 0 | 2 | 0 | 0 | 0 | 1 | 0 | 1 | 0 | 1 | 0 | 2 | 0 | 1 |
| L-amyg | 4 | 4 | 4 | 5 | 2 | 5 | 3 | 5 | 0 | 2 | 4 | 5 | 4 | 0 | 1 | 6 | 10 |
| L-caudate | 0 | 0 | 0 | 0 | 7 | 3 | 10 | 2 | 7 | 0 | 7 | 5 | 2 | 3 | 9 | 8 | 7 |
| L-hipp | 5 | 3 | 2 | 7 | 4 | 10 | 4 | 0 | 0 | 6 | 11 | 10 | 5 | 6 | 6 | 8 | 2 |
| L-dACC | 1 | 1 | 1 | 3 | 0 | 5 | 0 | 0 | 4 | 2 | 6 | 4 | 4 | 4 | 5 | 8 | 0 |
| L-dlPFC | 8 | 11 | 10 | 15 | 15 | 31 | 17 | 16 | 43 | 33 | 35 | 19 | 18 | 30 | 29 | 23 | 12 |
| L-dmPFC | 7 | 3 | 1 | 3 | 3 | 2 | 10 | 19 | 1 | 4 | 6 | 5 | 1 | 3 | 4 | 9 | 5 |
| L-insula | 0 | 0 | 0 | 1 | 0 | 0 | 0 | 2 | 3 | 0 | 0 | 0 | 0 | 0 | 0 | 0 | 0 |
| L-lOFC | 4 | 4 | 0 | 7 | 8 | 7 | 10 | 8 | 8 | 4 | 8 | 9 | 9 | 8 | 9 | 7 | 7 |
| L-mOFC | 2 | 1 | 2 | 2 | 1 | 3 | 4 | 1 | 1 | 3 | 1 | 0 | 0 | 1 | 0 | 1 | 0 |
| L-parahipp | 0 | 0 | 0 | 0 | 0 | 0 | 0 | 3 | 0 | 0 | 0 | 0 | 2 | 0 | 0 | 0 | 0 |
| L-postCC | 0 | 2 | 1 | 0 | 0 | 0 | 0 | 0 | 0 | 0 | 0 | 1 | 1 | 0 | 0 | 0 | 0 |
| L-rACC | 0 | 1 | 0 | 0 | 3 | 0 | 0 | 0 | 0 | 0 | 3 | 5 | 0 | 4 | 1 | 1 | 0 |
| L-temporal | 6 | 5 | 6 | 10 | 19 | 22 | 17 | 13 | 13 | 19 | 22 | 26 | 24 | 26 | 31 | 23 | 11 |
| L-vlPFC | 1 | 0 | 0 | 9 | 12 | 4 | 6 | 11 | 1 | 4 | 4 | 6 | 4 | 6 | 6 | 6 | 6 |
| R-NAcc | 0 | 0 | 0 | 0 | 3 | 0 | 0 | 0 | 0 | 0 | 0 | 2 | 0 | 1 | 0 | 0 | 0 |
| R-amyg | 4 | 3 | 1 | 5 | 5 | 0 | 0 | 0 | 2 | 5 | 5 | 0 | 2 | 6 | 0 | 3 | 5 |
| R-caudate | 0 | 0 | 1 | 8 | 6 | 0 | 0 | 0 | 0 | 0 | 6 | 1 | 1 | 9 | 7 | 0 | 7 |
| R-hipp | 2 | 2 | 5 | 3 | 7 | 3 | 0 | 1 | 6 | 11 | 7 | 15 | 7 | 13 | 5 | 6 | 7 |
| R-dACC | 0 | 0 | 2 | 1 | 1 | 6 | 1 | 0 | 2 | 2 | 7 | 4 | 6 | 1 | 2 | 3 | 0 |
| R-dlPFC | 6 | 9 | 14 | 23 | 12 | 7 | 2 | 1 | 10 | 24 | 20 | 24 | 8 | 29 | 29 | 18 | 6 |
| R-dmPFC | 7 | 5 | 3 | 12 | 3 | 1 | 10 | 0 | 1 | 8 | 0 | 2 | 0 | 9 | 6 | 5 | 10 |
| R-insula | 0 | 0 | 0 | 0 | 0 | 0 | 0 | 2 | 0 | 0 | 0 | 0 | 0 | 0 | 0 | 0 | 0 |
| R-lOFC | 5 | 6 | 3 | 8 | 8 | 2 | 15 | 13 | 3 | 10 | 7 | 8 | 7 | 3 | 8 | 6 | 6 |
| R-mOFC | 0 | 1 | 0 | 3 | 1 | 0 | 4 | 2 | 1 | 2 | 2 | 0 | 4 | 0 | 1 | 1 | 1 |
| R-parahipp | 0 | 0 | 0 | 0 | 0 | 0 | 0 | 3 | 0 | 0 | 0 | 0 | 5 | 0 | 0 | 0 | 1 |
| R-postCC | 0 | 3 | 0 | 0 | 0 | 0 | 0 | 0 | 0 | 0 | 0 | 0 | 0 | 3 | 1 | 0 | 0 |
| R-rACC | 0 | 0 | 0 | 1 | 0 | 0 | 1 | 1 | 1 | 0 | 3 | 2 | 0 | 0 | 4 | 0 | 1 |
| R-temporal | 4 | 7 | 5 | 12 | 16 | 16 | 0 | 21 | 10 | 18 | 8 | 24 | 11 | 22 | 23 | 13 | 19 |
| R-vlPFC | 3 | 0 | 3 | 12 | 3 | 12 | 8 | 15 | 2 | 5 | 14 | 8 | 4 | 8 | 6 | 8 | 6 |
| Total | 69 | 71 | 64 | 150 | 141 | 139 | 122 | 139 | 120 | 162 | 189 | 183 | 131 | 194 | 195 | 163 | 130 |

# F  IMPLEMENTATION DETAILS

## F.1  MODEL CONFIGURATIONS

RECTOR-Transformer block is used in the encoder, predictor, and decoder. The detailed configurations can be found in Table 27.

Table 27: Model configurations. RECTOR-L in (); LT: Layer Tuning.

| Parameter | Encoder | Predictor | Decoder |
|---|---|---|---|
| RECTOR-Transformer block | 8 | 2 | 4 |
| Hidden dimension | 64 (128) | 64 (128) | 64 (128) |
| Head | 4 | 4 | 4 |
| Feed-forward dimension | 256 (512) | 256 (512) | 256 (512) |
| Intermediate layers for LT | 3 | | |

## F.2  POSITIONAL EMBEDDINGS

To better capture both temporal dependency and spatial topology, we incorporate Rotary Position Embeddings (RoPE) on the temporal dimension (Su et al., 2024), and graph-Laplacian eigenvector embeddings on the spatial axis (Dwivedi & Bresson, 2020). Concretely, we construct a spatial graph, where channels within each region and regions themselves form fully connected subgraphs, and region-channel edges are built as in our Statically Structured Attention mask in Section 2.3. Then we compute its Laplacian eigenvectors to serve as fixed positional encodings for every region and channel

token. This hybrid scheme endows the model with rotary awareness over time while respecting the brain's anatomical structure.

## F.3 REGION CONDITIONING

EEG and sEEG data often involve high channel counts and long temporal sequences, making per-channel or per-time conditioning prohibitively expensive due to the dramatically increased token length. Therefore, in region conditioning, our approach attaches each *unique* region positional embedding from context or target blocks *only once*, rather than replicating it across channels or time steps. This lightweight conditioning adds minimal overhead, enabling faster training and inference while still capturing essential distributed spatial dependencies.

## F.4 INTERPRETABILITY MAPS

We computed Grad-CAM (Selvaraju et al., 2017) saliency heatmap for each emotion class. Specifically, after passing an input trial through the model, we apply Grad-CAM to the output layer (before cross-attention heads) to produce a per-channel/region importance score indicating how strongly each input location contributed to the predicted class. We extract self-attention scores from RECTOR-SA. For each attention head, we compute the mean attention score between every pair of region tokens across all time segments. We then average them over attention heads and rank these region-to-region scores and report the ten highest-scoring pairs as those regions whose interactions most strongly influenced the model's decision. All maps and attention summaries are averaged across subjects within each dataset.

## F.5 PRE-TRAINING ON FOUNDATION MODELS

For neural foundation models in this work, we used publicly available checkpoints and then pre-trained the model on the target dataset with the same number of iterations as RECTOR.

## F.6 HYPERPARAMETER SETTINGS

Table 28: Pre-training settings.

| Hyperparemeter | Setting |
|---|---|
| Epochs | 200 |
| Warmup epochs | 40 |
| Batch size | 256 |
| Dropout | 0.3 |
| Optimizer | AdamW |
| $\beta$ | (0.9, 0.999) |
| Scheduler | Cosine Annealing Scheduler |
| Learning rate | 5e-4 |
| Minimal learning rate | 1e-6 |
| Weight decay | 5e-2 |
| EMA momentum schedule | Linear scheduler |
| EMA start momentum | 0.9 |
| EMA final momentum | 1.0 |
| Context mask ratio | 0.75 |
| Target mask ratio | 0.9/0.9/0.9/0.9/0.9 |
| Patch size | 0.5 second |

## F.7 PRE-TRAINING TIME

The pre-training time for RECTOR, RECTOR+, and RECTOR-L+ on various datasets is reported in Table 30.

Table 29: Fine-tuning settings.

| Hyperparemeter | Setting |
|---|---|
| Epochs | 50 |
| Warmup epochs | 10 |
| Batch size | 256 |
| Dropout | 0.3 |
| Optimizer | AdamW |
| $\beta$ | (0.9, 0.999) |
| Scheduler | Cosine Annealing Scheduler |
| Learning rate | 5e-4 |
| Minimal learning rate | 1e-6 |
| Weight decay | 5e-2 |
| Label smoothing | 0.1 |

Table 30: RECTOR's pre-training time on various datasets.

| Model | Dataset | Pre-Training Time |
|---|---|---|
| RECTOR | SEED | 56 mins |
| | SEED-IV | 54 mins |
| | DEAP | 28 mins |
| | MSIT | 52 mins |
| | ECR | 53 mins |
| RECTOR+ | SEED+SEED-IV+DEAP+CHB-MIT | 10.3 hours |
| | MSIT+ECR | 1.8 hours |
| RECTOR-L+ | SEED+SEED-IV+DEAP+CHB-MIT | 11 hours |
| | MSIT+ECR | 1.9 hours |

## F.8 MODEL SIZE

We provide a detailed model size comparison for all baselines and RECTOR variants for EEG and sEEG applications Tab. 31. Our RECTOR and RECTOR-L are in the medium range in terms of the model size. This analysis confirms that RECTOR achieves state-of-the-art performance primarily through architectural innovation, rather than scaling of parameters alone.

Table 31: Model size for baselines and RECTOR for EEG and sEEG datasets.

| Model | EEG Model Size | sEEG Model Size |
|---|---|---|
| TSception | 25K | 923K |
| EEG Conformer | 281K | 4.6M |
| LGGNet | 166K | 15.2M |
| PGCN | 12.3M | |
| MASA-TCN | 824K | |
| EmT | 704K | |
| MMM | 40K | |
| LaBraM | 5.8M | |
| REmoNet | 440K | |
| CBraMod | 4.0M | |
| PopT | 20M | 20M |
| Seegnificant | | 789K |
| BrainBERT | | 43.6M |
| Brant | | 500M |
| Du-IN | | 4.4M |
| MAE | 311K | 345K |
| JEPA | 311K | 345K |
| CAE | 311K | 345K |
| RECTOR | 472K | 537K |
| RECTOR-L | 1.8M | 2.0M |

## F.9 CHANNEL MATCHING FOR EXPANDED PRE-TRAINING

To accommodate differing channel montages during expanded pre-training, we employ two strategies:

1. SEED/SEED-IV → DEAP (62 → 32 channels): We restrict the encoder to use the 32 channels common to both datasets, preserving a consistent 32-channel architecture throughout pre-training and fine-tuning.

2. DEAP → SEED/SEED-IV (32 → 62 channels): We use a 62-channel encoder augmented with 30 null tokens. An attention mask prevents any interaction with these null tokens during pre-training on DEAP. When pre-training returns to SEED/SEED-IV, we remove the mask and activate all 62 channels for full token mixing.

### F.10 COMPUTE RESOURCES

All experiments were conducted using Python 3.8.16 and PyTorch 2.0.1 with CUDA 11.7 for GPU acceleration. The primary training environment ran on Red Hat Enterprise Linux 7.9 with AMD Ryzen Threadripper PRO 5995WX 64-Cores CPU, and $2 \times$ NVIDIA GeForce RTX 4090 24 GB.

### F.11 PSEUDOCODE

We decompose RECTOR into two main stages and present pseudocode for each: self-supervised pre-training in Algorithm 1 and downstream fine-tuning in Algorithm 2. Together, these algorithms illustrate the end-to-end training pipeline of RECTOR.

## G EVALUATION METRICS

We evaluated the performance of RECTOR and all baselines using the Area Under the Receiver Operating Characteristic Curve (AUROC) for task engagement classification on MSIT and ECR, and the weighted F1-Score for emotion recognition on SEED, SEED-IV, and DEAP. We also used balanced accuracy and Cohen's kappa for emotion recognition in the Appendix. In the subject-dependent setting, results are averaged per subject in MSIT, ECR, and DEAP, and across subjects and sessions in SEED and SEED-IV. In the subject-independent setting, results are averaged across subjects for every dataset. All metrics are reported as Mean$_{\pm \text{Std}}$.

We indicated the statistical significance for classification performance using a Wilcoxon signed-rank test ($p < 0.05$). The test was applied across subjects for MSIT, ECR, and DEAP, and across both subjects and sessions for SEED and SEED-IV.

## H LIMITATIONS

While RECTOR establishes a powerful end-to-end region–channel–temporal modeling framework, several limitations merit attention. First, our anatomical parcellation and region–channel assignments are defined a priori; extending RECTOR to alternative atlases or subject-specific electrode layouts with distinct region-channel structure will require additional adaptation. Second, even though we sparsify the hierarchical self-attention mechanism through dynamic gating, together with multi-view masking, it introduces extra computational and memory overhead, which could challenge real-time or resource-constrained deployments. Third, although our interpretability analyses (CAMs and attention scores) align with established neurophysiological findings, they remain correlational; rigorous validation against ground-truth neurobiological markers is needed to confirm causal relevance.

Despite these limitations, RECTOR still represents a major advance in self-supervised neural representation learning, delivering anatomically grounded embeddings that are both robust and interpretable. We look into more lightweight attention schemes and adaptive parcellations to reduce computational overhead and accommodate alternative atlases; pursue multimodal extensions to fuse complementary biosignals; and scale pre-training across larger, heterogeneous repositories to further bolster generalization and translational impact.

---

**Algorithm 1** Self-Supervised Pre-training of RECTOR

---

**Input:** Input batch $\mathbf{X} = \left[\mathbf{X}^{\mathrm{C}}; \mathbf{X}^{\mathrm{R}}\right]$, context encoder $f(\cdot\,; \theta)$, target encoder $\tilde{f}(\cdot\,; \tilde{\theta})$, predictor $g(\cdot\,; \psi)$, decoder $h(\cdot\,; \phi)$, context range ratios $\rho_{\mathbf{c}}, \eta_{\mathbf{c}}$, target range ratios $\rho_{\mathbf{t}^{(m)}}, \eta_{\mathbf{t}^{(m)}}$ for $m \in \mathcal{M}$, learning rate $\alpha$, EMA rate $\beta$

**Output:** Updated encoder $f(\cdot\,; \theta)$

1: **for** each $\mathbf{X}$ **do**
2:                                                       ▷ (a) Sample context & target blocks
3:     $\mathbf{M}_{\mathbf{c}}, \{\mathbf{M}_{\mathbf{t}^{(m)}}\}_{m \in \mathcal{M}} \leftarrow \texttt{RECTOR-Mask}(\mathbf{X}, \rho_{\mathbf{c}}, \eta_{\mathbf{c}}, \{\rho_{\mathbf{t}^{(m)}}\}_{m \in \mathcal{M}}, \{\eta_{\mathbf{t}^{(m)}}\}_{m \in \mathcal{M}})$
4:     $\mathbf{X}_{\mathbf{c}} \leftarrow \mathbf{X} \odot \mathbf{M}_{\mathbf{c}}$
5:     **for all** $m \in \mathcal{M}$ **do**
6:         $\mathbf{X}_{\mathbf{t}^{(m)}} \leftarrow \mathbf{X} \odot \mathbf{M}_{\mathbf{t}^{(m)}}$
7:     **end for**
8:                                            ▷ (b) Compute positional embeddings
9:     $\mathbf{P}_{\mathbf{c}}, \{\mathbf{P}_{\mathbf{t}^{(m)}}\}_{m \in \mathcal{M}}, \mathbf{P} \leftarrow \texttt{PositionalEmbedding}(\mathbf{X}_{\mathbf{c}}, \{\mathbf{X}_{\mathbf{t}^{(m)}}\}_{m \in \mathcal{M}})$
10:                           ▷ (c) Compute and concatenate conditioning tokens
11:     $\mathbf{P}_{\mathbf{c}}^{\mathrm{Cd}}, \{\mathbf{P}_{\mathbf{t}^{(m)}}^{\mathrm{Cd}}\}_{m \in \mathcal{M}} \leftarrow \texttt{RegionalEmbedding}(\mathbf{X}_{\mathbf{c}}, \{\mathbf{X}_{\mathbf{t}^{(m)}}\}_{m \in \mathcal{M}})$
12:     $\mathbf{X}_{\mathbf{c}} \leftarrow \left[\mathbf{X}_{\mathbf{c}}; \{\mathbf{P}_{\mathbf{t}^{(m)}}^{\mathrm{Cd}}\}_{m \in \mathcal{M}}\right]$
13:     $\mathbf{X} \leftarrow \left[\mathbf{X}; \mathbf{P}_{\mathbf{c}}^{\mathrm{Cd}}\right]$
14:                                                ▷ (d) Encode context & targets
15:     $\mathbf{Z}_{\mathbf{c}} \leftarrow f(\mathbf{X}_{\mathbf{c}}, \mathbf{P}_{\mathbf{c}}; \theta)$
16:     **for all** $m \in \mathcal{M}$ **do**
17:         $\mathbf{Z}_{\mathbf{t}^{(m)}} \leftarrow \tilde{f}(\mathbf{X}, \mathbf{P}; \tilde{\theta}) \odot \mathbf{M}_{\mathbf{t}^{(m)}}$
18:     **end for**
19:                                                    ▷ (e) Predict & reconstruct
20:     **for all** $m \in \mathcal{M}$ **do**
21:         $\hat{\mathbf{Z}}_{\mathbf{t}^{(m)}} \leftarrow g(\mathbf{Z}_{\mathbf{c}}, \mathbf{P}_{\mathbf{t}^{(m)}}; \psi)$
22:         $\hat{\mathbf{X}}_{\mathbf{t}^{(m)}} \leftarrow h(\hat{\mathbf{Z}}_{\mathbf{t}^{(m)}}, \mathbf{P}_{\mathbf{t}^{(m)}}; \phi)$
23:     **end for**
24:                                                         ▷ (f) Compute losses
25:     $\mathcal{L}_{\mathrm{input}}, \mathcal{L}_{\mathrm{rep}}, \mathcal{L}_{\mathrm{C}} \leftarrow 0, 0, 0$
26:     **for all** $m \in \mathcal{M}$ **do**
27:         $\mathcal{L}_{\mathrm{input}} \leftarrow \mathcal{L}_{\mathrm{input}} + \texttt{MSE}(\hat{\mathbf{X}}_{\mathbf{t}^{(m)}}, \mathbf{X}_{\mathbf{t}^{(m)}})$
28:         $\mathcal{L}_{\mathrm{rep}} \leftarrow \mathcal{L}_{\mathrm{rep}} + \texttt{MSE}(\hat{\mathbf{Z}}_{\mathbf{t}^{(m)}}, \texttt{SG}(\mathbf{Z}_{\mathbf{t}^{(m)}}))$
29:         **for all** $m' \in \mathcal{M} \setminus \{m\}$ **do**
30:             $\mathcal{L}_{\mathrm{C}} \leftarrow \mathcal{L}_{\mathrm{C}} + \texttt{MSE}(\hat{\mathbf{Z}}_{\mathbf{t}^{(m)}}, \hat{\mathbf{Z}}_{\mathbf{t}^{(m')}})$
31:         **end for**
32:     **end for**
33:     $\mathcal{L}_{\mathrm{VCReg}} \leftarrow \texttt{VCReg}(\mathbf{Z}_{\mathbf{c}})$
34:     $\mathcal{L}_{\mathrm{RCReg}} \leftarrow \texttt{RCReg}(\mathbf{Z}_{\mathbf{c}})$
35:     $\mathcal{L} \leftarrow \mathcal{L}_{\mathrm{input}} + \mathcal{L}_{\mathrm{rep}} + \mathcal{L}_{\mathrm{C}} + \mathcal{L}_{\mathrm{RCReg}} + \mathcal{L}_{\mathrm{VCReg}}$
36:                                                    ▷ (g) Backprop & update
37:     $\theta \leftarrow \theta - \alpha \nabla_{\theta} \mathcal{L}, \quad \psi \leftarrow \psi - \alpha \nabla_{\psi} \mathcal{L}, \quad \phi \leftarrow \phi - \alpha \nabla_{\phi} \mathcal{L}$
38:     $\tilde{\theta} \leftarrow \beta \tilde{\theta} + (1 - \beta) \theta$
39: **end for**

---

# I   BROADER IMPACTS

RECTOR pushes the frontier of self-supervised neural representation learning for more objective, data-driven biomarkers of affective and cognitive disorders. By jointly modeling region-, channel-, and temporal-level dynamics, RECTOR could improve the precision of both non-invasive (EEG) and invasive (sEEG) diagnostics, enabling earlier detection and more personalized interventions in neurocognitive care. Its class activation maps and attention scores provide interpretable insights into distributed brain activity, which can foster trust with clinicians and further guide neuroscientific research. Future work should expand RECTOR to larger, more diverse cohorts and integrate multimodal data to ensure robustness and real-world clinical utility.

---

**Algorithm 2** Fine-Tuning RECTOR with RECTOR-LT

---

**Input:**    Input batch $(\mathbf{X}, y)$, $\mathbf{X} = \left[\mathbf{X}^{\mathrm{C}}; \mathbf{X}^{\mathrm{R}}\right]$, pre-trained encoder $f(\cdot)$, intermediate layers of the
        encoder $L_{\mathrm{I}}$, query tokens $\{\mathbf{q}^{(u)}\}_{(u) \in \{\mathrm{R,C,T}\}}$, cross attention heads
        $\{\texttt{CrossAttn}^{(u)}(\cdot)\}_{(u) \in \{\mathrm{R,C,T}\}}$, prediction head $\mathbf{W}_{\mathrm{out}}$, learning rate $\alpha$

**Output:** Updated  encoder  $f(\cdot)$,  query  tokens  $\{\mathbf{q}^{(u)}\}_{(u) \in \{\mathrm{R,C,T}\}}$,  cross  attention  heads
     $\{\texttt{CrossAttn}^{(u)}(\cdot)\}_{(u) \in \{\mathrm{R,C,T}\}}$, prediction head $\mathbf{W}_{\mathrm{out}}$

 1: **for** each $(\mathbf{X}, y)$ **do**
 2:                                                              ▷ (a) Compute positional embeddings
 3:     $\mathbf{P} \leftarrow \texttt{PositionalEmbedding}(\mathbf{X})$
 4:                                                ▷ (b) Compute and concatenate conditioning tokens
 5:     $\mathbf{P}^{\mathrm{Cd}} \leftarrow \texttt{RegionalEmbedding}(\mathbf{X})$
 6:     $\mathbf{X} \leftarrow \left[\mathbf{X}; \mathbf{P}^{\mathrm{Cd}}\right]$
 7:                                                      ▷ (c) Extract & pool intermediate features
 8:     **for** $\ell \in L_{\mathrm{I}}$ **do**
 9:         $\mathbf{Z}_{(\ell)} \leftarrow f_{(\ell)}(\mathbf{X}, \mathbf{P})$
10:         $\mathbf{Z}_{(\ell)}^{\mathrm{R}}, \mathbf{Z}_{(\ell)}^{\mathrm{C}}, \mathbf{Z}_{(\ell)}^{\mathrm{T}} \leftarrow \texttt{AvgPool}(\mathbf{Z}_{(\ell)})$
11:     **end for**
12:                                                           ▷ (d) Cross-attention aggregation
13:     **for** $u \in \{\mathrm{R, C, T}\}$ **do**
14:         $h^{(u)} \leftarrow \texttt{CrossAttn}^{(u)}\big(\mathbf{q}^{(u)}, \left[\mathbf{Z}_{(\ell)}^{(u)}\right]_{\ell \in L_{\mathrm{I}}}, \left[\mathbf{Z}_{(\ell)}^{(u)}\right]_{\ell \in L_{\mathrm{I}}}\big)$
15:     **end for**
16:     $h \leftarrow [h^{\mathrm{R}}; h^{\mathrm{C}}; h^{\mathrm{T}}] \mathbf{W}_{\mathrm{out}}$
17:                                                        ▷ (e) Compute loss, backprop & update
18:     $\mathcal{L} \leftarrow \texttt{CrossEntropy}(h, y)$
19:     Update all weights via $-\alpha \nabla \mathcal{L}$
20: **end for**

---

