# OpenReview forum: "RECTOR: Masked Region-Channel-Temporal Modeling for Cognitive Representation Learning"
_ICLR.cc/2026/Conference — Submitted to ICLR 2026_

### Official Review · Reviewer_v9Zs · 2025-10-27

**Soundness:** 3
**Presentation:** 1
**Contribution:** 2
**Rating:** 4
**Confidence:** 4

**Summary:**

This paper proposes RECTOR, a self-supervised framework for EEG and sEEG cognitive representation learning that explicitly models region–channel–temporal interactions. The key contributions include: (1) RECTOR-SA, a hierarchical sparse attention mechanism incorporating anatomical priors and dynamic gating; (2) RECTOR-Mask, a structured multi-view masking strategy that creates region- and time-aware masked modeling targets; (3) NC²-MM, a unified learning objective that combines masked modeling and contrastive learning within one architecture; and (4) RCReg, a specialized regularization for improving region–channel token representations. The model achieves state-of-the-art results across EEG emotion recognition and sEEG task-engagement classification benchmarks, with supporting ablations and interpretability analyses.

**Strengths:**

1. Ambitious attempt to integrate spatial priors and self-supervision in neural signal modeling.

2. Structured masking and hierarchical attention are intuitively motivated.

3. Experimental results are comprehensive, covering multiple datasets, protocols, and baselines, including ablations that validate each core component of the architecture.

4. The method provides neuroscientifically interpretable results at both region and channel levels, demonstrating alignment with known physiological patterns.

**Weaknesses:**

1.The figures in the manuscript need improvement, especially Figures 2 and 3. Figure 4 is significantly clearer in comparison.

2.The method’s novelty appears incremental rather than fundamental. Most components (structured masking, region tokens, gated attention, variance/covariance regularization) are adaptations of well-known techniques with domain-specific adjustments rather than a distinctly new contribution.

3.The writing is dense, and the paper tends to overstate its contributions relative to the demonstrated novelty.

4.The anatomical prior design is under-justified. Region partitioning is treated as fixed and universally correct, but inter-subject anatomical variability is substantial in EEG/sEEG. The paper does not assess the robustness or validity of this assumption.

5.Pretraining only on each target dataset weakens the claim of general-purpose self-supervised learning.

6.Critical methodological details are placed in the appendices and should be included in the main paper to ensure clarity and reproducibility.

7.Considering the complexity of the proposed method, the absence of released code makes reproducibility difficult.

**Questions:**

1.Could Figures 2 and 3 be redesigned with improved layout and clearer color schemes to enhance readability?

2.What is the key novel contribution beyond combining existing components such as structured masking and hierarchical attention?

3.How do you justify the strength of your claims relative to the demonstrated novelty?

4.How robust is the anatomical prior (fixed region partitioning) to inter-subject variability in EEG/sEEG?

5.How does pretraining only on each target dataset support claims of general SSL generalization?

6.Can you move critical methodological details from the appendix into the main text to improve clarity and reproducibility?

7.How do you conduct the leave-one-subject-out (LOSO) evaluation? Is there a hold-out validation set used to determine the number of training epochs and hyperparameters?

8.Will the code and pretrained models be released to ensure reproducibility given the complexity of the method?

**Details Of Ethics Concerns:**

N.A.

---

> ### Author Response · Authors · 2025-11-22
> **Response to Reviewer v9Zs**
>
> ## **1. Novelty and Contribution (Re W2, W3, Q2, Q3) (1/2)**
>
> > What is the key novel contribution beyond combining existing components such as structured masking and hierarchical attention? How do you justify the strength of your claims relative to the demonstrated novelty?
>
> We thank the reviewer for this question, as it allows us to clarify the fundamental novelty of our work. The reviewer's framing of our components as "adaptations of well-known techniques" is a perspective we respectfully disagree with, as **it overlooks the fact that these components are novel (not identical) solutions to novel, domain-specific problems that do not exist in fields like Computer Vision and EEG/sEEG applications.**
>
> **The "fundamental new direction" of our work is the introduction of a holistic, end-to-end self-supervised paradigm specifically tailored for region-channel-temporal EEG/sEEG modeling. Achieving this required the invention of specialized, co-designed components, including RECTOR-SA, RECTOR-Mask, $\textbf{NC}^\textbf{2}\textbf{-MM}$, and RCReg.** The reviewer's concern about RECTOR-SA (hierarchical attention, gated attention, region token), RECTOR-Mask (structured masking), and RCReg (variance/covariance regularization) is addressed below:
>
> ### (1) **RECTOR-SA**
> (1a) **Hierarchical Attention**
> * **Generic Hierarchical Attention is about pooling and reducing complexity.** Swin Transformer [1] starts with small patches, attends locally, and then merges them into larger patches and attends again. The goal is to efficiently create a high-level representation. Variants of Hierarchical Attention Transformer [2] use a separate segment-wise encoder for local attention and a cross-segment encoder for global attention to reduce computational complexity.
> * While beyond reducing computational cost in spatio-temporal attention, **RECTOR-SA is fundamentally different, as its goal is to disentangle representations into two distinct, parallel streams: region tokens (for region-common, high-level, shared dynamics) and channel tokens (for channel-specific, fine-grained, local dynamics).** This disentanglement is a novel objective that generic hierarchical attention does not have, and it's what allows our RCReg loss to work.
>
> (1b) **Gated Attention**
>
> **The vast majority of "gated" or "sparse" transformer literature falls into three categories, all of which are different from RECTOR's gated attention**:
> * **FFN Gating** (e.g., SwiGLU [3]), which filters features, not connections in attention. We also use this in our FFN (see Appendix B.8).
> * **Routing Gating** (e.g., Mixture-of-Experts [4]), which selects FFNs, not attention partners.
> * **Fixed-Pattern Sparsity** (e.g., Sparse Transformer [5], Longformer [6]), which use rigid, pre-defined patterns (like windows) to improve speed
> * Our RECTOR-SA introduces a fourth, new paradigm. **We use a dynamic, data-driven top-$p$ gate to prune the attention matrix itself.** It has a novel justification: **it is not for speed, but for robustness. It acts as a learned denoiser, pruning spurious, noisy connections and focusing on the real signals between EEG/sEEG channels.** The top-$p$ allows the model to adapt its own sparsity to match the complexity of the brain state, a flexibility that top-$k$ (dynamic but not adaptive) or fixed-pattern methods lack."
>
> ### **(2) RECTOR-Mask**
> **RECTOR-Mask** is not an "adaptation." It's **a novel, co-designed component invented to solve a new, unsolved problem: how to create a multi-view, semantically-aware pretext task on a hierarchical region-channel-temporal representation.**
>
> * The problem of "how to do structured masking for a hierarchical, region-channel-temporal representation" was an unsolved problem. RECTOR-Mask is the first and novel solution to this problem. Our ablations (Tables 3, 4, 11, 12) prove this: Random Mask and Block Mask fail on this complex space, confirming our novel design is essential.
> * As shown in Figure 4a, **RECTOR-Mask** doesn't only create a target; it **generates five distinct, semantically different "views"** (e.g., cross-region, cross-channel-time). This is a required innovation for our $\text{NC}^2\text{-MM}$ objective, which uses representation alignment and input reconstruction to learn **predictive** representations and uses contrastive loss to learn **consistent representations between these five different views**. This hybrid goal does not exist in other masking methods.
> * **It creates a much more challenging and diverse pretext task.** Predicting a cross-time block (easier) teaches the model something different than predicting a cross-region-time block (harder). By sampling from all five domains, **RECTOR-Mask forces the model to learn richer and more robust neural dynamics than any other structured masking could.**
> * **To our knowledge, there are no other works that use a hierarchical, multi-view, spatio-temporal structured masking strategy like RECTOR-Mask.**

---

> ### Author Response · Authors · 2025-11-22
> **Response to Reviewer v9Zs**
>
> ## **1. Novelty and Contribution (Re W2, W3, Q2, Q3) (2/2)**
>
> ### **(3) RCReg and Region Token**
>
> Both RCReg and Region Token are fundamentally different from the other works because they are co-designed to solve a new problem that prior models did not even attempt: **hierarchical disentanglement.**
>
> (3a) **Region Token** (In the paper, we didn't list this as our contribution, but here we clarify as requested): **RECTOR is the first to model both the Region Token and its constituent Channel Tokens in parallel.** The Region Token is not a pooled replacement of Channel Tokens as in other work; it is a parallel, learnable hub designed to capture only the shared dynamics. All fine-grained channel-specific information is preserved in the Channel Tokens. **The goal of using Region Tokens in RECTOR is for hierarchical disentanglement.**
>
>
>
> (3b) RCReg
>
> **Other variance/covariance regularization (VCReg) and our RCReg operate on different dimensions to solve different problems.**
> * **VCReg** operates on the feature dimension to **prevent dimensional collapse**. **We also used this as mentioned in Appendix B.4** but we didn't list it as our major contribution.
> * **RCReg operates on the token dimension to solve a new, complex problem: hierarchical representational mixing.** It asks: "Are the channel tokens within a region redundant?" and more importantly, "Is the region token just copying information that its channel tokens already represent?" RCReg directly penalizes the cross-covariance between the region token and its channel tokens. This **forces the Region Token to learn only the shared, high-level information, while the Channel Tokens are free to learn their specific, local dynamics. No other regularization term is designed to do this.**
>
> **These fundamental design differences are what enable RCReg's novelty, and they prove our entire paradigm is a holistic, co-designed, and novel system.**

---

> ### Author Response · Authors · 2025-11-22
> **Response to Reviewer v9Zs**
>
> ## **2. Anatomical Priors and Parcellations (Re W4, Q4)**
>
> > Region partitioning is treated as fixed and universally correct, but inter-subject anatomical variability is substantial in EEG/sEEG, ..., How robust is the anatomical prior (fixed region partitioning) to inter-subject variability in EEG/sEEG?
>
> We thank the reviewer for this insightful question, as it directly addresses the validity of our anatomical prior in the face of real-world anatomical variance. **Our paper provides direct experimental proof that our approach is highly robust to this exact variability.**
>
> (1) **"Soft Bias" with "Dynamic Weights"**
>
> **Our design is robust because the prior is a "soft bias," not a "hard constraint."** As detailed in Section 2.3 (Figure 3), the static "Anatomical Attention" mask only guides the model, **but the values of this attention are dynamic and learned from the input. This provides a strong, neuro-anatomically grounded inductive bias (by forbidding nonsensical connections) while retaining the flexibility of dynamic, data-driven attention of each individual subject.**
>
> (2) **sEEG (MSIT/ECR) Datasets**
>
> **Our sEEG evaluation is a direct and rigorous test of the inter-subject variability.** While the anatomical atlas (DKT) [7] is fixed, **the physical implementation (electrode implantation) is unique for every subject**. As shown in Appendix E, Table 21, the electrode implantation and electrode-to-region mapping are different for every subject. **Therefore, our model must handle massive inter-subject anatomical variability to function at all on this dataset.** RECTOR's state-of-the-art performance on MSIT/ECR (Table 2) is the strongest possible proof that our architecture is robust to this variability.
>
> (3) **EEG Supporting Evidence**
>
> **This inter-subject robustness is further strongly confirmed by our SOTA performance in the Subject-Independent (SI) leave-one-subject-out EEG setting** (Tables 1, 7, etc.), which also requires generalizing the model to a new, unseen subject. If the anatomical prior were brittle and not robust to inter-subject variability, the SI performance would fail. The fact that it succeeds and is SOTA proves that the RECTOR-SA design (with current parcellation) is a highly effective and robust way to generalize across subjects despite their anatomical differences.
>
> (4) **Robustness to Parcellation**
>
> Finally, **we conducted a sensitivity analysis to assess RECTOR's robustness to different anatomical parcellation schemes on the SEED, SEED-IV, and DEAP datasets under subject-independent settings.**
> * In addition to our 11-region baseline, we evaluated performance using a 9-region hemispheric scheme [8] and a high-resolution 17-region parcellation scheme [9], as shown in Fig. 8 of the revised manuscript.
> * The results in Table R8 (added to Appendix C.5, Table 22 of the revised manuscript) demonstrate that **RECTOR is highly robust to these inter-subject variations by incorporating different anatomical priors**. Minor performance degradation was observed when using the dense 17-region parcellation scheme on the DEAP (32-channel) dataset. We attribute this slight drop to the low channel density of the input. Extracting 17 distinct region-common but not channel-specific features from only 32 channels is very difficult, and it significantly reduces the benefits inherent in our hierarchical structure. **It confirms that RECTOR effectively learns the fundamental hierarchical nature of the data, and various fixed parcellations consistently show strong robustness to inter-subject variability in EEG/sEEG**
>
> **This evidence all confirms that our approach is not brittle, but is a robust and effective way to model hierarchical brain data despite inter-subject variability.**
>
> **Table R8: Classification performance on SEED, SEED-IV, and DEAP with various parcellation schemes under subject-independent (SI) settings** (Weighted F1 Score (\%)). **BOLD**: the best performance. * denotes it is significantly better than the second-best model.**
>
> | Model                  |       SEED:SI        |      SEED-IV:SI      |   DEAP-Valence:SI    |   DEAP-Arousal:SI    |
> |:---------------------- |:--------------------:|:--------------------:|:--------------------:|:--------------------:|
> | 9-Region               |  **60.85** ± 09.22   |    43.87 ± 06.83     | 65.80 ± 11.43 | 67.53 ± 13.32 |
> | 17-Region              |    59.10 ± 09.62     |  **44.89** ± 06.72   |    64.26 ± 12.70     |    65.97 ± 14.46     |
> | 11-Region (**RECTOR**) | 60.58± 09.12 |44.79 ± 06.56 |  **66.22** ± 11.33   |  **68.01** ± 12.72   |

---

> ### Author Response · Authors · 2025-11-22
> **Response to Reviewer v9Zs**
>
> ## **3. Pre-training Schemes (Re W5, Q5)**
>
> > How does pretraining only on each target dataset support claims of general SSL generalization?
>
> We thank the reviewer for this question, as it touches on the core distinction between a general-purpose paradigm and a foundation model.
>
> (1) **Our primary contribution is a novel SSL paradigm for EEG/sEEG**. To scientifically validate its effectiveness, we first followed **a standard and rigorous evaluation protocol**: pre-training and fine-tuning on the same target dataset. **This is a common practice in SSL literature** (e.g., MAE [10] in computer vision and COMET [11] in medical time-series including EEG),  as it provides a direct comparison against other supervised and self-supervised baselines, **isolating the benefit of the SSL task itself from the usage of extra datasets.**
>
> (2) We did not stop at this baseline. **To directly address the claim of general-purpose generalization, we performed the exact experiments the reviewer suggested.** As detailed in Section 3.4, our RECTOR+ and RECTOR-L+ models were pre-trained on large, expanded, multi-site datasets (e.g., combining SEED, SEED-IV, DEAP, and CHB-MIT). The results in Tables 5, 6, 9, and 10 show that RECTOR's performance consistently improves with this expanded pre-training. **This proves that our paradigm is not limited to single datasets but effectively learns generalizable representations from diverse, combined data, directly supporting its utility for general SSL.**
>
> Thus, our paper provides evidence for both: **(1) the specific effectiveness of the RECTOR paradigm on target data (the standard comparison)** and **(2) its generalization when used in the broader, multi-site pre-training context the reviewer is asking about.**
>
> ---
>
> ## **4. Figures (Re W1, Q1)**
>
> > Could Figures 2 and 3 be redesigned with improved layout and clearer color schemes to enhance readability?
>
> We thank the reviewer for these constructive suggestions to improve the clarity of our diagrams. We have extensively redesigned Figs. 2 and 3 in the revised manuscript as requested:
>
> (1) Fig. 2: RECTOR Architecture Overview
> * **New Panel Structure**: The figure now comprises three panels: (a) SSL pipeline, (b) RECTOR-Transformer block, and a newly added (c) Fine-tuning pipeline. Panel (c) explicitly illustrates the transition from pre-training to specific downstream tasks.
> * **Visual Simplification**: To streamline the data flow in Panel (a), we removed non-essential variables (e.g., positional embeddings) and added a dedicated EEG/sEEG input block to clarify input structure.
> * **Color Consistency**: We harmonized the color scheme with the text in Section 2.2. We now exclusively highlight the four core contributions: RECTOR-SA, RECTOR-Mask, $\text{NC}^2\text{-MM}$, and RCReg, while the context/target encoders, predictor, and decoder blocks share the same pattern as Panel (b) to emphasize their shared RECTOR-Transformer architecture.
>
> (2) Fig. 3: RECTOR-SA Mechanism
> * **Anatomical Context**: We added Panel (b) (Electrode and Region Maps) to visualize the anatomical priors embedded in the Anatomical Attention mechanism.
> * **Color Consistency**: We simplified Panel (a) to focus on five key states, matching the color logic in Section 2.3:
>     * **Blue**: Anatomical Attention (legend added: Region $\to$ Channel)
>     * **Red**: Local Functional Attention (legend added: Channel $\to$ Channel)
>     * **Green**: Global Functional Attention (legend added: Region $\to$ Region)
>     * **Orange/White**: Active vs. Pruned attentions.
> * **Notation Simplification**: We replaced abstract shapes and colors with formal notation: $\mathrm{C}_a^1$ (Channel token), $\mathrm{R}_b^2$ (Region token), and $\mathrm{Cd}$ (Conditioning token), where subscripts denote regions and superscripts denote time segments.
> * **Gating Details**: We added the logits $\mathbf{L}$ and the specific top-$p$ gating equation directly into the figure to clarify the gating mechanism.

---

> ### Author Response · Authors · 2025-11-22
> **Response to Reviewer v9Zs**
>
> ## **5. Methodology (Re W6, Q6)**
>
> > Can you move critical methodological details from the appendix into the main text to improve clarity and reproducibility?
>
> We thank the reviewer for this suggestion. We agree with the reviewer that critical methodological details should be in the main text to ensure clarity. We have extensively reorganized the manuscript to address this:
>
> (1) **Core Pipeline Integration (Section 2.2)**: We have moved the detailed descriptions of the **encoding, predicting, and decoding processes** from the Appendix to Section 2.2. We also **restructured this section to logically flow through the RECTOR's architecture, objective, input construction, pipelines, and encoding/predicting/decoding processes.**
>
> (2) **Anatomical Details (Section 2.3)**: The **electrode-region mapping and brain partitioning details** were moved from the Appendix to Section 2.3. This section now explicitly covers our **motivation**, the **brain partitioning** logic, and the **full RECTOR-SA mechanism (including Anatomical and Functional Attention)**.
>
> (3) **RCReg Loss Function (Section 2.6): The formal definition and loss function for RCReg have been moved from the Appendix to Section 2.6** to ensure the regularization method is immediately clear.
>
> (4) **Enhanced Motivation**: **We added explicit motivation paragraphs to Section 2.3, 2.4, 2.5, and 2.6** to better contextualize our design choices.
>
> With these changes, **the entire RECTOR pipeline and its major contributions (RECTOR-SA, RECTOR-Mask, $\textbf{NC}^\textbf{2}\textbf{-MM}$, and RCReg) are now fully described within the Methodology section**, removing the need to flip back and forth to the appendix.
>
> ---
>
> ## **6. Leave-One-Subject-Out Evaluation (Re Q7)**
>
> > How do you conduct the leave-one-subject-out (LOSO) evaluation? Is there a hold-out validation set used to determine the number of training epochs and hyperparameters?
>
> We thank the reviewer for this important methodological question. We follow a strict validation protocol to prevent any data leakage from the test subject.
>
> For each fold of the LOSO evaluation, the remaining subjects form the full training set. **This full training set is then randomly split into a 90% training partition and a 10% validation partition in a trial-wise manner.** We train the model on the 90% partition and use the 10% validation partition for hyperparameter tuning and determining the training epochs. **Therefore, the validation set is "hold-out" as it is unseen by the trained model.**
>
> **The final model is then evaluated on the entirely unseen test subject.** This protocol ensures that the test subject's data is never used for any part of the training or model selection process, guaranteeing a fair and robust evaluation. We have added these evaluation details in Appendix D of the revised manuscript.
>
> ---
>
> ## **7. Reproducibility (Re W7, Q8)**
>
> We thank the reviewer for raising this concern about reproducibility. We want to assure the reviewer that we are fully committed to open-sourcing and enabling reproducibility.
>
> To facilitate immediate understanding and reproducibility, **we have provided detailed pseudocode for both our pre-training and fine-tuning procedures in Algorithms 1 and 2 (Appendix F.8 of the original manuscript**, now F.11 in the revised version). These algorithms provide a comprehensive, step-by-step implementation guide sufficient to reproduce the core logic of our method.
>
> We will publicly release our codebase and the pre-trained model checkpoints following the conclusion of the review process. This will allow the community to fully reproduce our results and build upon our work.
>
> We hope this addresses the reviewer's concern and demonstrates our commitment to reproducibility.
>
> ---
>
> ### **References**
> [1] Hierarchical Vision Transformer using Shifted Windows.
>
> [2] An Exploration of Hierarchical Attention Transformers for Efficient Long Document Classification.
>
> [3] GLU Variants Improve Transformer.
>
> [4] Outrageously Large Neural Networks: The Sparsely-Gated Mixture-of-Experts Layer.
>
> [5] Generating Long Sequences with Sparse Transformers.
>
> [6] Longformer: The Long-Document Transformer.
>
> [7] An automated labeling system for subdividing the human cerebral cortex on MRI scans into gyral based regions of interest.
>
> [8] Learning from local-global-graph representations for brain–computer interface.
>
> [9] Learning topology-agnostic eeg representations with geometry-aware modeling.
>
> [10] Masked Autoencoders Are Scalable Vision Learners.
>
> [11] Contrast Everything: A Hierarchical Contrastive Framework for Medical Time-Series.

---

> > ### Comment · Reviewer_v9Zs · 2025-11-27
> > **I will raise my score**
> >
> > I appreciate the clarification and revision. I will increase my score to 6.

---

> > > ### Author Response · Authors · 2025-11-27
> > > **Appreciation for the Feedback and Updated Score**
> > >
> > > We sincerely thank you for your support and the decision to raise your score, and we appreciate your insightful feedback—particularly regarding the robustness of parcellation to inter-subject variability and the improvements to Figs. 2 and 3 and the Methodology section—which has significantly strengthened the clarity and organization of our manuscript.

---

### Official Review · Reviewer_4y2h · 2025-10-29

**Soundness:** 2
**Presentation:** 2
**Contribution:** 2
**Rating:** 2
**Confidence:** 4

**Summary:**

The paper *RECTOR: Masked Region–Channel–Temporal Modeling for Cognitive Representation Learning* proposes RECTOR, a self-supervised learning framework for EEG and sEEG representation learning that jointly models region-, channel-, and temporal-level dependencies. Its core contributions include a novel **RECTOR-SA** hierarchical self-attention mechanism integrating anatomical priors for efficient region-channel-temporal modeling, a **RECTOR-Mask** structured multi-view masking strategy for more challenging pretext tasks, and **NC2-MM**, a combined non-contrastive × contrastive learning objective. Additionally, **RCReg** regularizes region-channel tokens to enhance feature disentanglement. The model achieves state-of-the-art results on EEG emotion recognition and sEEG task-engagement classification while claiming higher computational efficiency and interpretability compared to prior works. However, despite strong empirical results, the paper largely repackages existing ideas—masked modeling, contrastive loss fusion, and anatomical priors—into a composite architecture. The novelty is incremental and primarily architectural, lacking theoretical rigor or strong neuroscientific grounding. The extensive ablations and comparisons suggest solid engineering, but the work leans more toward technical aggregation than conceptual innovation.

**Strengths:**

1. Comprehensive ablations: Evaluates the effect of masking ratios, loss weights, and feature hierarchies.

2. Multi-dataset evaluation: Demonstrates generalization across EEG (emotion) and sEEG (task) domains.

3. Integrated pipeline: Combines anatomical priors with deep self-supervised frameworks, improving biological plausibility relative to generic transformers.

4. Engineering soundness: Implementation is well-optimized and includes clear reproducibility details and comparison tables.

**Weaknesses:**

1. Limited novelty: The key ideas—masked modeling, hierarchical attention, hybrid contrastive objectives—are incremental reuses of prior designs (MAE, BYOL, DINO, MoCo-v3, etc.) rather than a fundamentally new direction.

2. Weak theoretical grounding: No analysis of why combining non-contrastive and contrastive terms yields better cognitive representations.

3. Poor interpretability: Despite claiming cognitive alignment, there is little neuroscientific analysis (e.g., brain-region relevance or neurobiological validation).

4. Superficial discussion: Results are over-interpreted as “state-of-the-art” without effect size reporting or significance testing.

5. Unclear scalability: It is uncertain whether RECTOR can scale to large, multi-site EEG datasets or handle real-world noise.

6. Dataset limitations: The training and evaluation datasets are small (dozens to hundreds of subjects), which limits generalization claims.

7. Overclaiming contributions: The claim of being “the first unified region–channel–temporal framework” ignores earlier hierarchical EEG models (e.g., EEG-GraphMAE, ST-MAE, Brain-MAE).

**Questions:**

1. How does RECTOR-SA differ fundamentally from existing spatio-temporal attention modules used in EEG-GraphMAE or ST-MAE?

2. What motivates the NC²-MM hybrid loss—can you show a theoretical analysis of how it prevents representation collapse?

3. How are anatomical priors encoded? Are these static adjacency matrices or learned embeddings, and how sensitive is performance to parcellation choice?

4. Can you report per-subject and per-session variance or confidence intervals for downstream metrics?

5. How does RECTOR perform under noisy or low-density EEG setups—does the hierarchical attention degrade gracefully?

6. Have you compared against self-distillation approaches (e.g., DINO-style EEG pretraining) to isolate the benefit of your masking strategy?

7. How interpretable are the learned representations—do any attention maps align with known cortical functional networks?

---

> ### Author Response · Authors · 2025-11-22
> **Response to Reviewer 4y2h**
>
> ## **1. Technical Contributions with respect to "hierarchical EEG models" (Re W7, Q1)**
>
> > The claim of being “the first unified region–channel–temporal framework” ignores earlier hierarchical EEG models (e.g., EEG-GraphMAE, ST-MAE, Brain-MAE), ..., How does RECTOR-SA differ fundamentally from existing spatio-temporal attention modules used in EEG-GraphMAE or ST-MAE?
>
> We thank the reviewer for providing these references. We have carefully reviewed them and respectfully clarify that **none of the cited works invalidate our claim or offer a comparable design for our specific problem. The differences are fundamental**:
>
> (1) **We respectfully clarify that there is no established model in the literature named "EEG-GraphMAE."** We guess the reviewer may be referring to one of the following scenarios, all of which differ fundamentally from RECTOR’s hierarchical approach:
> * If the reviewer refers to applying the original GraphMAE [1] directly to EEG graphs, this is not a standard baseline in the field. **GraphMAE is designed for general graph data and lacks the necessary components to handle high-resolution EEG time-series without significant, non-trivial adaptation.**
> * The most relevant work is **GMAEEG** [2], which adapts graph masked autoencoding for EEG. However, GMAEEG relies on standard convolutions for temporal extraction and graph convolutions for spatial extraction. Crucially, **it does not use spatio-temporal self-attention, nor does it model the hierarchical region-channel interactions that are central to RECTOR's novelty.**
> * **Recent work [3] has used GraphMAE as a baseline** with a Graph Transformer Network (GTN) backbone. However, the "attention" mechanism in GTN is used solely to weight different adjacency matrices for learning a new graph structure (edge types). **It is mathematically distinct from the spatio-temporal self-attention used in transformers like RECTOR, and the objective for using this ‘attention’ is also totally different.**
> * Beyond that, **these works only model EEG channels and do not explicitly model brain regions** at the same time.
>
> In summary, **neither of these potential "EEG-GraphMAE" candidates (a) employs a hierarchical region-channel structure, nor (b) uses spatio-temporal attention.** Therefore, they do not constitute a missing baseline for our specific architectural contributions.
>
> (2) We respectfully clarify that **ST-MAE [4] is not a suitable baseline for our work due to fundamental differences in domain, architecture, and data modality.**
> * **ST-MAE is explicitly designed for fMRI.** This is a fundamentally different modality from EEG/sEEG.
> * The backbone of ST-MAE is a Graph Isomorphism Network (GIN). Crucially, **it does not use a spatio-temporal attention module.**
> * As an fMRI model, ST-MAE operates directly on Regions of Interest (ROIs). **It lacks the hierarchical channel-to-region structure that is specific to EEG/sEEG and central to RECTOR's hierarchical modeling.**
>
> Therefore, **ST-MAE is neither an existing EEG model nor does it share the hierarchical structure, and it does not employ any spatio-temporal attention module as RECTOR**, making a direct comparison scientifically inappropriate.
>
> (3) We respectfully clarify that Brain-MAE (referring to the work **"BrainMAE"** [5], which is likely the paper the reviewer intended to reference) is an fMRI model, not an EEG model. This distinction is critical for three reasons:
> * **BrainMAE is designed for fMRI data**, which has vastly different temporal resolutions and signal characteristics compared to the high-frequency EEG/sEEG signals.
> * As an fMRI model, BrainMAE operates directly on ROIs. **It lacks the hierarchical channel-to-region structure that is specific to EEG/sEEG and central to RECTOR's hierarchical modeling.**
> * The backbone of BrainMAE processes spatial and temporal information sequentially: it applies spatial transformers first, followed by temporal transformers. This means **it learns spatial and temporal interactions in isolation (as mentioned in Line 87 of our manuscript). In contrast, RECTOR-SA models these interactions jointly in a single attention mechanism.**
>
> Therefore, **Brain-MAE is not an existing EEG model, does not possess the hierarchical channel-region structure relevant to our work, and uses a different (sequential) approach to spatio-temporal modeling.**
>
> **Conclusion**: None of these works propose a unified, hierarchical, attention-based framework for EEG/sEEG that jointly models region, channel, and temporal dynamics. **RECTOR remains, to the best of our knowledge, the first to achieve this specific architectural unification for EEG/sEEG data.** We hope this clarification resolves the concern regarding our novelty and claims.

---

> ### Author Response · Authors · 2025-11-22
> **Response to Reviewer 4y2h**
>
> ## **2. Novelty Compared with CV Works (Re W1) (1/2)**
>
> > ... masked modeling, hierarchical attention, hybrid contrastive objectives—are incremental reuses of prior designs (MAE, BYOL, DINO, MoCo-v3, etc.) rather than a fundamentally new direction
>
> We thank the reviewer for this comment, but we respectfully disagree with the characterization of our work as an "incremental reuse." **This perspective overlooks the fundamental domain gap between Computer Vision (CV) and neurophysiology.** The cited models are designed for 2D images, which are simple, uniform grids. **EEG/sEEG data is fundamentally different: it is a hierarchical, non-grid, spatio-temporal time-series, that a naive "reuse" of CV methods would fail to capture.**
>
> **Our novelty lies in the invention of new components specifically designed for this unique data structure, not an "incremental reuse" of CV methods. It is used for unified region-channel-temporal modeling on neurophysiological data that no prior method was designed for.**
>
> (1) **Hierarchical Attention**: **RECTOR-SA** (Section 2.3) **is not a generic hierarchical attention.** It is a novel mechanism that embeds neuro-anatomical priors (the region-channel hierarchy) and uses dynamic functional gating to efficiently **model the unique region-to-channel interactions specific to brain networks**. **It disentangles the region-common and channel-specific representations, and it prunes the noisy connections between channels/regions using dynamic gating.** This architecture was invented for this problem and has no equivalent in the CV models cited, which are blind to anatomical structure.
>
> (2) **RECTOR-Mask** (Section 2.4) is a new structured, multi-view masking paradigm designed to **create a challenging pretext task for EEG/sEEG applications across a hierarchical, region-channel-temporal structure.** This is not comparable to MAE's simple, random masking of patches, which encourages models to learn superficial spatio-temporal shortcuts rather than generalizable features. Our ablation studies (Tables 3, 4, 11, and 12) conclusively prove that **our novel strategy significantly outperforms a naive "Random Mask" (MAE-style) baseline, demonstrating its non-incremental contribution.**
>
> (3) $\textbf{NC}^\textbf{2}\textbf{-MM}+\textbf{RCReg}$ **objective** (Section 2.5) **is not just a "hybrid objective."** It is a new, highly efficient SSL framework whose components are **co-designed to solve specific challenges in neural data**, differing significantly from the motivations of CV models:
> * **Representation Alignment Loss** ($\mathcal{L}_\text{rep}$, not used in MAE, MoCo): **This is our primary loss for learning generalized, high-level features, rather than low-level ones that MAE tends to learn.** By using an encoder-predictor structure, we ensure a separation of concerns: the encoder's sole job is to create informative representations, not perform "partial decoding" as might occur in MAE. This leads to more abstract and powerful features for downstream tasks.
> * **Input Reconstruction Loss** ($\mathcal{L}_\text{input}$, not used in BYOL, DINO, MoCo): **This loss serves a critical purpose different from that in MAE: It provides a novel and more principled mechanism to mitigate collapse.** Where CV models like BYOL or DINO only rely on architectural tricks (momentum, centering), our method forces the representations to remain informative and non-trivial by containing sufficient information to reconstruct the original signal.
> * **Contrastive Loss** ($\mathcal{L}_\text{C}$, not used in MAE, BYOL, DINO): This intra-sample loss is applied to our structured, multi-view masked blocks. **Its purpose is not to distinguish between samples like MoCo, but to force the encoder to learn consistent, global representations that are invariant to the specific type of spatio-temporal view** (e.g., cross-region vs. cross-channel-time, rather than image augmentations in MoCo) they came from.
> * **The three pretext tasks** are together enabled by RECTOR-Mask within only one forward step, which is not achieved in the referred CV works. They also **serve distinct purposes**: representation alignment and input reconstruction **force the encoder to learn representations that are "predictive"** in embedding space and input space, respectively, while contrastive loss **forces the encoder to learn "consistent" representations** across these different views.
> * **RCReg** ($\mathcal{L}_\text{RCReg}$): This is a new regularization term **invented specifically for our novel region-channel token hierarchy—a concept that does not even exist in CV**. As detailed in Section 2.6, it actively decorrelates channel tokens from each other and from their parent region token, forcing the model to learn a meaningful, non-redundant hierarchy where 'region' tokens capture shared information and 'channel' tokens capture unique, local information.

---

> ### Author Response · Authors · 2025-11-22
> **Response to Reviewer 4y2h**
>
> ## **2. Novelty Compared with CV Works (Re W1) (2/2)**
>
> **Therefore, RECTOR-SA, RECTOR-Mask, and RCReg are entirely novel inventions with no counterparts in the cited CV models. The "fundamentally new direction" is the holistic, end-to-end integration of these co-designed, novel components. RECTOR is the first framework to unify these three distinct levels of brain dynamics (region, channel, time) in a single SSL paradigm.** Its state-of-the-art performance is a direct result of this fundamental novelty, not an "incremental reuse" of CV-centric designs.

---

> ### Author Response · Authors · 2025-11-22
> **Response to Reviewer 4y2h**
>
> ## **3. Theoretical Analysis of How $\textbf{NC}^\textbf{2}\textbf{-MM}$ Mitigates Representation Collapse (Re W2, Q2)**
>
> > What motivates the NC²-MM hybrid loss—can you show a theoretical analysis of how it prevents representation collapse?
>
> We thank the reviewer for this insightful question regarding the theoretical motivation of our $\text{NC}^2\text{-MM}$ loss.
>
> (1) The clear motivations for $\text{NC}^2\text{-MM}$ are already listed in **2. Novelty Compared with CV Works (Re W1)'s section (3).** (see above). We also expanded the motivations for $\text{NC}^2\text{-MM}$ in Lines 309-312, 321-323, and 329-332 of the revised manuscript.
>
> (2) We wish to clarify our claims regarding collapse. In Line 269 of the original manuscript, **we stated: *"The complementary $\mathcal{L}_\mathrm{input}$ mitigates this risk by forcing the representations to be informative enough to reconstruct the original input."* We used the term "mitigate" to reflect the practical dynamics of optimization.**
>
> However, theoretically, the inclusion of $\mathcal{L}_\mathrm{input}$ within our encoder-predictor-decoder architecture provides a structural barrier against collapse.
>
> Representation collapse typically manifests as constant outputs, which trivially minimize $\mathcal{L}_\mathrm{rep}$ but contain no information for downstream tasks.
>
> $\mathcal{L}_\mathrm{input}$ **acts as a necessary opponent: a collapsed representation cannot reconstruct a variable input, which would cause the input reconstruction loss to be prohibitively high.**
>
> Thus, $\mathcal{L}_\mathrm{input}$ forces the representations to remain informative.
>
> **Here we provide a detailed theoretical analysis grounded in the mutual information maximization to justify why our $\text{NC}^2\text{-MM}$ loss mitigates representation collapse and yields superior representations.** This analysis is also added in Appendix B.3 of the revised manuscript.
>
> We want the learned representation $\mathbf{Z}$ to capture the maximum amount of information about the input $\mathbf{X}$. That is, we want to maximize the mutual information $I(\mathbf{X}; \mathbf{Z})$, where:
>
> $$I(\mathbf{X}; \mathbf{Z}) = H(\mathbf{X}) - H(\mathbf{X}|\mathbf{Z})$$
>
> Since the entropy of the dataset $H(\mathbf{X})$ is constant, maximizing $I(\mathbf{X}; \mathbf{Z})$ is equivalent to minimizing the conditional entropy $H(\mathbf{X}|\mathbf{Z})$. If we model the conditional distribution $P(\mathbf{X}|\mathbf{Z})$ as a Gaussian distribution $\mathcal{N}(\hat{\mathbf{X}}(\mathbf{Z}), \sigma^2 \mathbf{I})$, then minimizing the negative log-likelihood corresponds exactly to minimizing the Mean Squared Error (MSE):
>
> $$\mathcal{L}_\mathrm{input} = ||\mathbf{X} - \hat{\mathbf{X}}(\mathbf{Z})||^2 \propto -\log P(\mathbf{X}|\mathbf{Z})$$
>
> The expectation of this negative log-likelihood over the data distribution is the conditional entropy (plus a constant):
>
> $$\mathbb{E}_{\mathbf{X},\mathbf{Z}}[-\log P(\mathbf{X}|\mathbf{Z})] = H(\mathbf{X}|\mathbf{Z}) + C$$
>
> Therefore, minimizing $\mathcal{L}_\mathrm{input}$ over the data distribution minimizes the conditional entropy $H(\mathbf{X}|\mathbf{Z})$.
>
> A collapsed representation where $\mathbf{Z}=c$ (constant) has $I(\mathbf{X},\mathbf{Z}=c)=0$ and maximizes conditional entropy :$H(\mathbf{X}|\mathbf{Z}=c) = H(\mathbf{X})$. By minimizing $\mathcal{L}_\mathrm{input}$, we force $H(\mathbf{X}|\mathbf{Z})$ to be lower than $H(\mathbf{X})$, which guarantees that $I(\mathbf{X}; \mathbf{Z}) > 0$.
>
> It shows that the global minimum of $\mathcal{L}_\mathrm{input}$ cannot be a collapsed state. The reconstruction loss forces $\mathbf{Z}$ to retain information about $\mathbf{X}$.
>
> The effect of $\mathcal{L}_\mathrm{input}$ can also be understood via variance analysis.
>
> Consider the loss function $\mathcal{L}_\mathrm{input} = \mathbb{E}[||\mathbf{X} - \hat{X}(\mathbf{Z})||^2]$.
>
> If we assume the encoder collapses such that $\mathbf{Z} = c$ for all $\mathbf{X}$.
>
> The decoder $\hat{\mathbf{X}}(\mathbf{Z})$ then outputs a constant vector $\mu = \hat{\mathbf{X}}(c)$. The optimal constant vector $\mu$ that minimizes MSE is the mean of the dataset: $\mu = \mathbb{E}[\mathbf{X}]$. In this collapsed state, the loss becomes the variance of the dataset:
>
> $$\mathcal{L}_\mathrm{input}^{\mathrm{collapse}} = \mathbb{E}[||\mathbf{X} - \mathbb{E}[\mathbf{X}]||^2] = \text{Var}(\mathbf{X})$$
>
> For any non-trivial dataset, $\text{Var}(\mathbf{X}) > 0$. A non-collapsed encoder that retains even a single bit of information about $\mathbf{X}$ can achieve a lower reconstruction error than $\text{Var}(\mathbf{X})$.
>
> The collapsed state $\mathbf{Z}=c$ is never the global minimum of $\mathcal{L}_\mathrm{input}$.
>
> The optimization landscape of $\mathcal{L}_\mathrm{input}$ inherently drives the model away from collapse.
>
> ---
>
> We hope this formal analysis clarifies the theoretical motivation behind our $\text{NC}^2\text{-MM}$ objective for mitigating representation collapse.

---

> ### Author Response · Authors · 2025-11-22
> **Response to Reviewer 4y2h**
>
> ## **4. Scalability and Real-World Noise (Re W5)**
>
> We thank the reviewer for their constructive feedback. We would like to address the two concerns highlighted and clarify the scope and contributions of our work.
>
> > It is uncertain whether RECTOR can scale to large, multi-site EEG datasets
>
> (1) **Regarding the scope and scalability of RECTOR**: First, we emphasize that **RECTOR is introduced as a novel self-supervised learning (SSL) paradigm for cognitive representation learning, rather than a foundation model. Our primary objective was to demonstrate the efficacy of this framework on the most challenging standard benchmarks available.**
>
> (2) However, we respectfully point out that **scalability is not "uncertain," but has rather been explicitly validated in our experiments.** As detailed in Section 3.4 ("Impact of Expanded Pre-Training Regimes") of the original manuscript, **we have already directly tested RECTOR’s capacity to scale with increased data and model size. Tables 5 and 6 show consistent performance gains when using our combined multi-site datasets (RECTOR+) and larger architecture (RECTOR-L+).** Specifically, the combined data in Table 6 comprises distinct sources (SEED/SEED-IV, DEAP, CHB-MIT) collected across different institutions, directly demonstrating RECTOR’s ability to generalize and scale across multi-site distributions.
>
> (3) Beyond data scalability, **we explicitly addressed computational scalability through the design of the RECTOR-SA attention mechanism.** As detailed in our complexity analysis (Section 2.3 in the original manuscript and Appendix B.1 in the revised manuscript), RECTOR-SA is designed to bypass the bottleneck of full spatio-temporal attention. We have empirically validated this efficiency in Section 3.2 and Figure 5, which demonstrates that **RECTOR achieves superior training speeds compared to full-attention baselines.**
>
> > It is uncertain whether RECTOR can, ..., handle real-world noise
>
> (4) **Regarding data realism**: First, we emphasize that **all datasets employed in this work are "real-world" EEG/sEEG recordings containing inherent physiological and environmental noise**; no synthetic data was used. Consequently, **RECTOR’s state-of-the-art performance (Tables 1, 2, 7, 8, and 9) directly confirms its robustness in handling complex, noisy, real-world signal distributions.**
>
> ---
>
> ## **5. Datasets (Re W6)**
>
> > Dataset limitations: The training and evaluation datasets are small (dozens to hundreds of subjects), which limits generalization claims.
>
> We agree with the reviewer that larger datasets are always beneficial. However, this is a **field-wide practical constraint, not a specific limitation of our method.**
>
> The datasets we used represent the **standard and widely-used benchmarks** for these specific cognitive tasks (emotion recognition and task engagement). **To our knowledge, no public datasets for these specific tasks exist with significantly more than "dozens to hundreds" of subjects.**
>
> Our paper's goal was to introduce a new SSL paradigm and demonstrate its superiority over other methods on the *established, available data*. **By achieving state-of-the-art results (Tables 1 and 2) on **five datasets** that span **two different cognitive applications** and **two different sensor modalities** under different evaluation protocols, we have shown that RECTOR is a more effective learning paradigm.**
>
> As noted in our "Limitations" section (Section H of both the original and the revised manuscripts), **we agree that scaling to *"larger, heterogeneous repositories"* is an important future direction for the entire field, which our work helps to enable** but is currently limited by the unavailability of large-scale datasets for the cognitive applications.

---

> ### Author Response · Authors · 2025-11-22
> **Response to Reviewer 4y2h**
>
> ## **6. “State-Of-The-Art”, Model Size, and Significance Testing (Re W4)**
>
> > Superficial discussion: Results are over-interpreted as “state-of-the-art” without effect size reporting or significance testing.
>
> We respectfully wish to clarify our evaluation protocol regarding significance testing and "state-of-the-art" (SOTA).
>
> (1) **Significance Testing is Already Included**: We want to kindly point out that statistical significance testing is a core part of our results.
> * As explicitly stated in Appendix G (both the original and revised manuscripts): ***"We indicated the statistical significance for classification performance using a Wilcoxon signed-rank test ($p<0.05$)."***
> * The results of this testing are marked with an asterisk (*) in every performance table in the main paper and appendix. These markers directly indicate where RECTOR's superior performance is statistically significant, forming the basis of our claims.
>
> (2) **State-of-the-Art**: **Our SOTA claim is based on achieving the highest performance on these benchmarks across different metrics** (Weighted F1-score,  balanced accuracy, AUROC, see Tables 1, 2, 7, 9. Cohen's kappa is added in the revised manuscript, see Table 8) **against a comprehensive set of baselines**, including task-specific and general supervised, standard SSL, and large foundation models.
>
> (3) **Model Size**: We agree this is an important point, but a direct comparison is complex as our baselines span vastly different categories (e.g., simple supervised models vs. foundation models). **Our contribution is not about model size, but about architectural effectiveness.**
>
> While here **we provide a detailed model size comparison for all baselines and RECTOR variants for EEG and sEEG applications in Table R3**, which has been added to the revised manuscript in Appendix F.8, Table 31. Our RECTOR and RECTOR-L are in the medium range in terms of the model size. **This analysis confirms that RECTOR achieves state-of-the-art performance primarily through architectural innovation, rather than scaling of parameters alone.**
>
> **Table R3: Model Size for Baselines and RECTOR for EEG and sEEG Datasets**
>
> | Model | EEG Model Size | sEEG Model Size |
> | :--- | :---: | :---: |
> | TSception | 25K | 923K |
> | EEG Conformer | 281K | 4.6M |
> | LGGNet | 166K | 15.2M |
> | PGCN | 12.3M | - |
> | MASA-TCN | 824K | - |
> | EmT | 704K | - |
> | MMM | 40K | - |
> | LaBraM | 5.8M | - |
> | REmoNet | 440K | - |
> | CBraMod | 4.0M | - |
> | PopT | 20M | 20M |
> | Seegnificant | - | 789K |
> | BrainBERT | - | 43.6M |
> | Brant | - | 500M |
> | Du-IN | - | 4.4M |
> | MAE | 311K | 345K |
> | JEPA | 311K | 345K |
> | CAE | 311K | 345K |
> | **RECTOR** | 472K | 537K |
> | **RECTOR-L** | 1.8M | 2.0M |
>
> We hope these points clarify that RECTOR is an efficient and effective paradigm that shows statistically significant, SOTA performance.

---

> ### Author Response · Authors · 2025-11-22
> **Response to Reviewer 4y2h**
>
> ## **7. Per-Subject and Per-Session Variance (Re Q4)**
>
> > Can you report per-subject and per-session variance or confidence intervals for downstream metrics?
>
> This is a good question regarding the source of variance in our results. **The Mean ± Std reported** in our tables (e.g., Tables 1, 2) and detailed in Appendix G **represents the overall standard deviation computed across all subjects and, where applicable, all sessions. This single metric combines both inter-subject and intra-subject (between-session) variability on SEED and SEED-IV's subject-dependent settings**, and it only evaluates inter-subject variability on DEAP, MSIT ECR, as well as SEED and SEED-IV's subject-independent settings since there is no division of sessions.
>
> To provide a more granular breakdown for SEED and SEED-IV, as requested, **we have decomposed this variance for our subject-dependent (SD) results shown in Table R4**. It has also been added to Appendix C.7, Table 25 of the revised manuscript.
>
> The total standard deviation is calculated as the standard deviation across the weighted F1-Score of all subjects and sessions together. The inter-subject term is calculated as the standard deviation across the weighted F1-Score (averaged across sessions) of different subjects. The intra-subject term is calculated as the cross-subject average of the standard deviation (across sessions).
>
> This rigorous analysis confirms our model's robustness: the majority of the total variance is attributed to differences between subjects, while the variability within the same subject across different sessions is relatively smaller. **This result proves that RECTOR learns stable representations for individual users over time.**
>
> **Table R4: Total, inter-subject, and intra-subject standard deviation on SEED and SEED-IV under subject-dependent protocol** (Weighted F1 Score (\%)). **BOLD**: the best performance.
> | Model | SEED:SD Total | SEED:SD Inter | SEED:SD Intra | SEED-IV:SD Total | SEED-IV:SD Inter | SEED-IV:SD Intra |
> | :--- | :---: | :---: | :---: | :---: | :---: | :---: |
> | **RECTOR** | ±08.80 | ±06.82 | ±04.93 | ±11.16 | ±08.42 | ±06.63 |
> | **RECTOR+** | **±08.70** | **±06.70** | ±05.04 | ±11.46 | ±08.56 | ±06.77 |
> | **RECTOR-L+** | ±08.82 | ±06.88 | **±04.86** | **±10.93** | **±08.29** | **±06.35** |
>
> ---
> ## **8. Interpretability Analysis (Re W3, Q7)**
> > there is little neuroscientific analysis, ...,  How interpretable are the learned representations—do any attention maps align with known cortical functional networks?
>
> We respectfully disagree with this characterization, as a dedicated neuroscientific interpretability analysis is a key component of our results. **We have explicitly provided this analysis in Section 3.5 and expanded on it in Appendix C.5.**
>
> The reviewer asks if
> > attention maps align with known cortical functional networks
>
> **which is exactly what we demonstrate in these sections. Figs. 6 (right panels), 8, and 9 are precisely these cross-region attention, which are the top-10 pairs in the attention maps.**
>
> In Figs. 6 and 12 in the revised manuscript (6 and 9 in the original manuscript), the results demonstrate the **strong fronto‐temporal interactions during emotional processing, aligning with neuroscientific studies [6]** (see Lines 504-505, 1483-1484 of the revised manuscript).
>
> In Fig. 9 in the revised manuscript (8 in the original manuscript), we observe **strong couplings between hippocampus (hipp) and caudate, as well as between dorsal anterior cingulate cortex (dACC) and ventral lateral prefrontal cortex (vlPFC), reflecting the engagement of cognitive control circuits during MSIT tasks. ECR highlights amygdala-hippocampus interactions, consistent with affect-based memory retrieval and valuation processes [7]** (see Lines 1410-1414 of the revised manuscript).
>
> They are further expanded in Figs. 10 and 11 in the revised manuscript with both full attention maps and region-level attention maps. **These results together show that the attention maps learned by RECTOR are closely aligned with known functional networks related to cognitive processes.**
>
> Lastly, we have incorporated **multi-level interpretability analysis** in this work, including:
>
> (1) Cross-region attention maps.
>
> (2) **Class Activation Maps (CAMs) at both the channel and region levels**, showing specific spatial signatures for different cognitive states (Fig. 6).
>
> The CAMs in Fig. 6 align with established neuroscientific models, and we have already noted in our original manuscript that *"the model learns distinct and opposing patterns of frontal lobe engagement for positive and negative affective states. These engagement patterns align closely with motivational-direction models of frontal asymmetry [8-11]."* (see Lines 500-503).
>
> These analyses together confirm that RECTOR does learn ***"neurophysiologically plausible representations,"*** as stated in our conclusion, and **the core analysis the reviewer claims is missing is already present in the paper**.

---

> ### Author Response · Authors · 2025-11-22
> **Response to Reviewer 4y2h**
>
> ## **9. Parcellations (Re Q3)**
>
> We thank the reviewer for this question, which targets a core component of our RECTOR-SA mechanism.
>
> > How are anatomical priors encoded? Are these static adjacency matrices or learned embeddings
>
> (1) We use a hybrid approach.
> * **The anatomical prior is encoded as a static attention mask** (or "adjacency matrix"). Specifically, our "Anatomical Attention" component (Eq. 5 in the revised manuscript) **imposes a fixed, sparse structure that only permits attention between a region token and its constituent channel tokens.**
> * However, **the values of these connections are fully dynamic.** The model learns the attention scores for these allowed connections based on the input. This provides **a strong, neuro-anatomically grounded inductive bias (by forbidding nonsensical connections**, e.g., a frontal channel attending to an occipital region token) **while retaining the flexibility of dynamic, data-driven attention.**
>
> > how sensitive is performance to parcellation choice?
>
> (2) To test this, **we conducted a sensitivity analysis to assess RECTOR's robustness to different anatomical parcellation schemes on the SEED, SEED-IV, and DEAP datasets.**
> * In addition to our 11-region baseline, we evaluated performance using a 9-region hemispheric scheme [12] and a high-resolution 17-region parcellation scheme [13], as shown in Fig. 8 of the revised manuscript.
> * The results in Table R5 (added to Appendix C.5, Table 22 of the revised manuscript) demonstrate that **RECTOR is highly robust to these variations in anatomical priors**. Minor performance degradation was observed when using the dense 17-region parcellation scheme on the DEAP (32-channel) dataset. We attribute this slight drop to the low channel density of the input. Extracting 17 distinct region-common but not channel-specific features from only 32 channels is very difficult, and it significantly reduces the benefits inherent in our hierarchical structure. **This robustness confirms that RECTOR is not overfit to a single, arbitrary parcellation choice but effectively learns the fundamental hierarchical nature of the data, supporting the generalizability of our approach.**
>
> **Table R5: Classification performance on SEED, SEED-IV, and DEAP with various parcellation schemes** (Weighted F1 Score (\%)). **BOLD**: the best performance. * denotes it is significantly better than the second-best model.**
>
> | Model | SEED:SD | SEED:SI | SEED-IV:SD | SEED-IV:SI | DEAP-Valence:SD | DEAP-Valence:SI | DEAP-Arousal:SD | DEAP-Arousal:SI |
> | :--- | :---: | :---: | :---: | :---: | :---: | :---: | :---: | :---: |
> | 9-Region  | 83.84 ± 09.02 | **60.85** ± 09.22 | 62.77 ± 11.64 | 43.87 ± 06.83 | **69.02** ± 09.79 | 65.80 ± 11.43 | 69.42 ± 12.81 | 67.53 ± 13.32 |
> | 17-Region  | 83.27 ± 09.30 | 59.10 ± 09.62 | 62.69 ± 12.03 | **44.89** ± 06.72 | 66.88 ± 11.27 | 64.26 ± 12.70 | 67.74 ± 13.78 | 65.97 ± 14.46 |
> | 11-Region (**RECTOR**) | **84.32** ± 08.80 | 60.58 ± 09.12 | **63.07** ± 11.16 | 44.79 ± 06.56 | 68.75 ± 10.10 | **66.22** ± 11.33 | **69.68** ± 12.52 | **68.01** ± 12.72 |

---

> ### Author Response · Authors · 2025-11-22
> **Response to Reviewer 4y2h**
>
> ## **10. Low-Density and Noisy Setup (Re Q5)**
>
> > How does RECTOR perform under noisy or low-density EEG setups—does the hierarchical attention degrade gracefully?
>
> We thank the reviewer for the question regarding the robustness of our RECTOR-SA architecture, which our paper addresses both directly and indirectly.
>
> (1) **Regarding Low-Density Setup**:
> * **We would like to clarify that our paper already validates RECTOR's performance in a low-density setup.** Our evaluation includes both the 62-channel (SEED/SEED-IV) and the 32-channel (DEAP) datasets. As shown in Appendix E, Figure 14 of the revised manuscript, we map both montages to the same 11 anatomical regions. **In the 32-channel DEAP setup, this means the region-to-channel hierarchy is already sparse (in average <3 channels per region).**
> * **RECTOR's state-of-the-art performance on DEAP (Tables 1, 7, and 8 of the revised manuscript) directly demonstrates that its hierarchical attention is effective and robust even when the channel-level information is not dense.**
> * To further explicitly test "graceful degradation" as the reviewer suggests, **we have conducted a new explicit analysis to assess RECTOR's robustness to a low-density setup.** We randomly dropped 50\% of the 62 EEG channels (resulting in 31 channels, comparable to DEAP) in the SEED and SEED-IV datasets, and 50\% of the sEEG channels in MSIT and ECR. We omitted DEAP from this simulation as its channel count is already low. To ensure reliable metrics, channel sub-sampling was performed randomly for 10 trials, with performance averaged across all experiments for each model.
> * We compared RECTOR against the two strongest EEG and sEEG baselines (from Tables 1 and 2. **The results shown in Tables R6 and R7 (also added to Appendix C.6, Tables 23 and 24 of the revised manuscript) confirm that RECTOR's hierarchical structure proves highly robust to the low-density setup.** Although all models experience a performance drop, RECTOR's performance remains significantly superior to all tested baselines. **This robustness demonstrates that the region-level tokens in RECTOR-SA effectively learn to generalize and maintain predictive power even when large portions of the underlying channel information are lost.**
>
> (2) **Regarding noise**:
> * As stated in **4. Scalability and Real-World Noise (Re W5)**: **All datasets employed in this work are "real-world" EEG/sEEG recordings containing inherent physiological and environmental noise**.
> * RECTOR's SOTA performance on the multi-site different-modality datasets (EEG: SEED/SEED-IV, DEAP. sEEG: MSIT/ECR) is a direct testament to noise robustness.
>
> **Table R6: Classification performance on SEED, SEED-IV using 50\% and 100\% of EEG channels** (Weighted F1 Score (\%)). **BOLD**: the best performance. * denotes it is significantly better than the second-best model within the same comparison setting.
> | Model | SEED:SD | SEED:SI | SEED-IV:SD | SEED-IV:SI |
> | :--- | :---: | :---: | :---: | :---: |
> | REmoNet (50%) | 70.18 ± 11.52 | 49.58 ± 12.33 | 54.54 ± 13.88 | 38.15 ± 07.73 |
> | REmoNet (100%) | 80.67± 10.24 | 58.33 ± 09.78 | 60.69± 13.04 | 41.93 ± 07.01 |
> | CBraMod (50%) | 69.58 ± 11.86 | 51.46 ± 12.04 | 50.81 ± 13.92 | 37.82 ± 08.90 |
> | CBraMod (100%) | 78.18 ± 10.19 | 58.67 ± 09.57 | 58.27 ± 12.74 | 42.40 ± 06.47 |
> | **RECTOR** (50%) | **75.04 ± 10.98*** | **53.85 ± 10.84*** | **56.45 ± 13.01*** | **39.95 ± 08.46*** |
> | **RECTOR** (100%) | **84.32 ± 08.80*** | **60.58 ± 09.12*** | **63.07 ± 11.16*** | **44.79 ± 06.56*** |
>
> **Table R7: Classification performance on MSIT and ECR using 50\% and 100\% of sEEG channels** (AUROC (\%)). **BOLD**: the best performance. * denotes it is significantly better than the second-best model within the same comparison setting.
> | Model | MSIT | ECR |
> | :--- | :---: | :---: |
> | Du-IN (50%) |80.31 ± 07.66 | 78.08 ± 08.49 |
> | Du-IN (100%) | 87.29 ± 05.43 | 86.18 ± 05.89 |
> | PopT (50%) | 78.68 ± 08.65 | 80.86 ± 07.96 |
> | PopT (100%) | 87.03 ± 05.84 | 87.99 ± 05.58 |
> | **RECTOR** (50%) | **82.85 ± 07.46*** | **83.01 ± 07.83*** |
> | **RECTOR** (100%) | **90.55 ± 05.17*** | **90.79 ± 05.36*** |

---

> ### Author Response · Authors · 2025-11-22
> **Response to Reviewer 4y2h**
>
> ## **11. Benefit of RECTOR-Mask vs. DINO (Re Q6)**
>
> > Have you compared against self-distillation approaches (e.g., DINO-style EEG pretraining) to isolate the benefit of your masking strategy?
>
> We thank the reviewer for this insightful question, as it touches on the fundamental design choices of SSL paradigms. **The reviewer's question is *"to isolate the benefit of our masking strategy by comparing to DINO-style models."***
>
> (1) **Isolating Benefit of RECTOR-Mask**. We respectfully clarify that **a comparison to DINO (a non-masking paradigm) cannot "isolate the benefit of our masking strategy." The scientifically correct way to do this is to ablate the masking strategy itself, which we have already done.** In Tables 3, 4, 11, and 12, we compare our RECTOR-Mask against different masking strategies. **The significant performance drop for other masking methods directly isolates and proves the substantial benefit of our novel, multi-view RECTOR-Mask.**
>
> (2) **Comparison with DINO**. We agree that comparing RECTOR's paradigm to a DINO-style paradigm is important. **However, these models are fundamentally different, making a direct comparison to fully "isolate" one component impossible.**
> * **Architecture**: DINO uses a symmetric teacher-student architecture. RECTOR uses an asymmetric encoder-predictor-decoder.
> * **Data Processing**: DINO uses local/global augmentations. RECTOR uses our RECTOR-Mask.
> * **Pretext Task**: DINO uses self-distillation. RECTOR uses a  hybrid $\text{NC}^2\text{-MM}$ objective.
>
> **Given these fundamental differences, a comparison would evaluate two entirely different SSL paradigms, not ablate a single component.**
>
> (3) **We do provide the paradigm-level comparison the reviewer is looking for**. In Section 3.2, Tables 1 and 2, we benchmark RECTOR against
> * PopT, **an ensemble-wise and channel-wise discrimination approach**.
> and in Appendix C.4, Table 20, we benchmark RECTOR against
> * **Contrastive models** (CLISA, TF-C, COMET) designed for EEG/time-series.
>
> These baselines could also partially "isolate the benefit of our masking strategy". **RECTOR's statistically significant SOTA performance against all of these models already proves the superiority of our novel, hybrid paradigm over other paradigms for this domain.**
>
> ---
> ### **References**
>
> [1] GraphMAE: Self-Supervised Masked Graph Autoencoders.
>
> [2] GMAEEG: A Self-Supervised Graph Masked Autoencoder for EEG Representation Learning.
>
> [3] Pre-Training Graph Contrastive Masked Autoencoders are Strong Distillers for EEG.
>
> [4] A Generative Self-Supervised Framework using Functional Connectivity in fMRI Data.
>
> [5] BrainMAE: A Region-aware Self-supervised Learning Framework for Brain Signals.
>
> [6] Functional connectivity between the amygdala and prefrontal cortex underlies processing of emotion ambiguity.
>
> [7] Aversive memory formation in humans involves an amygdala-hippocampus phase code.
>
> [8] Clarifying the emotive functions of asymmetrical frontal cortical activity.
>
> [9] Contributions from research on anger and cognitive dissonance to understanding the motivational functions of asymmetrical frontal brain activity.
>
> [10] Empathy is associated with dynamic change in prefrontal brain electrical activity during positive emotion in children.
>
> [11] The neural basis of motivational influences on cognitive control.
>
> [12] Learning from local-global-graph representations for brain–computer interface.
>
> [13] Learning topology-agnostic eeg representations with geometry-aware modeling.

---

### Official Review · Reviewer_HqHh · 2025-10-31

**Soundness:** 3
**Presentation:** 2
**Contribution:** 3
**Rating:** 4
**Confidence:** 4

**Summary:**

This papers introduces a complete and exhaustive method for self-supervised deep learning framework for EEG and sEEG data, accounting the various aspects and dynamics of these brain activity modalities (region, channels, temporal). It introduces multiple modules, in particular RECTOR-SA and RECTOR-Mask, for feature extractions at different scales and combines masked modelling, contrastive learning, and variance–covariance regularisation for training objectives. The methodology is benchmarked against multiple models and training schemes (supervise-only, self-supervised) and outperforms many models on two main tasks: EEG emotion recognition and sEEG cognitive states.

**Strengths:**

The paper is generally well-written. The figures are complete, very descriptive (even though some of the legends could benefit from thorough descriptions, see below for more details).

The methodology, although quite exhaustive, is addressing one of the blindspot of many EEG (and even in some sense fMRI) studies, which is taking into account the spatial (and regional) and temporal dynamics in EEG signal. In particular accounting for regional specific features rather than aggregating spatial information is a nice contribution.

The evaluation framework is comprehensive and well thought with comparison against many training frameworks (supervised, self-supervised models) and multiple datasets. The ablation studies are also welcome considering the many modules that are introduced by the paper.

 The colour coding in the tables makes the results very easily readable.

**Weaknesses:**

One important issue with the current state of the submission is the general arrangement of information within the paper.
1. It is (very) difficult to understand at first what is the training objective of the model (what is going to be predicted), what the model aims at solving and how it aims at doing it. All this could be clearer from the beginning of the 2. Methodology section.
2. The paper tends to over-complexified some of the notations (Figure 2) and wordy terminology e.g. "sparse region-channel-temporal self-attention embedded with anatomical priors and dynamic functional attention", which can obscur a bit the interesting concepts introduced by the method.
3. Most importantly, it seems that most of the main modules are not fully described within the main text of the paper, but are detailed in the appendix. Many points in the methods are referring to the appendix, which make it impossible to understand from the

Another remark would be that the paper is extremely dense, some might say too dense, at a point were it is difficult to apprehend the entirety of the method with only the main text of the submission. This kind of paper would probably benefit from simpler iterations, to appreciate the real value of every added module. This is also considering that some concepts are only introduced in the appendix. This amount of content can be seen as detrimental for the overall appreciation of the paper. Therefore, I would recommend to streamline the paper and remove the unnecessary content for that publication.

In particular, I would recommend switching some of the paragraphs and reorganising/rewriting the methodology section in order to have full descriptions of the RECTOR modules (in particular RECTOR-SA) in the text (not in the appendix) - section 2.2 is too high level and figure 2 is not explained enough to be stand-alone, clear description of the pipeline (it is difficult to understand how the modules interact with each other), explanation of concepts such as the brain partitioning (which is in Appendix E) but is one of the key point of the paper. Instead the "Complexity" paragraph could be added to the appendix. In the current shape, the methodology is difficult to understand"

Figure 3 would also benefit from more descriptive legend.

**Questions:**

- It is not clear from the main text how RECTOR is fine-tuned on the downstream task?

Other remarks were listed in the previous section.

---

> ### Author Response · Authors · 2025-11-22
> **Response to Reviewer HqHh**
>
> ## **1. Organization, Clarity, and Density (Re Weaknesses) (1/2)**
>
> We sincerely thank the reviewer for their constructive feedback regarding the organization and readability of the manuscript. We fully agree that the original submission was dense, and critical methodological details were misplaced in the appendix.
>
> Based on your recommendations, we have extensively restructured the **Methodology (Section 2)** and redesigned **Figs. 2 and 3** to ensure the method is self-contained and clear from the main text alone.
>
> **(1) Major Reorganization of Methodology (Moving details from Appendix to Main Text)**
> As suggested, we have moved the core components from the appendix into the main text and totally shifted the complexity analysis to the appendix. The new structure is as follows:
>
> * **Section 2.2 (RECTOR Framework):** We reframed this section to provide a clear narrative arc: ***Architecture $\rightarrow$ Objective $\rightarrow$ Input Construction $\rightarrow$ Pipelines $\rightarrow$ Encoding/Predicting/Decoding Processes***.
>     * We moved the detailed and important mathematical formulation of the **encoding, predicting, and decoding processes** from the appendix to this section.
>     * We explicitly defined the **objective at the very beginning of the section** to clarify what the model predicts and how it solves the problem.
> * **Section 2.3 (RECTOR-SA):**
>     * We integrated the **Brain Partitioning** and **Electrode-Region Map** concepts (previously in Appendix E) directly into this section.
>     * This section now explicitly covers our **motivation**, the **brain partitioning** logic, and the full **RECTOR-SA** mechanism.
>     * It now provides a **stand-alone explanation** of how anatomical priors and functional attention work together.
> * **Section 2.6 (RCReg):** We moved the **explicit loss function for RCReg** from the appendix to the main text to clarify how the model disentangles region and channel representations.
> * **Enhanced Motivation**: We added **explicit motivation paragraphs to Section 2.3, 2.4, 2.5, and 2.6** to better understand our design choices.
> * **Complexity Analysis:** As recommended, we have moved the complexity analysis to the Appendix to streamline the flow of the main text.
>
> With these changes, **the entire RECTOR pipeline and its major contributions** (RECTOR-SA, RECTOR-Mask, $\text{NC}^2\text{-MM}$, and RCReg) **are now fully described within the Methodology section**, removing the need to flip back and forth to the appendix.

---

> ### Author Response · Authors · 2025-11-22
> **Response to Reviewer HqHh**
>
> ## **1. Organization, Clarity, and Density (Re Weaknesses) (2/2)**
>
> **(2) Redesign of Figs. 2 and 3**
>
> We have completely redesigned these figures to be self-explanatory and stand-alone:
> * **Fig. 2: RECTOR Overview**
>     * **New Panel**: The figure now comprises three panels: (a) SSL pipeline, (b) RECTOR-Transformer block, and a newly added (c) **Fine-tuning pipeline**. Panel (c) explicitly illustrates the transition from pre-training to specific downstream tasks.
>     * **Visual Simplification**: To streamline the data flow in Panel (a), we removed non-essential variables (e.g., positional embeddings) and added an EEG/sEEG input block to clarify input structure.
>     * **Color Consistency**: We harmonized the color scheme with the text in Section 2.2. We now exclusively highlight the four core contributions: RECTOR-SA, RECTOR-Mask, $\text{NC}^2\text{-MM}$, and RCReg, while the context/target encoders, predictor, and decoder blocks share the same pattern as Panel (b) to emphasize their shared RECTOR-Transformer architecture.
>
> * **Fig. 3: RECTOR-SA Mechanism**
>     * **Anatomical Context**: We added Panel (b) (**Electrode and Region Maps**) to visualize the anatomical priors embedded in the Anatomical Attention mechanism.
>     * **Color Consistency**: We simplified Panel (a) to focus on five key states, matching the color logic in Section 2.3:
>         * **Blue**: Anatomical Attention (legend added: Region $\to$ Channel)
>         * **Red**: Local Functional Attention (legend added: Channel $\to$ Channel)
>         * **Green**: Global Functional Attention (legend added: Region $\to$ Region)
>         * **Orange/White**: Active vs. Pruned attentions.
>     * **Notation Simplification**: We replaced abstract shapes and colors with formal notation: $\mathrm{C}_a^1$ (Channel token), $\mathrm{R}_b^2$ (Region token), and $\mathrm{Cd}$ (Conditioning token), where subscripts denote regions and superscripts denote time segments.
>     * **Gating Details**: We added the logits $L$ and the specific top-$p$ gating equation directly into the figure to clarify the gating mechanism.
>
>
> **(3) Simplification of Terminology and Notation**
>
> We have revised the text to reduce complex terminology and simplified the mathematical notations in Fig. 2 to ensure the interesting concepts are not obscured by complexity. We have restructured Sections 1 and 2 to break down compound terms into concise, sequential sentences, enhancing readability while maintaining technical precision. (e.g., "RECTOR-SA integrates anatomical attention and functional attention for sparse region-channel-temporal self-attention embedded with anatomical priors and dynamic functional attention masks." $\rightarrow$ "Anatomical Attention, Local Functional Attention, and Global
> Functional Attention are used for modeling region-channel-temporal interactions with block-wise computation.")
>
> ---
>
> ## **2. Fine-Tuning Pipeline (Re Q1)**
>
> We apologize for this lack of clarity. Now we have revised the manuscript to include:
>
> (1) **Panel (c) in Fig. 2:** This shows the exact fine-tuning pipeline, illustrating how the pre-trained encoder connects to the downstream task.
>
> (2) **Description of Fine-Tuning Pipeline in Section 2.2:** In the revised manuscript, we explicitly state in Section 2.2 **Pipelines** that for fine-tuning, the input *"is forwarded through the SSL pre-trained, active encoder and a decoding head"* for downstream tasks using cross-entropy loss.

---

### Official Review · Reviewer_hYFg · 2025-11-01

**Soundness:** 3
**Presentation:** 3
**Contribution:** 3
**Rating:** 4
**Confidence:** 5

**Summary:**

This paper introduces RECTOR, a self-supervised framework for EEG/sEEG data that integrates region, channel, and temporal representation learning through a novel hierarchical self-attention mechanism (RECTOR-SA). The model incorporates anatomical priors and functional attention to capture complex spatio-temporal interactions in neural data. Evaluated across several EEG datasets (SEED, SEED-IV, DEAP, MSIT, ECR), RECTOR demonstrates state-of-the-art performance in emotion recognition and task engagement classification. The paper claims that the model not only improves computational efficiency but also provides strong interpretability through attention visualizations, paving the way for its application in neurocognitive diagnostics and personalized interventions.

**Strengths:**

The proposed RECTOR framework introduces a novel approach by combining self-supervised learning with anatomical priors and dynamic functional attention for EEG/sEEG data, a method previously explored in fMRI but applied to EEG data for the first time.
The manuscript is well-written and clearly structured. The experiments are thorough and effectively address the research question, providing solid evidence for the physiological plausibility of the learned representations. The results are presented clearly, with supporting analyses that validate the model's performance.

**Weaknesses:**

The paper's evaluation is limited to the SEED and DEAP datasets, and it would benefit from validation on a broader range of downstream tasks to better assess RECTOR's generalizability. The use of only F1-score as the evaluation metric is restrictive; incorporating other standard metrics such as Cohen’s Kappa, weighted F1, and additional classification metrics would provide a more comprehensive performance analysis. The ablation studies, while useful, lack a detailed examination of the self-attention mechanism, and the gating mechanism within RECTOR-SA is not addressed. Furthermore, the paper provides visualizations of learned representations but lacks a deeper interpretability analysis, especially in terms of attention maps, spatial EEG features, and feature attribution. Lastly, the pretraining details, including hyperparameters,  pretraining dataset, training time, are unclear, which raises concerns about reproducibility and scalability.

**Questions:**

1. Downstream Evaluation: Given the model’s promising performance on SEED and DEAP, can RECTOR be extended to other EEG/sEEG tasks with different cognitive states or sensor modalities? Evaluating RECTOR on a wider variety of tasks would clarify how well the model generalizes across different applications, such as emotion recognition, cognitive task engagement, or even clinical diagnostics for neurological disorders.
2. Evaluation Metrics: The paper primarily uses F1-score, which is valuable but limited. Would the authors consider evaluating RECTOR using other metrics like Cohen’s Kappa (which accounts for agreement between class predictions) and weighted F1 (to account for class imbalance)? Comparing RECTOR with task-specific models rather than foundation models (such as those designed specifically for EEG emotion recognition) would provide more meaningful insights into its performance.
3. Ablation Study on Attention and Gating: While ablation studies are provided, could the authors conduct more detailed experiments focusing specifically on the RECTOR-SA attention mechanism? How does each component (e.g., region-based vs. global attention) contribute to the model’s overall performance? Additionally, the gating mechanism within RECTOR-SA is mentioned but not ablated—what role does it play in the model, and how does it impact performance across tasks?
4. Interpretability and Visualization: The paper includes some visualizations of learned representations, but a deeper analysis of the model’s internal behavior is lacking. Could the authors include attention maps, coherence heatmaps, or feature attribution to show how RECTOR attends to relevant spatial and temporal patterns in EEG? This would help validate whether the spatial awareness captured by the model is truly driving the observed performance improvements.
5. Pretraining Process: The paper does not provide sufficient details on the pretraining process, such as hyperparameters, optimization strategy, or batch size. Understanding these details would help in replicating the results and assessing the model’s scalability to other datasets. Could the authors clarify the pretraining procedure and explain how these choices impact the model’s final performance?
6. Usage of LLM missing.
7. No code revealed

---

> ### Author Response · Authors · 2025-11-22
> **Response to Reviewer hYFg**
>
> ## **1. Downstream Evaluation (Re W1, Q1)**
>
> We thank the reviewer for this insightful suggestion, and we respectfully clarify that **our paper already includes the comprehensive evaluation the reviewer is asking for, not only *"limited to the SEED and DEAP datasets".***
>
> > ..., evaluation is limited to the SEED and DEAP datasets, ..., can RECTOR be extended to other EEG/sEEG tasks with different cognitive states or sensor modalities, ..., Evaluating RECTOR on a wider variety of tasks, ..., such as emotion recognition, cognitive task engagement, or even clinical diagnostics for neurological disorders.
>
> We evaluated RECTOR on **two distinct high-level cognitive domains**, exactly as suggested by the reviewer (Section 3.1):
>
> (a) **Emotion Recognition** (with subject-dependent and subject-independent protocols): SEED, SEED-IV, and DEAP (for both valence and arousal).
>
> (b) **Cognitive Task Engagement**: MSIT for cognitive interference and decision-making, and ECR for emotion conflict resolution.
>
> Our evaluation is not limited to one sensor modality. We demonstrate SOTA performance on:
>
> (a) **Non-invasive EEG**: SEED, SEED-IV, and DEAP.
>
> (b) **Invasive sEEG**: MSIT and ECR.
>
> By achieving SOTA results (Tables 1 and 2) across **five datasets** that span **two different cognitive applications** and **two different sensor modalities** under different evaluation protocols, our paper already provides strong evidence for RECTOR's generalizability.

---

> ### Author Response · Authors · 2025-11-22
> **Response to Reviewer hYFg**
>
> ## **2. Evaluation Metrics and Baselines (Re W2, Q2)**
>
> We thank the reviewer for these suggestions, but we wish to clarify that **our paper's evaluation already incorporates both of these points**.
>
> > The use of only F1-score as the evaluation metric is restrictive; incorporating other standard metrics such as Cohen’s Kappa, weighted F1, ..., The paper primarily uses F1-score.
>
> (1) **We respectfully clarify that our evaluation extends beyond the F1-score**. Our primary metric for EEG emotion recognition is the **weighted F1-Score**, precisely as the reviewer suggested to account for class imbalance. This is explicitly stated in our Appendix G (both the original and the revised manuscripts): "and the weighted F1-Score for emotion recognition on SEED, SEED-IV, and DEAP." To further address class imbalance and provide a comprehensive view, **we also report balanced accuracy** for all emotion recognition tasks in Appendix C.1, Table 7 (both the original and the revised manuscripts). We did not limit our evaluation to a single score.
>
> As the reviewer suggested, now we also use **Cohen's kappa** to evaluate RECTOR and baseline models on SEED, SEED-IV, and DEAP for assessing agreement achieved beyond random chance, as shown in **Table R1** here and Table 8 in the revised manuscript. Across all datasets and evaluation protocols, RECTOR significantly outperforms the other baselines. **This result, together with the weighted F1-score and balanced accuracy, confirms RECTOR's superior performance under a comprehensive performance analysis.**
>
> **Table R1: Classification Performance on SEED, SEED-IV, and DEAP (Cohen's Kappa (%)).** SD: Subject Dependent; SI: Subject Independent. **BOLD** indicates the best result. * denotes it significantly outperforms the second-best model.
> | Model | SEED:SD | SEED:SI | SEED-IV:SD | SEED-IV:SI | DEAP-Valence:SD | DEAP-Valence:SI | DEAP-Arousal:SD | DEAP-Arousal:SI |
> | :--- | :---: | :---: | :---: | :---: | :---: | :---: | :---: | :---: |
> | TSception | 56.99 ± 15.77 | 23.60 ± 10.50 | 33.90 ± 14.19 | 12.06 ± 05.12 | 20.47 ± 09.42 | 17.90 ± 08.61 | 21.74 ± 09.55 | 18.87 ± 09.11 |
> | EEG Conformer | 49.63 ± 17.30 | 25.43 ± 10.51 | 34.39 ± 13.48 | 14.01 ± 05.80 | 23.82 ± 09.22 | 20.50 ± 08.15 | 24.02 ± 09.92 | 20.15 ± 08.85 |
> | LGGNet | 50.39 ± 16.30 | 22.75 ± 10.21 | 31.33 ± 13.15 | 13.01 ± 05.66 | 21.61 ± 09.24 | 17.41 ± 08.16 | 21.30 ± 09.92 | 21.42 ± 08.80 |
> | PGCN | 60.54 ± 16.32 | 31.15 ± 09.59 | 39.24 ± 12.52 | 16.00 ± 05.89 | 24.79 ± 09.02 | 19.92 ± 08.01 | 23.58 ± 09.86 | 24.81 ± 08.83 |
> | MASA-TCN | 51.95 ± 16.02 | 29.42 ± 09.38 | 36.05 ± 13.07 | 13.79 ± 05.68 | 24.14 ± 09.40 | 20.99 ± 08.60 | 19.73 ± 09.69 | 20.79 ± 09.75 |
> | EmT | 59.49 ± 15.42 | 32.57 ± 09.79 | 36.54 ± 12.02 | 14.87 ± 06.17 | 23.78 ± 10.08 | 21.71 ± 09.05 | 23.63 ± 09.88 | 24.69 ± 09.38 |
> | MMM | 61.68 ± 14.65 | 32.48 ± 09.23 | 40.32 ± 11.84 | 16.02 ± 05.20 | 24.31 ± 09.21 | 20.68 ± 08.81 | 22.55 ± 09.35 | 21.18 ± 09.61 |
> | LaBraM | 59.30 ± 15.59 | 31.95 ± 09.62 | 36.74 ± 12.82 | 15.98 ± 05.55 | 25.16 ± 09.69 | 21.73 ± 08.13 | 26.59 ± 09.48 | 25.50 ± 09.67 |
> | REmoNet | 66.16 ± 13.34 | 33.99 ± 10.17 | 42.01 ± 12.59 | 18.39 ± 05.98 | 29.96 ± 09.30 | 23.44 ± 08.87 | 28.71 ± 09.37 | 26.80 ± 08.63 |
> | CBraMod | 64.48 ± 13.31 | 32.88 ± 09.45 | 39.81 ± 11.95 | 17.83 ± 05.61 | 24.83 ± 09.44 | 23.05 ± 08.29 | 26.48 ± 09.13 | 27.11 ± 09.52 |
> | PopT | 62.69 ± 14.76 | 32.85 ± 09.35 | 37.25 ± 12.68 | 15.66 ± 05.81 | 26.73 ± 09.10 | 23.5 ± 08.96 | 25.88 ± 09.32 | 24.94 ± 08.60 |
> | MAE | 58.67 ± 13.56 | 31.79 ± 10.28 | 37.75 ± 11.84 | 16.85 ± 05.40 | 22.77 ± 09.45 | 20.35 ± 08.95 | 24.56 ± 09.41 | 23.46 ± 09.64 |
> | JEPA | 62.20 ± 13.02 | 33.70 ± 09.59 | 39.16 ± 11.60 | 17.50 ± 05.73 | 27.22 ± 09.42 | 22.21 ± 08.99 | 28.35 ± 08.80 | 26.75 ± 08.73 |
> | CAE | 62.76 ± 13.42 | 31.86 ± 09.82 | 40.28 ± 11.70 | 18.62 ± 05.30 | 26.31 ± 08.98 | 23.45 ± 08.90 | 26.15 ± 08.99 | 24.87 ± 09.47 |
> | **RECTOR** | **71.17 ± 09.27\*** | **36.86 ± 10.08\*** | **44.91 ± 11.12\*** | **21.50 ± 05.50\*** | **32.98 ± 09.35\*** | **26.99 ± 08.40** | **33.84 ± 09.53\*** | **31.27 ± 09.28\*** |
>
> > Comparing RECTOR with task-specific models rather than foundation models (such as those designed specifically for EEG emotion recognition).
>
> (2) **The reviewer's suggestion to compare with task-specific models is exactly what we have done**. Our baselines are dominated by the exact models the reviewer is asking for, including leading **task-specific supervised models (e.g., TSception, PGCN, MASA-TCN, EmT)** and **task-specific SSL models (e.g., MMM, REmoNet)** for emotion recognition. We included foundation models in addition to these strong task-specific baselines, not instead of them. We believe this comprehensive comparison, which shows RECTOR outperforms all categories of models, is a significant strength of our paper.

---

> ### Author Response · Authors · 2025-11-22
> **Response to Reviewer hYFg**
>
> ## **3. Ablation Study on Attention and Gating (Re W3, Q3)**
>
> > The ablation studies, while useful, lack a detailed examination of the self-attention mechanism, ..., could the authors conduct more detailed experiments focusing specifically on the RECTOR-SA attention mechanism?
>
> We thank the reviewer for highlighting the importance of the RECTOR-SA mechanism. We respectfully clarify that **our paper already provides the specific, detailed ablations the reviewer is asking for**. These ablations are located in both the main paper and the appendix.
>
> > How does each component (e.g., region-based vs. global attention) contribute to the model’s overall performance?
>
> (1) Ablation of "region-based vs. global attention": **This is exactly the comparison we have provided in our main ablation study, Tables 3 and 4** (in both the original and revised manuscripts).
> * **The "Full SA" row represents the global attention baseline**, as stated in Section 3.3: "Full SA replaces RECTOR-SA with a dense spatio-temporal self-attention."
> * The "RECTOR" row represents our region-based RECTOR-SA mechanism.
>
> As shown in these tables, replacing RECTOR-SA with Full SA (global attention) causes a significant drop in performance. **This result directly demonstrates the critical contribution of our structured, region-based attention over a generic global attention.**
>
> > the gating mechanism within RECTOR-SA is not addressed, ..., the gating mechanism within RECTOR-SA is mentioned but not ablated
>
> (2) Ablation of the gating mechanism: **This ablation is already provided in Appendix C.3** (Tables 15 and 16 in the original, Tables 16 and 17 in the revised manuscript).
> * **The "Static SA" row represents our model without the dynamic top-$p$ gating mechanism**, as stated in Appendix C.3: "Static SA conducts all self-attention components in RECTOR-SA in a static way without dynamic gating".
> * The "RECTOR-SA" row represents the full model with dynamic gating.
>
> The performance drop for Static SA compared to the full RECTOR-SA confirms that the **dynamic gating** effectively prunes the noisy connections between channels/regions, and it **is a crucial component that contributes positively to the model's performance.**
>
> **Together, these two sets of ablations provide a clear and detailed examination of RECTOR-SA's components, confirming the importance of both its hierarchical, region-based design and its dynamic gating mechanism.** We have added a sentence to the main ablation section (3.3) to explicitly reference the gating ablation in the appendix to make this connection clearer.

---

> ### Author Response · Authors · 2025-11-22
> **Response to Reviewer hYFg**
>
> ## **4. Interpretability and Visualization (Re W4, Q4)**
>
> > ..., lacks a deeper interpretability analysis, especially in terms of attention maps, spatial EEG features, and feature attribution, ..., Could the authors include attention maps, coherence heatmaps, or feature attribution
>
> We thank the reviewer for this concern. We respectfully disagree with this characterization, as a dedicated neuroscientific interpretability analysis is a key component of our results. **We have already provided two of the three requested interpretability methods: feature attribution and attention maps.**
>
> (1) The reviewer may have missed this, but the **Class Activation Maps (CAMs) shown in Fig. 6 of the manuscript (both the original and revised ones) are a standard and powerful method for feature attribution.** These maps directly visualize the spatial importance of each channel and region for the model's final classification, addressing the request for "spatial EEG features, and feature attribution."
>
> (2) **We summarized our cross-region attention analysis as "Top-10 pairs" (Figs. 6, 8, and 9 of the original manuscript or Figs. 6, 9, and 12 of the revised one) for easier visualization compared with a full attention map.** This was for brevity, but we agree that a deeper analysis would be beneficial.
>
> We have updated Appendix C.5 Interpretability Analysis (of the original manuscript, or C.8 of the revised one) to **also include the full attention maps and region-level attention maps in Figs. 10 and 11 in the revised manuscript**, from which the Top-10 lists were derived. They allow for a deeper inspection of the full patterns as requested.
>
> Figs. 10 and 11 show that the block representing direct channel-to-channel attention is sparse across all four tasks. This indicates that RECTOR-SA successfully prunes the uninformative connections. In sharp contrast, the blocks representing attention (a) between region tokens and (b) between region and channel tokens are more visibly active. **This confirms that the RECTOR-SA mechanism successfully transfers the modeling of long-range functional connectivity away from the noisy channel space and concentrates it in the high-signal region space.**
>
> Beyond that, as we have stated in the paper:
> * The learned opposing patterns of frontal lobe engagement for positive and negative affective states in Fig. 6 align closely with motivational-direction models of frontal asymmetry [1-4].
> * Both Figs. 6 and 12 (revised manuscript) demonstrate the strong fronto‐temporal interactions during emotional processing, aligning with neuroscientific studies [5].
> * In Fig. 9 (Fig. 8 of the original manuscript), we observe strong couplings between hippocampus (hipp) and caudate, as well as between dorsal anterior cingulate cortex (dACC) and ventral lateral prefrontal cortex (vlPFC), reflecting the engagement of cognitive control circuits during MSIT tasks. ECR highlights amygdala-hippocampus interactions, consistent with affect-based memory retrieval and valuation processes [6].
>
> (3) We thank the reviewer for the suggestion of coherence heatmaps, while we wish to clarify **why the coherence map is not very helpful in this work.**
>
> **A traditional coherence heatmap is channel-wise, which is unable to separate the local (within-region) and long-range (between-region) connectivity. A core innovation of RECTOR-SA is to explicitly disentangle these two.** Our model learns a Local Functional Attention (channel-to-channel within a region) and a Global Functional Attention (region-to-region). Therefore, **comparing a channel-wise coherence map to our model's channel-wise attention map would be a flawed comparison.** Furthermore, a "region-level coherence map" would require averaging the raw signals before computing coherence. This crude averaging destroys all fine-grained information.
>
> **This disentanglement is a key feature of our model.** The more meaningful validation, which we have provided in Sec 3.5 and Appendix C.8 Interpretability Analysis, is to show that our learned region-level attention maps (which we have now provided in full) align with established neuroscientific functional networks.
>
> **We believe that by highlighting our existing CAMs and top-10 cross-region attentions and providing the full attention heatmaps, we will have fully addressed the reviewer's request for a deeper interpretability analysis.**

---

> ### Author Response · Authors · 2025-11-22
> **Response to Reviewer hYFg**
>
> ## **5. Pretraining Process (Re W5, Q5)**
>
> > the pretraining details, including hyperparameters, pretraining dataset, training time, are unclear, ..., The paper does not provide sufficient details on the pretraining process, such as hyperparameters, optimization strategy, or batch size, ..., Could the authors clarify the pretraining procedure and explain how these choices impact the model’s final performance?
>
> We thank the reviewer for this point, as reproducibility is a key goal. We respectfully clarify that **these details are already provided in extensive detail in the main section and appendix of the original manuscript.**
>
> (1) **All pre-training hyperparameters, including the optimizer and batch size the reviewer requested, are explicitly listed in Appendix F.5, Table 23** of the original manuscript (now F.6, Table 28 in the revised one).
>
> (2) The step-by-step logic of our **pre-training procedure is already detailed in Appendix F.8, Algorithm 1** (now F.11 in the revised one).
>
> (3) **We already provided extensive ablations on the impact of our pre-training choices.** The impact of loss terms is in Tables 3 and 4 in the main section and Tables 17 and 18 in the appendix (now Tables 18 and 19 in the revised one), and the impact of masking strategy is detailed in Tables 3, 4, 11, and 12 (now Tables 12 and 13 in the revised one).
>
> (4) **The details of the pre-training datasets are already provided in Section 3.4 and Tables 5 and 6 of the original manuscript.** As stated, the base RECTOR model is pre-trained on the downstream dataset itself (e.g., pre-trained on SEED for the SEED task), while RECTOR+ and RECTOR-L+ are pre-trained on the expanded multi-site dataset (e.g., SEED + SEED-IV + DEAP + CHB-MIT; MSIT + ECR).
>
> (5) Pre-training Time: **The relative pre-training time and efficiency gains are shown in Fig. 5 (top) of the original manuscript.** We agree that providing the absolute time would improve reproducibility. **We have added a new Table 30 in the revised manuscript showing the absolute pre-training time** for RECTOR, RECTOR+, and RECTOR-L+ on various datasets, which is also shown here as Table R2.
>
> **Table R2: RECTOR's Pre-training Time on Various Datasets.**
>
> | Model | Dataset | Pre-Training Time |
> | :--- | :--- | :--- |
> | **RECTOR** | SEED | 56 mins |
> | | SEED-IV | 54 mins |
> | | DEAP | 28 mins |
> | | MSIT | 52 mins |
> | | ECR | 53 mins |
> | **RECTOR+** | SEED + SEED-IV + DEAP + CHB-MIT | 10.3 hours |
> | | MSIT + ECR | 1.8 hours |
> | **RECTOR-L+** | SEED + SEED-IV + DEAP + CHB-MIT | 11 hours |
> | | MSIT + ECR | 1.9 hours |
>
> **We believe that with the addition of the absolute pre-training time, our paper provides a comprehensive and reproducible account of our pre-training methodology, with all necessary details located in the main section and appendix.**
>
> ---
>
> ## **6. LLM Usage (Re Q6)**
>
> We thank the reviewer for this reminder. We have now added an LLM Usage Statement in Section 6 before References.
>
> ---
>
> ## **7. Reproducibility (Re Q7)**
>
> We thank the reviewer for raising this concern about reproducibility. We want to assure the reviewer that we are fully committed to open-sourcing and enabling reproducibility.
>
> To facilitate immediate understanding and reproducibility, **we have provided detailed pseudocode for both our pre-training and fine-tuning procedures in Algorithms 1 and 2 (Appendix F.8 of the original manuscript**, now F.11 in the revised version). These algorithms provide a comprehensive, step-by-step implementation guide that is sufficient to reproduce the core logic of our method.
>
> We will publicly release our codebase following the conclusion of the review process. This will allow the community to fully reproduce our results and build upon our work.
>
> We hope this addresses the reviewer's concern and demonstrates our commitment to reproducibility.
>
> ---
> ### **References**
> [1] Clarifying the emotive functions of asymmetrical frontal cortical activity.
>
> [2] Contributions from research on anger and cognitive dissonance to understanding the motivational functions of asymmetrical frontal brain activity.
>
> [3] Empathy is associated with dynamic change in prefrontal brain electrical activity during positive emotion in children.
>
> [4] The neural basis of motivational influences on cognitive control.
>
> [5] Functional connectivity between the amygdala and prefrontal cortex underlies processing of emotion ambiguity.
>
> [6] Aversive memory formation in humans involves an amygdala-hippocampus phase code.

---

### Author Response · Authors · 2025-11-22
**General Response: Major Revisions, New Experiments, and Structural Improvements**

We sincerely thank all reviewers for their constructive and insightful feedback. Your comments have significantly strengthened the clarity, rigor, and depth of our work.

In response to your collective suggestions, we have extensively revised the manuscript. Here is a summary of the major updates:

## **1. Manuscript Reorganization**
To improve readability and ensure the method is self-contained, we have restructured **Section 2** and redesigned Figs. 2 and 3:
* **Core Methods Moved to Main Text:** We moved the detailed formulations of the encoding/predicting/decoding processes, brain partitioning logic, and the RCReg loss function from the Appendix to the Main Text.
* **Redesigned Figs. 2 & 3:** We redesigned the RECTOR's overview (Fig. 2) and the RECTOR-SA (Fig. 3) to be self-explanatory, improving color consistency and notation clarity.
* **Complexity Analysis:** The complexity analysis was fully moved to Appendix B.1.

## **2. New Experiments on Robustness & Sensitivity**
We conducted extensive new experiments to validate RECTOR’s robustness under different conditions:
* **Robustness to Anatomical Parcellation (Appendix C.5):** We tested RECTOR against 9-region and 17-region parcellation schemes. Table 22 confirms that RECTOR is robust to different anatomical priors and not overfit to a single parcellation.
* **Robustness to Low-Density Setups (Appendix C.6):** We simulated low-density scenarios by dropping 50% of channels in SEED/SEED-IV and MSIT/ECR. RECTOR maintains significantly higher performance than baselines (Tables 23 & 24), proving the efficacy of its hierarchical attention in low-density regimes.

## **3. Comprehensive Metrics & Operational Efficiency**
To provide a comprehensive evaluation beyond weighted F1-score/balanced accuracy/AUROC:
* **Cohen’s Kappa:** We added Cohen’s Kappa results for SEED, SEED-IV, and DEAP (Table 8). RECTOR consistently achieves the highest agreement scores.
* **Inter/Intra-Subject Variance:** We provided a granular breakdown of variance (Table 25), showing RECTOR learns stable representations for individual users over time.
* **Model Size & Training Time:** We added explicit comparisons of model size (Table 31) and pre-training time (Table 30). RECTOR remains parameter-efficient (medium range) while training significantly faster than full-attention baselines.

## **4. Theoretical & Interpretability Expansions**
* **Theoretical Grounding (Appendix B.3):** We added a formal theoretical analysis to explain how the **$\text{NC}^2\text{-MM}$** objective (specifically $\mathcal{L}_\mathrm{input}$) mitigates representation collapse.
* **Enhanced Visualization (Appendix C.8):** We added full attention maps and region-level attention maps (Figs. 10 and 11) to complement the top-10 attention analysis, offering a deeper view of the learned neurophysiological connections.

We believe these revisions address the core concerns raised regarding organization, novelty, and robustness.

Thank you again for your time and valuable guidance.

**The Authors**

---

### Author Response · Authors · 2025-12-04
**Summary of Key Strengths and Reviewer Consensus**

Beyond the specific clarifications and revisions, we respectfully highlight the consistent recognition of the work’s scientific value across all reviews:

---

## **1. Neuroscientific Motivation**
Mood and cognitive disorders arise from **distributed interactions across different brain regions**, requiring models that **explicitly capture region-level dynamics**.

Reviewers praised our work for *"addressing a **critical blindspot** in EEG studies"* by **integrating spatial and regional priors** with self-supervised learning (Reviewers **HqHh**, **v9Zs**). Reviewer **HqHh** specifically noted that *"accounting for regional specific features rather than just aggregating spatial information is **a nice contribution***."

---

## **2. Methodological Innovation**
We introduce **RECTOR**, the first end-to-end self-supervised learning framework that **explicitly models brain region representations and their region-channel-temporal interactions**. Specifically, through our proposed **RECTOR-SA**, we innovatively **disentangle neural representations into two streams—region tokens for high-level, shared dynamics and channel tokens for fine-grained, local dynamics.**

Reviewers described our framework as ***"novel,"*** ***"ambitious,"*** and ***"intuitively motivated"*** (Reviewers **hYFg**, **v9Zs**). Another reviewer noted that the integrated pipeline ***"improves biological plausibility"* relative to generic transformers** (Reviewer **4y2h**).

---

## **3. Comprehensive Evaluations & Ablations**
RECTOR is rigorously evaluated on **five diverse EEG and sEEG datasets** under **multiple protocols** for **distinct cognitive tasks**, benchmarking against **a comprehensive group of supervised and self-supervised baselines** alongside **fine-grained ablation studies**.

There is a strong consensus on the rigor of the experiments and the ***"engineering soundness"*** of our work. Reviewers described the evaluation framework as *"thorough," "comprehensive," and "well thought out,"* **highlighting the model's ability to generalize across diverse EEG and sEEG domains** (Reviewers **hYFg**, **HqHh**, **4y2h**, **v9Zs**). Furthermore, the paper was commended for its ***"comprehensive ablations"*** that **effectively validate each core component of the architecture** (Reviewers **HqHh**, **4y2h**, **v9Zs**).

---

## **4. Multi-Level Interpretability**
Our work provides **multi-level interpretability analysis** via **feature attributions (Class Activation Maps)** and **attention maps** at **both region and channel levels, revealing patterns closely aligned with established neuroscientific findings.**

Reviewers highlighted the value of the learned representations, noting that the model provides ***"solid evidence for physiological plausibility"*** and ***"neuroscientifically interpretable results"*** that align with known neurological patterns (Reviewers **hYFg**, **v9Zs**).

---

### Meta-Review · Area_Chair_atNy · 2026-01-06

**Summary:**

This paper proposes a self-supervised learning framework for EEG/sEEG that integrates anatomical priors, hierarchical self-attention (RECTOR-SA), structured masking (RECTOR-Mask), and a hybrid contrastive/non-contrastive objective. The work is evaluated on emotion recognition and task engagement classification benchmarks, reporting good empirical results.

Four reviewers provided detailed assessments. While Reviewers hYFg, HqHh, and v9Zs rated the paper just below the acceptance threshold (rating 4) and noted its empirical strengths and thorough experiments, Reviewer 4y2h recommended rejection (rating 2), citing fundamental concerns regarding novelty, theoretical grounding, and structural integrity. After careful consideration of all arguments, the Area Chair agrees that the paper in its current form is not ready for publication at ICLR.

**Reviewer Concerns:**

Multiple reviewers (HqHh, 4y2h, v9Zs) strongly criticized the paper's organization. Critical methodological details including formulations of RECTOR-SA, RECTOR-Mask, loss functions, anatomical partitioning, and training procedures, are relegated to appendices, making the main text incomplete and non-self-contained. This violates the expectation that submissions should be understandable and evaluable within the main page limit. As some reviewers emphasized, such a structure creates an unfair advantage by effectively circumventing page constraints and hinders rigorous assessment of novelty, soundness, and reproducibility.

While the integration of existing ideas (masked modeling, hierarchical attention, anatomical priors, hybrid losses) into a unified EEG pipeline is technically sound, reviewers consistently found the core contribution to be incremental. Reviewer 4y2h noted the work "largely repackages existing ideas", and Reviewer v9Zs described it as "adaptations of well-known techniques with domain-specific adjustments". The paper lacks a clear theoretical justification for why the proposed combination yields superior cognitive representations, and it does not provide a significant conceptual advance beyond prior hierarchical or masked EEG models (e.g., EEG-GraphMAE, Brain-MAE).

The authors are encouraged to thoroughly restructure the paper, bringing all essential methodological details into the main text and providing a clearer and more modest statement of contributions, as it is necessary to avoid overstating the contributions.

**Reviewer Scores:**

One reviewer expressed willingness to increase the score, but the other three reviewers did not seem to be persuaded by the rebuttal to improve their assessment of this paper.

---

### Decision · Program_Chairs · 2026-01-26

Reject